# *SLC45A4* is a pain gene encoding a neuronal polyamine transporter

Steven J. Middleton[1,14], Sigurbjörn Markússon[2,3,14], Mikael Åkerlund[4], Justin C. Deme[5], Mandy Tseng[1], Wenqianglong Li[1], Sana R. Zuberi[1], Gabriel Kuteyi[2,3], Peter Sarkies[2], Georgios Baskozos[1], Jimena Perez-Sanchez[1], Adham Farah[1], Harry L. Hébert[6], Sylvanus Toikumo[7,8], Zhanru Yu[9,10], Susan Maxwell[1], Yin Y. Dong[1], Benedikt M. Kessler[9,10], Henry R. Kranzler[7,8], John E. Linley[11], Blair H. Smith[6], Susan M. Lea[5], Joanne L. Parker[2,3], Valeriya Lyssenko[4,12,13], Simon Newstead[2,3,14 ✉] & David L. Bennett[1,14 ✉]

Polyamines are regulatory metabolites with key roles in transcription, translation, cell signalling and autophagy[1]. They are implicated in multiple neurological disorders, including stroke, epilepsy and neurodegeneration, and can regulate neuronal excitability through interactions with ion channels[2]. Polyamines have been linked to pain, showing altered levels in human persistent pain states and modulation of pain behaviour in animal models[3]. However, the systems governing polyamine transport within the nervous system remain unclear. Here, undertaking a genome-wide association study (GWAS) of chronic pain intensity in the UK Biobank (UKB), we found a significant association between pain intensity and variants mapping to the *SLC45A4* gene locus. In the mouse nervous system, *Slc45a4* expression is enriched in all sensory neuron subtypes within the dorsal root ganglion, including nociceptors. Cell-based assays show that SLC45A4 is a selective plasma membrane polyamine transporter, and the cryo-electron microscopy (cryo-EM) structure reveals a regulatory domain and basis for polyamine recognition. Mice lacking SLC45A4 show normal mechanosensitivity but reduced sensitivity to noxious heat- and algogen-induced tonic pain that is associated with reduced excitability of C-polymodal nociceptors. Our findings therefore establish a role for neuronal polyamine transport in pain perception and identify a target for therapeutic intervention in pain treatment.

Chronic pain affects around one in five of the adult population and has a major negative impact on quality of life, with profound socio-economic consequences[4,5]. Indeed, when all pain conditions are taken into account, chronic pain is the leading cause of disability worldwide[6]. Unfortunately, current treatments are often inadequate due to poor efficacy and tolerability[5,7]. The molecular causes of chronic pain are not fully defined, although enhanced excitability of nociceptors (sensory neurons that detect injurious stimuli) and the central nociceptive circuits to which they project are important determinants of sensitized pain states[8,9].

One group of endogenous metabolites that has been proposed to contribute to chronic pain are the polyamines, which include putrescine (Put), spermine (Spm) and spermidine (Spd). These are ubiquitous polycationic alkylamines that have important roles in the synthesis and stability of nucleic acids, cell signalling (including the stress response) and growth, but they can also modulate ion channel function[1]. Serum

and tissue polyamine levels are raised in pain states such as inflammation and rheumatoid arthritis[10,11]. Injection of polyamines into the paw[3] or spinal intrathecal space[12] elicits pain behaviour in rodents, and blocking their synthesis has been reported to reduce inflammatory pain[13]. However, it is unclear from previous studies as to the direction of effect, locus of action or how best to intervene therapeutically. The regulation of ion channels by polyamines is critically dependent on their location (intracellular versus extracellular)[14,15], so understanding how these metabolites are transported is essential in determining the effect on neuronal function. Furthermore, excess polyamines are toxic, necessitating tight cellular control over homeostasis to preserve cellular function; however, how this is achieved in the body remains unclear[16].

To date, only intracellular polyamine transport systems have been identified, which include the lysosomal polyamine transporter ATP13A2 (PARK9)[17], which is linked to Parkinson's disease, and the vesicular polyamine transporters ATP13A3 and SLC18B1 (VPAT)[18,19], leaving the

[1]Nuffield Department of Clinical Neurosciences, The University of Oxford, Oxford, UK. [2]Department of Biochemistry, University of Oxford, Oxford, UK. [3]The Kavli Institute for Nanoscience Discovery, University of Oxford, Oxford, UK. [4]Department of Clinical Sciences, Lund University Diabetes Centre, Lund University, Lund, Sweden. [5]Center for Structural Biology, Center for Cancer Research, National Cancer Institute, Frederick, MD, USA. [6]Chronic Pain Research Group, Division of Population Health and Genomics, Ninewells Hospital and Medical School, University of Dundee, Dundee, UK. [7]Mental Illness Research, Education and Clinical Center, Crescenz VA Medical Center, Philadelphia, PA, USA. [8]Department of Psychiatry, University of Pennsylvania Perelman School of Medicine, Philadelphia, PA, USA. [9]Chinese Academy of Medical Sciences Oxford Institute, University of Oxford, Oxford, UK. [10]Target Discovery Institute, Centre for Medicines Discovery, Nuffield Department of Medicine, University of Oxford, Oxford, UK. [11]Neuroscience, BioPharmaceuticals R&D, AstraZeneca, Cambridge, UK. [12]Department of Clinical Science, University of Bergen, Bergen, Norway. [13]Steno Diabetes Center Copenhagen, Herlev, Denmark. [14]These authors contributed equally: Steven J. Middleton, Sigurbjörn Markússon, Simon Newstead, David L. Bennett. ✉e-mail: Simon.newstead@bioch.ox.ac.uk; David.bennett@ndcn.ox.ac.uk

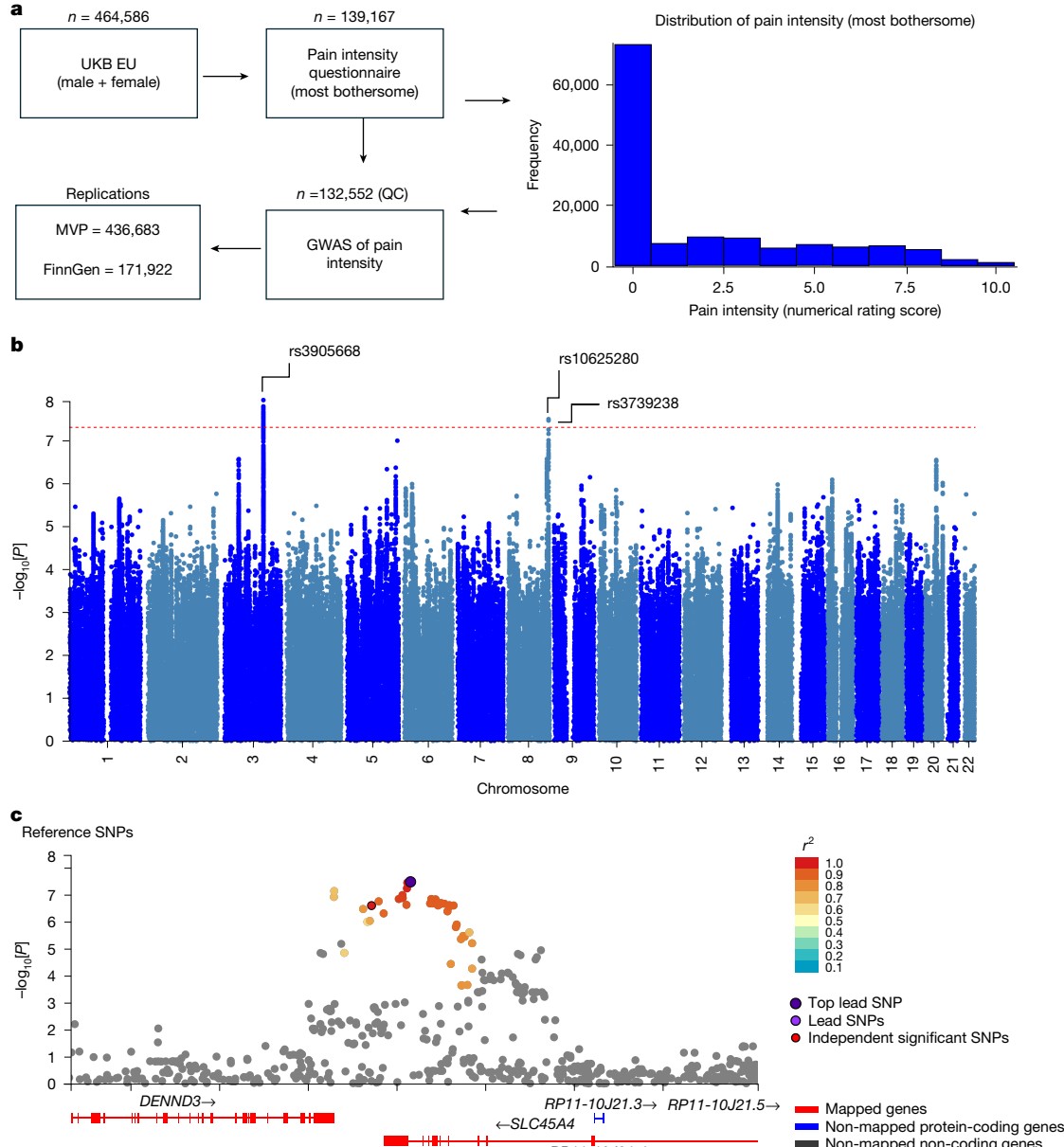

**Fig. 1 | UKB pain intensity (most bothersome) GWAS identifies novel signals.** **a**, The frequency distribution of pain intensity scores among participants, measured using the numerical rating scale. Pain scores range from 0 (no pain) to 10 (worst possible pain). QC, quality control. **b**, In a Manhattan plot from the UKB pain intensity GWAS (*n* = 132,552 participants), the genome-wide significant threshold is demarcated by a horizontal red line at $-\log_{10}$-transformed unadjusted $P = 5 \times 10^{-8}$, obtained by linear regression testing for the association of the residualized phenotype and genetic markers using two-step Regenie.

**c**, A regional plot centred on rs10625280, the principal SNV within the *SLC45A4* gene, elucidates associations within this genomic region in the UKB pain intensity GWAS (*n* = 132,552). SNVs are differentiated by a colour gradient on the basis of their linkage disequilibrium ($r^2$ values) with the top independent significant SNV. The top lead SNVs in genomic risk loci, lead SNVs and independent significant SNVs are distinctly marked—encircled in black and highlighted in dark purple, purple and red, respectively. SNPs, single-nucleotide polymorphisms.

identity of the plasma membrane polyamine system unknown. Here we demonstrate that *SLC45A4* encodes a plasma membrane polyamine transporter that is genetically linked to chronic pain in the human population, presenting an opportunity to understand the regulatory network linking polyamine biosynthesis and neuronal excitability.

## *SLC45A4* SNVs are linked to human pain

Pain has a substantial heritable component that varies depending on the subtype of pain studied, but the median heritability in twin studies is 36% (ref. 20). To investigate the genetic factors, we undertook a GWAS using data from the enhanced pain phenotyping questionnaires administered in the 2019 UKB[21], enabling accurate case definition of chronic pain[22]. As the outcome measure, we used reported pain intensity on a numerical rating scale for the most bothersome chronic pain (*n* = 132,552, European ancestry; the pain distribution is shown in Fig. 1a). We identified a total of 29 genome-wide significant single-nucleotide variants (SNVs) associated with pain intensity, including two independent loci with lead SNVs (Fig. 1b; details are provided in Supplementary Table 1, and the top 100 GWAS results are shown in Supplementary Table 2): rs3905668, chromosome 3:136212744 (based on the GRCh38 human reference genome) ($P = 1.22 \times 10^{-8}$), which is intergenic and near the *MSL2* gene, and rs10625280, chromosome 8:141213072 (GRCh38) ($P = 3.37 \times 10^{-8}$), which maps to the *SLC45A4* gene.

We focused on the *SLC45A4* locus (Fig. 1c) given that it encodes a solute carrier (SLC) transporter, the function of which was unclear (although it had previously been proposed as a sucrose–proton transporter[23]). rs10625280 is situated within an intron of the *SLC45A4* gene (Fig. 1c), with a subset of SNVs demonstrating linkage disequilibrium ($R^2 > 0.6$), localizing to the same genomic region. One of these is a missense variant in *SLC45A4* ($r^2 = 0.98$, rs3739238, chromosome 8: 141212346 (GRCh38), p.Asn718Asp, $P = 3.63 \times 10^{-8}$) that was significantly associated with pain intensity. Moreover, both SNVs were identified as expression quantitative trait loci (eQTL) for *SLC45A4* and *DENND3* across several databases, including Open Targets Genetics, EyeGEx, GTEx and eQTLGen database. The locus-to-gene (L2G) analysis pipeline prioritized *SLC45A4* with an overall L2G score of 0.86 (rs3739238)[24].

Having identified a genetic link between *SLC45A4* and pain perception in the UKB data, we then analysed data from the Million Veteran Program (MVP) in which a recent multiancestry genetic study of pain intensity in approximately 600,000 veterans identified 125 independent genetic loci, one of which was *SLC45A4* (ref. 25) (the greater number of independent loci identified in the MVP compared with in our UKB study may reflect the increased statistical power due to a larger cohort size and the multiancestry approach). The top *SLC45A4* SNV identified in MVP was rs59918340, chromosome 8: 141222157 (GRCh38). The direction of association was dependent on ancestry. A review of the SNVs identified in our UKB GWAS of chronic pain intensity successfully replicated both the lead SNV rs10625280 ($P = 1.21 \times 10^{-8}$) and missense variant rs3739238 ($P = 7.24 \times 10^{-9}$) in the European ancestry GWAS of pain intensity in MVP (Supplementary Table 1). The Finnish Genetics project (FinnGen)[26] GWAS also confirmed the association of these two variants with pain (Supplementary Table 1). The direction of effect was consistent across different databases and studies, underscoring the robustness of these genetic associations.

We used a phenome-wide association analysis (pheWAS) to examine secondary phenotypes linked to the genetic variants rs10625280, rs3739238 and the *SLC45A4* gene, using data from the UKB, FinnGen and the GWAS Catalogue. rs10625280 showed a significant association with 25 different traits, including skeletal (osteoarthritis) and several immunological traits concerning blood cell parameters (Supplementary Table 3 shows traits with significant association after Bonferroni correction). rs3739238 showed a significant association with 35 traits, spanning immunology, skeletal (osteoarthritis), metabolism, psychiatry and reproduction (Supplementary Table 3). The *SLC45A4* gene demonstrated associations with 55 distinct traits across a variety of health domains such as skeletal, reproductive, psychiatric, immunological, cardiovascular, neurological and metabolic functions (Supplementary Table 3). Moreover, the Open Targets Genetics L2G pipeline[24] identified a strong association between *SLC45A4* and multisite chronic pain ($P = 2 \times 10^{-8}$), highlighting its potential role in pain perception.

## SLC45A4 is a polyamine transporter

We next sought to understand the function of SLC45A4 at the molecular level. On the basis of distant homology to the plant sucrose transporter (SUC) in *Arabidopsis thaliana*, SLC45A4 was proposed as a proton-coupled sucrose transporter[23]. However, the human SLC45A4 protein shares only about 26% sequence identity with the *A. thaliana* SUC1 transporter, which, coupled with recent studies reporting conflicting functions for this protein in metal-ion transport[27], leaves the role of SLC45A4 in the cell an enigma. We therefore sought to identify possible substrates using a correlation analysis between metabolomics and expression datasets, which can predict substrates for SLC proteins[28]. Our analysis highlighted that the expression of SLC45A4 across more than 1,000 cell lines in the Cancer Cell Line Encyclopedia correlated positively to the levels of γ-aminobutyric acid (GABA) (Extended Data Fig. 1a). However, neither GABA nor sucrose elicited a change in the thermal stability of purified SLC45A4 (Fig. 2a and Extended Data Fig. 1b),

prompting us to consider alternative metabolites. Although GABA is commonly synthesized from glutamate by glutamate decarboxylase enzymes[29], an alternative pathway is available through the degradation of the polyamine Put[30]. We therefore tested a panel of substrates involved in GABA synthesis through the arginine–ornithine–Put (AOP) pathway. Notably, we observed a marked decrease in thermal stability in the presence of biogenic amines, with the polyamines Spm and Spd eliciting the largest response (Fig. 2a and Extended Data Fig. 1b). Cell-based radioactive uptake assays further confirmed that SLC45A4 functions as a non-selective polyamine transporter with an optimum pH of 7.5–8.5 (Fig. 2b,c and Extended Data Fig. 1d,e), with no inhibition observed in the presence of either L-lysine, L-ornithine or L-arginine (Extended Data Fig. 1c). Moreover, our data reveal that SLC45A4 is also selective in polyamine recognition. The longest polyamine, Spm, has an half-maximum inhibitory concentration ($IC_{50}$) of 240 μM, followed by Spd (123 μM), with the smaller polyamines Put and cadaverine (Cad) having the highest affinity (74 μM and 67 μM, respectively) (Fig. 2d). Although it has long been recognized that polyamines have a fundamental role in modulating neuronal excitability[31], the identity of a specific neuronal biogenic amine transporter has remained unclear. Our results identify SLC45A4 as a plasma membrane polyamine transporter.

## Cryo-EM structure of SLC45A4

To understand the structural basis for polyamine transport, we determined the structure of human SLC45A4 using cryo-EM. We determined the structure in detergent and lipid nanodiscs to 2.80 Å and 3.25 Å, respectively (Fig. 2e, Supplementary Table 4 and Extended Data Figs. 2 and 3). The structures are similar with a root mean squared deviation of 0.32 Å over 425 Cα atoms (Extended Data Fig. 4a). SLC45A4 consists of 12 transmembrane-spanning α-helices that adopt the canonical major facilitator superfamily (MFS) fold, consisting of two six-helical bundles in an inward open state[32] (Fig. 2f,g). A unique feature of SLC45A4 is the presence of a large 25.5 kDa cytoplasmic domain inserted between transmembrane helix 6 (TM6) and TM7. Although the majority of this domain is disordered in the volume, we observe clear density between Arg414 and Gln462 that extends into the canonical MFS-binding site formed between the two six-helical bundles and plugs the transporter in an autoinhibited state by packing against the intracellular gating helices TM4 and TM5, and TM10 and TM11 (Fig. 2e and Extended Data Figs. 3e and 4a). The plug domain (414–462) forms an extended coil that places two conserved basic side chains, Lys450 and Arg453, into a solvated central cavity within the transporter domain that is notable for the extreme negatively charged surface, which facilitates the anchoring of the plug domain into the cavity (Extended Data Fig. 4b). Lys450 interacts with Asp173, which together with Asp169 and Glu176 forms an acidic ladder on TM4, and also interacts through a cation–π interaction with Tyr672 (TM11) (Fig. 2h and Extended Data Fig. 4c). Arg453 interacts with Tyr66 (TM1) and makes a cation–π interaction with Trp519 (TM7), which, together with Tyr66 and Tyr672, is one of several conserved aromatic side chains within the binding site. Arg453 also interacts with a water-molecule network linking the side chain to Asp169. Additional interactions between the plug domain and the transporter are also observed. Notably, Ser452 hydrogen bonds with Glu176, while Tyr459 interacts with Arg180 (TM4), further stabilizing the plug domain against the intracellular gating helices in the transporter. We noticed that, in the nanodisc structure, the membrane thins by about 9 Å where the plug domain enters the transporter (Extended Data Fig. 4d). Previous structures of the lysosomal polyamine transporter ATP13A2 proposed a mechanism of membrane destabilization to facilitate polyamine exit into the cytoplasm[33]. Our structure suggests that a similar mechanism for polyamine release may operate within SLC45A4. Finally, Asn718 (the GWAS missense variant) is located on the cytoplasmic side of the membrane (Extended Data Fig. 4e). Substitution of Asn718 to Ala, Asp, Arg or Trp did not change transport activity (Extended Data Fig. 4e).

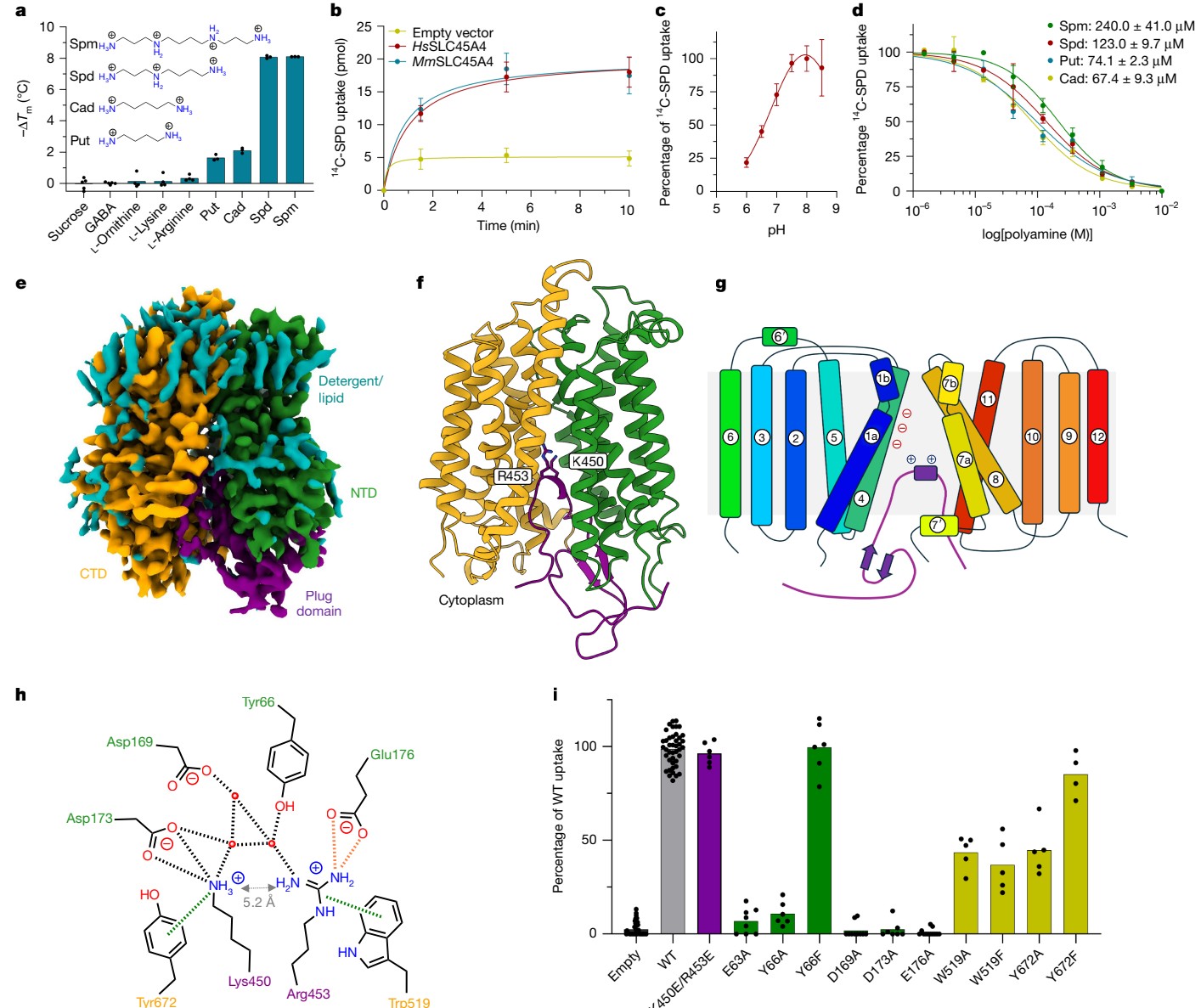

**Fig. 2 | SLC45A4 is a polyamine transporter with a plug domain. a**, Thermal destabilization of SLC45A4 in the presence of metabolites in the AOP pathway. $n$ = 3–5 thermal stability assays. **b**, Time course of $^{14}$C-SPD uptake in neuronal N2 cells overexpressing either human (*Homo sapiens*, *Hs*) or mouse (*Mus musculus*, *Mm*) SLC45A4 in comparison to an empty vector control. $n$ = 11 wells containing cells for each condition. **c**, Comparison of SLC45A4 activity in neuronal N2 cells under different external pH values. $n$ = 3–4 wells containing cells per pH condition. **d**, Polyamine competition of $^{14}$C-SPD uptake into neuronal N2 cells. Data are the calculated mean ± s.d. half-maximal inhibitory concentration values from three independent experiments. Inset: IC$_{50}$ fits; $n$ = 5 wells containing cells per condition. **e**, Cryo-EM density of human SLC45A4 in LMNG/CHS detergent, contoured at a threshold level of 0.25. CTD, C-terminal domain;

NTD, N-terminal domain. **f**, Cartoon representation of SLC45A4. The two six-TM-helix bundles of the MFS fold are coloured by domain and the plug domain, which inserts in between the two, is shown in purple. **g**, Topology diagram of SLC45A4, coloured blue to red from the amino terminus. **h**, Schematic of the interactions between Lys450 and Arg453 in the plug domain and the polyamine binding site in SLC45A4. Hydrogen bonds and salt bridges are shown as black dashed lines, cation–π bonds as green dashes and the charge interaction between Arg453 and Glu176, only observed in the nanodisc structure, as orange dashes. Residues are coloured by domain as in **e** and **f**. **i**, $^{14}$C-SPD uptake in neuronal N2 cells overexpressing SLC45A4 and mutants of residues shown in **h**. $n$ = 4–46 wells containing cells. All data are mean ± s.d. Exact $n$ values and details of replicates are provided in the Methods.

Common variants identified in GWASs have small effect sizes that can be difficult to detect in functional assays (which were undertaken in the NEURO2A cell line); alternatively, this variant might affect another aspect of SLC45A4 function in the cell.

Efforts to capture a substrate-bound state using both detergent and nanodisc samples proved to be unsuccessful. However, lysine and arginine contain primary and secondary amine groups like polyamines. The positioning of these side chains within the canonical MFS-binding site implies that the plug domain functions as a pseudosubstrate to

autoinhibit the transporter. Notably, the distance between the amines in the Lys450 and Arg453 side chains is around 5 Å (Fig. 2h and Extended Data Fig. 4c), which is close to the length of the higher-affinity substrates Put and Cad (Fig. 2a), suggesting that polyamines may interact similarly with the binding site. However, substituting Lys450 and Arg453 with glutamic acid did not affect the activity of SLC45A4 (Fig. 2i), demonstrating that these side chains do not have a role in PA transport and suggesting a regulatory role for the plug domain. Nevertheless, we used the interactions made by Lys450 and Arg453 to guide our mutational analysis.

The acidic ladder on TM4 is both strictly conserved within mammalian SLC45A4 transporters (Extended Data Fig. 1g) and essential for polyamine uptake (Fig. 2i), with alanine variants showing no detectable transport in our cell-based assay. Glu63 (TM1), another conserved acidic side chain that sits close to Glu176, is also essential. In the nanodisc structure, Arg453 adopts a different rotamer and interacts with Glu63 (Extended Data Fig. 4c), suggesting that this side chain also interacts with polyamine substrates along with the side chains in the acidic ladder. However, whereas Tyr66Ala and Tyr672Ala mutants showed reduced activity, conservative mutations to phenylalanine showed wild-type (WT) activity levels, suggesting that bulky hydrophobic side chains at these positions facilitate transport. Trp519 was less critical for SPD recognition, as mutating it to either alanine or phenylalanine only reduced function to around 50% compared with the WT levels. The types of interactions that we observed within the binding site of SLC45A4 are consistent with those reported for the lysosomal polyamine pump ATP13A2 (PARK9)[33] and polyamine binding protein, PotD, from *Escherichia coli*[34] (Extended Data Fig. 4f), suggesting that these three structurally distinct proteins use a combination of ionic and cation–π interactions to recognize and transport polyamines across the membrane.

## SLC45A4 is enriched in sensory neurons

Little is known about the expression of SLC45A4 along the pain pathway, so we aimed to characterize its expression in neural tissues. Published single-cell RNA-sequencing (RNA-seq) datasets of the mouse nervous system show that *Slc45a4* mRNA is broadly expressed but with a preponderance in dorsal root ganglia (DRG) sensory neuron subtypes[35–38] (Extended Data Fig. 5). Using quantitative PCR with reverse transcription (RT–qPCR) in WT C57BL/6 mice, we confirmed that *Slc45a4* expression was enriched in the DRG across the sensory neuraxis (Fig. 3a). In situ hybridization analysis of DRG sections showed that almost all DRG neurons expressed *Slc45a4* (Fig. 3b) and it was expressed by all subtypes— peptidergic nociceptors (neuropeptide CGRP⁺), non-peptidergic nociceptors (isolectin B4 binding), C-low threshold mechanoreceptors (tyrosine hydroxylase⁺) and myelinated mechanoreceptors and proprioceptors (neurofilament heavy chain, NF200⁺) (Extended Data Fig. 6), with higher expression in large-diameter (NF200⁺) afferents. We also confirmed *Slc45a4* mRNA expression in the mouse spinal cord, with high ventral-horn expression and limited dorsal-horn (laminae III and IV) expression (Extended Data Fig. 6). Existing data also show expression of *SLC45A4* in human sensory neurons (Extended Data Fig. 5). When electroporated into DRG neurons, eGFP-tagged SLC45A4 is trafficked to the plasma membrane (Fig. 3c), consistent with a role in membrane transport.

## SLC45A4 loss disrupts polyamines

To examine the link between SLC45A4 function and pain perception, we generated homozygote *Slc45a4⁻/⁻* knockout (KO) mice using CRISPR–Cas9 technology (Fig. 3d and Extended Data Fig. 7). The mice were born with no observable defects or perinatal mortality; however, we did note a lower-than-expected frequency of homozygous KO mice (Supplementary Table 5), suggesting possible early embryonic lethality. Weight gain was normal (Supplementary Table 5); however, we noted a transient white hair phenotype resulting in a 'salt and pepper' appearance of the coat that appeared on around day 20 and then faded and was absent by day 70 (Extended Data Fig. 7). This finding is identical to that reported in an independent *Slc45a4*-KO mouse (in which exon 5 was ablated)[39] and was proposed to be due to a transient defect in melanoblast differentiation; indeed, Spd has been shown to enhance melanin production[40].

Having identified SLC45A4 as a polyamine transporter, we investigated the effect of *Slc45a4* deletion on polyamine levels in the mouse nervous system. We found all polyamines were abundant in the brain,

spinal cord and DRG of WT mice (Fig. 3f). Notably, the loss of *Slc45a4* resulted in a reduction in spinal Spd and an increase in DRG Put (Fig. 3f). We also assessed polyamine levels in blood plasma (an extracellular compartment). All polyamines were detected in WT and KO plasma, but Spd was elevated in *Slc45a4*-KO plasma (Fig. 3f), indicative of altered transport between compartments. Together, these data highlight that polyamine homeostasis is disrupted when *Slc45a4* is ablated.

As polyamine levels have an important role in normal cell growth[1], we assessed sensory neuron anatomy. There was no loss or change in sensory neuron subpopulation distributions (Fig. 3g,h), and the intraepidermal nerve fibre density of all small fibres (PGP9.5⁺) or in the subpopulation of peptidergic small fibres (CGRP⁺) was normal in both glabrous and hairy skin in *Slc45a4*-KO mice compared with in WT mice (Fig. 3i,j). We also observed normal hair follicle innervation of NF200⁺ afferent lanceolate endings and CGRP⁺ circumferential high-threshold mechanoreceptors (Circ-HTMRs; Fig. 3k) in *Slc45a4*-KO mice. These findings show no major anatomical phenotype in our mouse model regarding sensory neuron survival or axon outgrowth.

## SLC45A4 loss affects pain behaviour

We undertook behavioural testing in mice from the age of 10 weeks when hair colour had normalized, and the experiments could be performed in a blinded manner. No deficits were observed in open-field behaviour tests of locomotion and exploration (Extended Data Fig. 8a–e). However, during a rotarod task of motor performance, in which mice were challenged to run on a rotating rod that steadily increased in speed (rpm), the latency to fall from the rota rod and the maximum final speed were both increased in KO mice relative to in WT (Fig. 4a), indicative of increased motor endurance. Anatomically, neuromuscular junctions in *Slc45a4*-KO mice were normal (Extended Data Fig. 9a,b). We next tested sensory behaviour. The reflex withdrawal responses of the mouse hindpaw were assessed in response to mechanical stimuli. The withdrawal threshold to a light touch stimulus (using von Frey hairs; Extended Data Fig. 8f) or the latency to withdraw from a noxious pin prick stimulus (Fig. 4b) were unchanged in *Slc45a4*-KO mice. However, *Slc45a4*-KO mice showed an increased latency to respond (that is, hyposensitivity) to being placed on a 48 °C (Fig. 4c) or 50 °C (Fig. 4d) hot plate, compared with WT and heterozygous mice. The mouse sensitivity to a 53 °C hot plate and a Hargreaves radiant heat source was unchanged (Extended Data Fig. 8g). However, when assessing the time spent in different regions of a thermal-gradient apparatus, we observed a marked difference in the distribution of the time spent in temperature zones, with KO and heterozygous mice shifting their preference towards warmer temperatures more than WT mice (Fig. 4e). There was no difference in the reflex withdrawal latency to dry ice (noxious cold) (Extended Data Fig. 8h). Our results demonstrate an altered thermal coding in *Slc45a4*-KO mice that appears specific for heat sensation over cold. Finally, we examined the nocifensive behaviours (paw lifting, licking and shaking) in response to intraplantar administration of the algogen formalin, as a model of tonic pain. This behaviour occurs in two phases. The first phase is thought to primarily relate to direct activation of nociceptors through the ligand-gated ion channel TRPA1 (refs. 41,42), and the second phase occurs due to spread of this activation and also spinal sensitization. *Slc45a4*-KO mice showed a significant reduction in their nocifensive behaviours during the first phase of the formalin response, compared with WT and heterozygous mice (Fig. 4f,g).

## SLC45A4 tunes nociceptor excitability

Polyamines are involved in GABA synthesis, so we wanted to determine whether GABA levels were altered in our KO mice, which might explain some of our findings. GABA levels in the brain were normal in *Slc45a4*-KO mice, but spinal GABA was significantly reduced compared

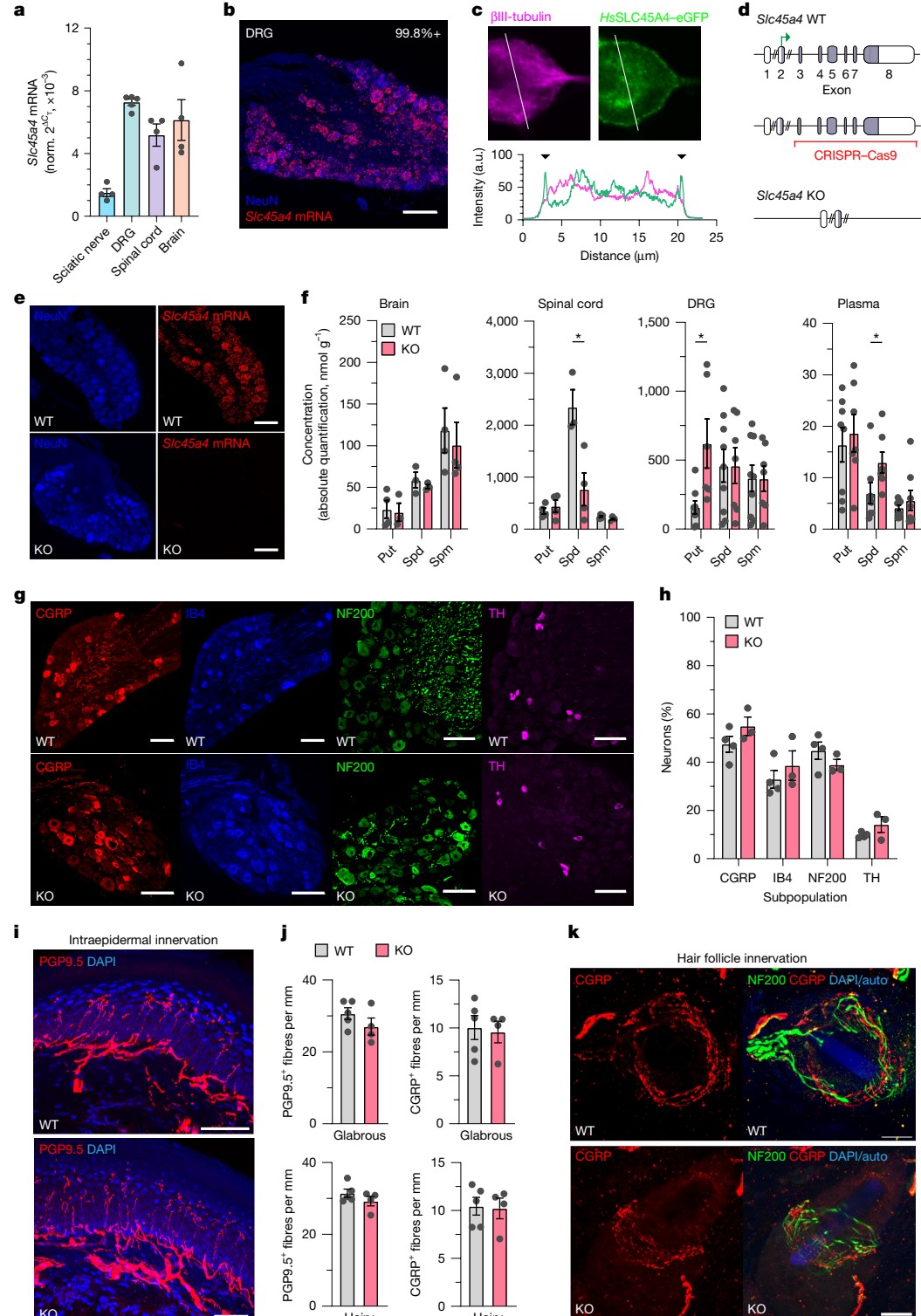

**Fig. 3 | Genetic ablation of *Slc45a4* results in dysregulation of polyamines that does not alter nociceptor anatomy. a**, qPCR analysis of *Slc45a4* mRNA along the sensory neuraxis. From left to right, sciatic nerve, DRG, spinal cord and brain; *n* = 4, 5, 4 and 4 mice, respectively. Norm., normalized. **b**, *Slc45a4* mRNA is expressed in all NeuN+ mouse DRG neurons. *n* = 3 mice, 1,777 cells. **c**, Human SLC45A4–eGFP transfected into mouse sensory neurons localizes to the plasma membrane (observed in more than three independent experiments). **d**, *Slc45a4*-KO strategy using CRISPR–Cas9 deletion of exons 3–8. **e**, *Slc45a4* mRNA is absent in *Slc45a4*-KO mice. **f**, Metabolomic analysis of polyamine (Put, Spd, Spm) levels. Compared with the WT, Spd is reduced in KO spinal cords (WT, *n* = 3; KO, *n* = 4; *t*-test, \**P* = 0.019), but elevated in KO serum (WT, *n* = 8; KO *n* = 7; Mann–Whitney *U*-test, \**P* = 0.014), and Put levels are elevated in KO DRGs

(WT, *n* = 8; KO, *n* = 6; *t*-test, \**P* = 0.014). **g**, Sensory neuron subpopulation markers in WT and KO mice. **h**, Quantification of each subpopulation marker between WT and KO mice. WT: *n* = 4 mice, 1,073 (CGRP), 747 (IB4), 742 (NF200) and 167 (TH) cells; KO: *n* = 3 mice, 761 (GCRP), 524 (IB4), 402 (NF200) and 151 (TH) cells. Statistical analysis was performed using two-way analysis of variance (ANOVA) with post hoc Holm–Šidák test; *P* > 0.05. Scale bars 100 µm. **i**, Intraepidermal innervation in WT and KO mice. Scale bars, 50 µm. **j**, Quantification of total (PGP9.5+) and CGRP+ fibre density in glabrous (top) and hairy skin (bottom). *n* = 5 (WT) and *n* = 4 (KO) mice; 3–6 sections per mouse. Statistical analysis was performed using *t*-tests; *P* > 0.05. **k**, Hair follicle innervation, in particular for CGRP+ Cir-HTMRs, appears normal in *Slc45a4*-KO mice. Scale bars 20 µm. All data are mean ± s.e.m.

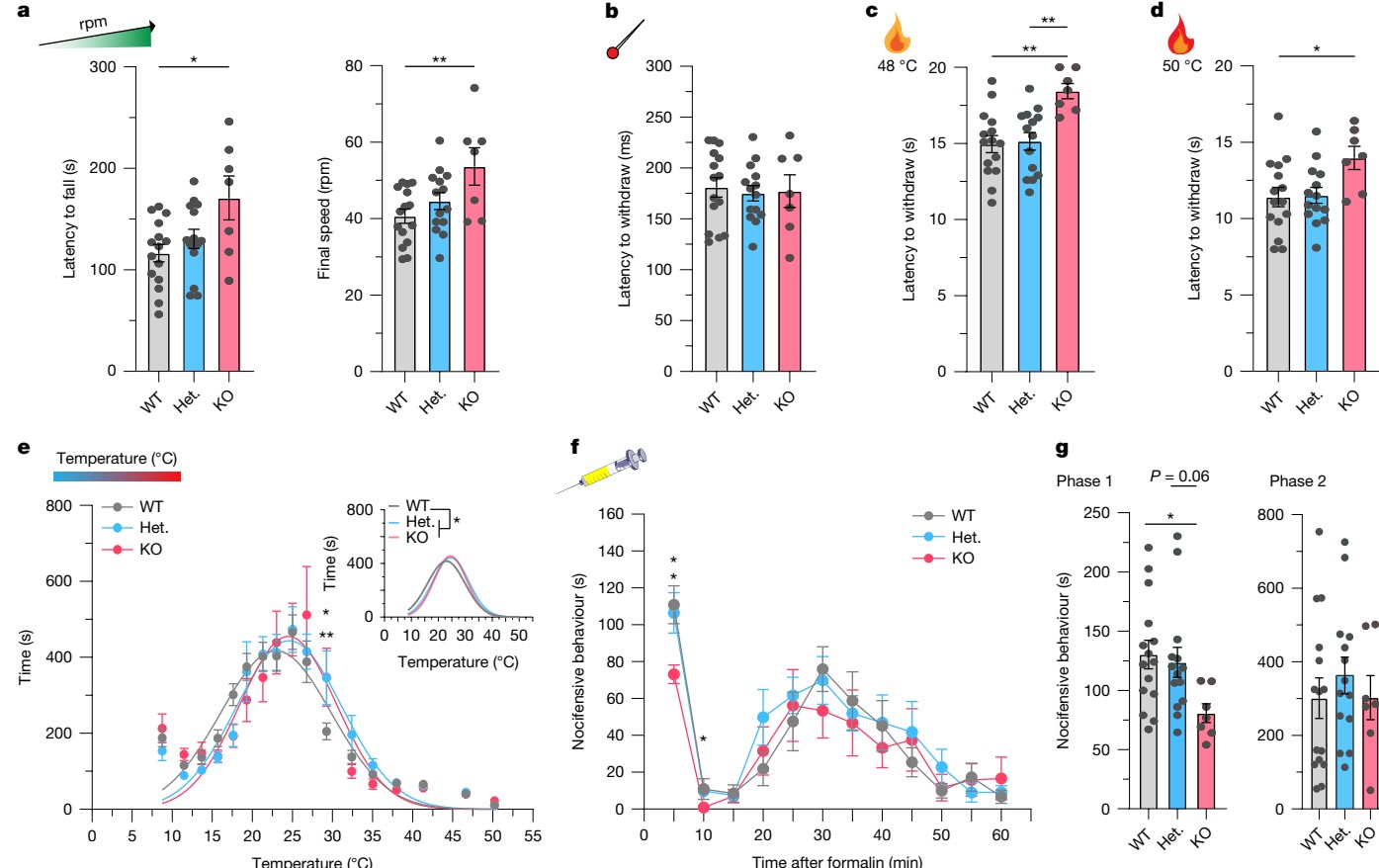

**Fig. 4 | SLC45A4 is important for motor endurance, heat sensitivity and tonic pain. a**, *Slc45a4*-KO mice show a longer latency to fall and a higher final speed when challenged with a rotarod that gradually increases in speed. *$P = 0.013$, **$P = 0.0085$. **b**, *Slc45a4* heterozygous and KO mice show a normal latency to withdraw from a noxious pin prick ($P > 0.05$). **c**, *Slc45a4*-KO mice take longer to withdraw from a 48 °C hot plate compared with WT and heterozygous mice (**$P = 0.002$ and $P = 0.0039$, respectively; 20 s cut-off). **d**, *Slc45a4*-KO mice have an increased latency to withdraw from a 50 °C hot plate compared with WT mice (*$P = 0.039$; 20 s cut-off). **e**, Heterozygous and KO mice had a rightward shift in their thermal gradient profile, towards warmer temperatures. Statistical analysis was performed using two-way ANOVA with post hoc Holm–Šidák test; WT versus heterozygous, **$P = 0.009$; WT versus KO, *$P = 0.027$. Datapoints and nonlinear fit Gaussian curves (inset) are shown. A single curve cannot explain

all datasets (extra sum-of-squares *F*-test, *$P = 0.024$). **f**, *Slc45a4*-KO mice display less nocifensive behaviours in response to a formalin injection compared with WT and heterozygous mice. Statistical analysis was performed using repeated-measures (RM) two-way ANOVA with post hoc Holm–Šidák test; 5 min: WT versus KO, *$P = 0.011$, heterozygous versus KO, *$P = 0.027$; 10 min: heterozygous versus KO, *$P = 0.018$. **g**, *Slc45a4*-KO mice show less nocifensive behaviours in phase 1 (0–15 min) compared with WT mice. Statistical analysis was performed using a Kruskal–Wallis test; *$P = 0.02$. phase 2 (15–60 min) is normal in all groups. For **a**–**g**, $n = 15$ (WT), $n = 14$ (heterozygous) and $n = 7$ (KO) mice. For **a**–**d** and **g**, statistical analysis was performed using one-way ANOVA with Tukey post hoc test. All data are mean ± s.e.m. The diagram of the syringe in panel **f** was created using Servier Medical Art (https://smart.servier.com/), licensed under a CC BY 4.0 license.

with in the WT mice (Extended Data Fig. 10a). To understand the effect of reduced spinal GABA, we performed patch-clamp analysis of dorsal horn lamina II neurons from *Slc45a4*-KO mice and found that they exhibited normal physiological properties (Extended Data Fig. 10b). We next assessed the lamina II inhibitory tone and found no change in amplitude or frequency of miniature inhibitory post-synaptic currents (mIPSCs) in *Slc45a4*-KO mice (Extended Data Fig. 10c). On closer examination, we found that the GABA levels in the dorsal horn were normal, and were significantly decreased only in the ventral horn of *Slc45a4*-KO mice compared with WT mice (Extended Data Fig. 10d). Collectively, this illustrates that loss of SLC45A4 can alter GABA levels, but that this is specific to the ventral horn, consistent with our anatomical assessment that the predominant locus of spinal SLC45A4 expression is the ventral horn. Moreover, a decrease in GABA in the ventral horn may lead to a reduction in motor neuron inhibitory tone that could explain our behavioural findings of increased motor endurance.

To directly explore a peripheral mechanism of pain modulation, we undertook patch-clamp analysis of dissociated small-sized sensory neurons, and focused on two major nociceptor populations: those that

bind to IB4 (predominantly non-peptidergic nociceptors) and those that do not bind to IB4 (which are predominantly but not exclusively peptidergic nociceptors; Methods) (Fig. 5a). We observed normal passive membrane properties and comparable action potential thresholds (Supplementary Table 6) in *Slc45a4*-KO neurons compared with in the WT. Notably, while IB4+ nociceptors were normal (Fig. 5b–d), we found a selective reduction in the suprathreshold excitability of IB4− nociceptors from *Slc45a4*-KO mice in response to static (step) and dynamic (ramp) current injections, compared with neurons from WT mice (Fig. 5b,e,f). Assessing soma excitability using artificial current injections is important, but we also wanted to understand how sensory information is coded at the level of the sensory terminal and nerve, so we conducted the ex vivo glabrous skin–nerve preparation (Fig. 5g). This enabled us to extensively characterize the stimulus–response functions of six major sensory afferent subclasses, including nociceptors. We found that, from WT and *Slc45a4*-KO mice, C-mechano-nociceptors (C-Ms) (Fig. 5h,i), Aδ-mechano-nociceptors, Aδ-D-hairs, Aβ-RA-LTMRs and Aβ-SA-LTMRs (Extended Data Fig. 11a–c) had comparable conduction velocities, mechanical or vibration thresholds, and stimulus–response

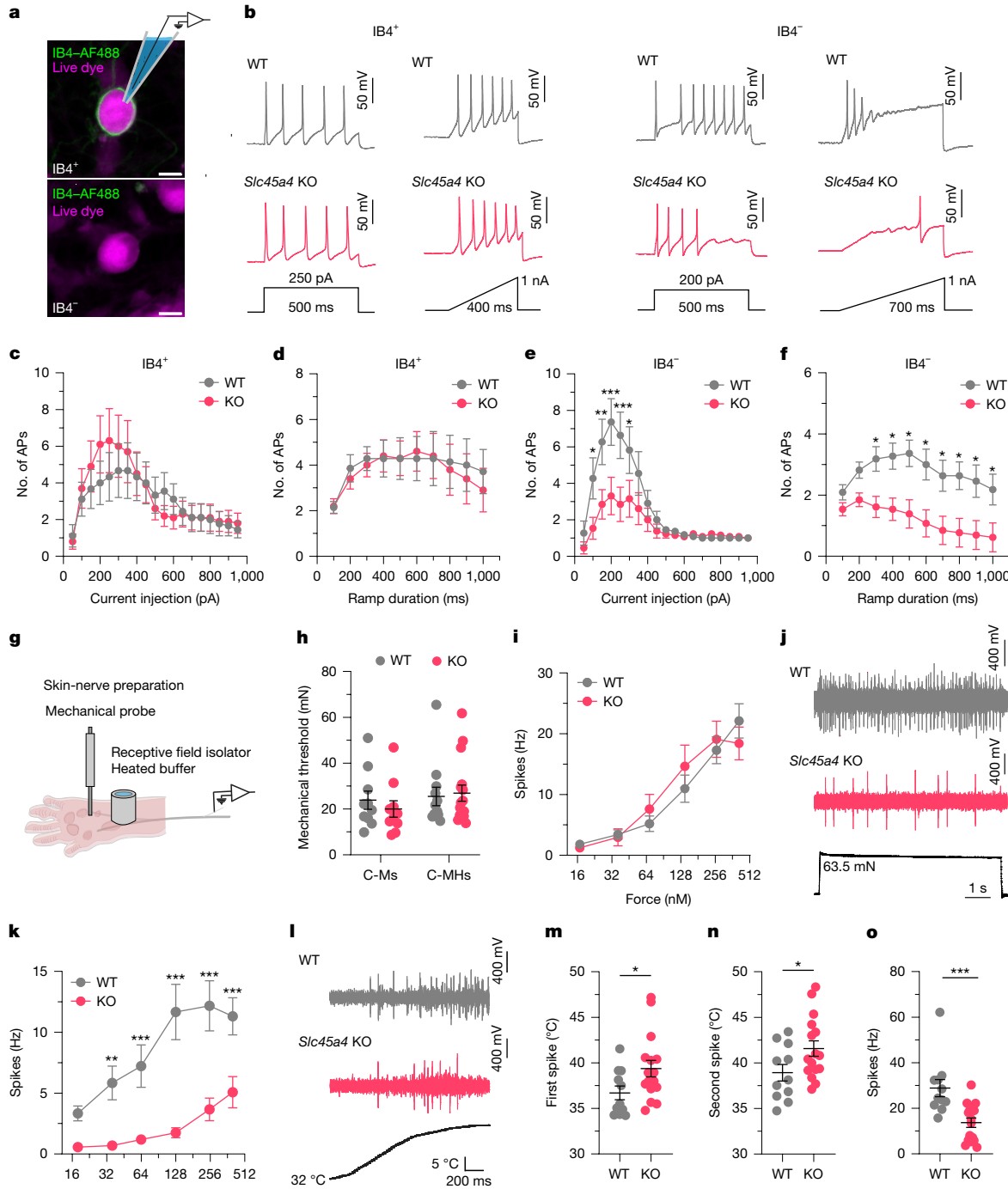

**Fig. 5 | SLC45A4 regulates the excitability of C-MHs. a**, IB4–AlexaFluor 488 (AF488) live binding and IB4⁻ nociceptors were characterized using patch-clamp analysis. Scale bar, 10 μm. **b**, Excitability traces of IB4⁺ and IB4⁻ neurons in response to step-current or ramp-current injections. **c**,**d**, KO IB4⁺ neurons show normal firing patterns to step-current (**c**) and ramp-current (**d**) injections. *n* = 9 (WT) and *n* = 10 (KO) cells. *P* > 0.05. AP, action potential. **e**,**f**, KO IB4⁻ neurons fire less in response to step-current injections (**e**; *n* = 11 (WT) and *n* = 13 (KO) cells; from 100 to 300 pA, *\*P* = 0.039, *\*\*P* = 0.0027, *\*\*\*P* = 0.0002, *\*\*\*P* = 0.0006, *\*P* = 0.047) and ramp-current injections (**f**) compared with WT neurons (from 300–1,000 ms, *\*P* = 0.036, *P* = 0.021, *P* = 0.011, *P* = 0.013, *P* = 0.021, *P* = 0.015, *P* = 0.021 and *P* = 0.036, respectively). **g**, Hindpaw glabrous skin-nerve preparation. **h**, Normal mechanical thresholds of C-Ms and C-MHs from WT and KO mice. C-Ms: *n* = 10 (WT) and *n* = 10 (KO) units; C-MHs: *n* = 12 (WT) and *n* = 16 (KO) units. Statistical analysis was performed using Mann–Whitney

*U*-tests; *P* > 0.05. **i**, WT and KO C-Ms respond similarly to suprathreshold mechanical stimuli (*P* > 0.05). **j**, C-MH fibres responding to a suprathreshold mechanical stimulus. **k**, KO C-MHs respond less to suprathreshold mechanical stimuli (from 35 to 402 mN, *\*\*P* = 0.002, *\*\*\*P* = 0.0004, *\*\*\*P* < 0.0001, *\*\*\*P* < 0.0001, *\*\*\*P* = 0.0002, respectively). **l**, C-MH fibre response to heat stimulus (32–50 °C). **m**,**n**, Temperature thresholds for the first (**m**) and second (**n**) spike are increased in KO C-MHs compared with in the WT. *n* = 11 (WT) and *n* = 16 (KO) units. Statistical analysis was performed using *t*-tests; *\*P* = 0.04 (first spike) and *\*P* = 0.04 (second spike). **o**, The spike frequency during a 32–50 °C stimulus is reduced in KO C-MHs versus in WT C-MHs. Statistical analysis was performed using Mann–Whitney *U*-tests; *\*\*\*P* = 0.0002. For **c**–**f**, **i** and **k**, statistical analysis was performed using RM two-way ANOVA with post hoc Holm–Šidák test. All data are mean ± s.e.m.

functions to mechanical stimuli. Notably, we observed selective and marked hypoexcitability of C-mechano-heat-nociceptors (C-MHs; also known as C-polymodal nociceptors) in response to both suprathreshold mechanical stimuli (Fig. 5j,k) and heat stimuli (Fig. 5l–o) in *Slc45a4*-KO mice compared with in the WT controls. Taken together, we show that SLC45A4 is important for regulating the somal and nerve excitability of a subclass of nociceptors that we can broadly define as polymodal and likely peptidergic nociceptors.

In summary, our results provide compelling evidence that *SLC45A4* encodes a neuronal membrane polyamine transporter that shows genetic association with human pain. SLC45A4 is expressed in sensory neurons, and ablation of its function results in altered polyamine homeostasis, thermal coding and pain perception in mice. Future studies will be needed to assess the effect of specific coding variants on both the regulation of polyamine transport and nociceptive function. These behavioural changes in *Slc45a4*-KO mice are accompanied by a marked reduction in the excitability of C-MHs, which are known to encode thermal pain and the response to many algogens. Mechanical pain responses remained intact after SLC45A4 loss, probably due to the preserved function of other mechanoreceptor populations. Polyamines are known to functionally modulate a number of ion channels that may be candidates for affecting the excitability of nociceptors, including the transient receptor potential family (TRPV1, TRPV3 and TRPV4)[14,15], inward-rectifier K[+] channels[43] and acid-sensitive ion channels (ASICs)[44]. Our observation that SLC45A4 specifically affects the C-polymodal nociceptor population may reflect the expression pattern of polyamine-sensitive channels in these neurons. Our study has identified *SLC45A4* as the neuronal polyamine transporter and suggests that SLC45A4 may be a promising molecular target to modulate pain perception in humans.

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

# Methods

## Study Population

The UKB is a prospective cohort study that has collected extensive phenotypic and genetic data from approximately 500,000 participants across the United Kingdom[21,45]. All of the participants provided written informed consent.

## Phenotype description and assessment

Phenotypes were defined using the 'Experience of Pain' UKB self-assessment questionnaire data under project ID 49572. This is part of the online UKB follow-up and was sent in May 2019 to all UKB participants with an active email address who had consented to further electronic contact ($n = 335,587$) (further details are provided in the Supplementary Methods). Definitions and characteristics of the phenotypic end points were described in detail previously[22]. Participants who have withdrawn (153 as of 15 September 2023) were excluded from the analysis. Chronic pain was defined by considering a screening question asking whether participants are "troubled by pain or discomfort present for more than 3 months" (item f120019). The question "area most bothered by pain in the last three months" (item f120037) was used to define the location of most bothersome pain and the intensity of the most bothersome pain was defined from items 120023–120035 asking for the "rating of pain" in the most bothersome pain location. People with no chronic pain had their intensity values imputed to 0. People who self-reported fibromyalgia (f120009), chronic fatigue syndrome/myalgic-encephalomyelitis (f120010) or chronic pain all over the body (f120021) were excluded from the analysis (11,951 with chronic pain and 821 with no chronic pain). We had valid most bothersome pain ratings for 139,167 (71,904 and 67,263 meeting criteria for no chronic pain and chronic pain, respectively).

## Genotyping, imputation and quality control

In this study, we used genotype datasets from the UKB to investigate genetic factors related to chronic pain. We used two specific releases: version 2 for directly genotyped variants and version 3 for imputed genotypes. Initially, approximately 50,000 UKB participants were genotyped using the Applied Biosystems UK BiLEVE Axiom Array by Affymetrix. Subsequently, around 440,000 participants were genotyped using the Applied Biosystems UKB Axiom Array.

Quality-control and imputation procedures were performed as detailed previously[45] and resulted in the released dataset of 488,377 samples and 805,426 directly typed markers from both arrays. This dataset was subjected to further quality control before phasing and imputation were performed using a combined Haplotype Reference Consortium (HRC) and UK10K reference panel. The imputed dataset has over 90 million autosomal SNVs, short indels and large structural variants for 487,442 individuals.

To assess genetic ancestry, we used the KING software (v.2.3.2)[46], using the 1000 Genomes Project as a reference panel. The directly genotyped dataset was used with additional quality control filters using PLINK (v.1.90b6.21, https://www.cog-genomics.org/plink/1.9/)[47] that included: autosomes, minor allele frequency ≥ 5%, not present in high linkage disequilibrium (LD) regions and LD pruning using a $R^2$ threshold of 0.2 with a window size of 50 markers and a step size of 5 markers. This analysis enabled us to identify five distinct subpopulations within the UKB: African (9,059 samples), admixed American (605 samples), East Asian (2,572 samples), European (464,586 samples) and South Asian (9,604 samples). Moreover, there were 1,951 samples for which the ancestry could not be determined and the ancestries were therefore categorized as missing.

Genotyping data, encompassing both directly genotyped and imputed variants, and excluding individuals who had withdrawn their consent from the UKB study were available for 487,071 samples. Additional quality-control measures were applied that further excluded 367 samples where the reported sex did not match the inferred sex

from their genetic data, 651 samples with suspected sex chromosome aneuploidy and 188 samples with more than ten putative third-degree relatives. In total, 1,024 samples were excluded, with some samples falling into multiple exclusion categories.

Samples of European ancestry were selected, resulting in 462,402 samples available for analyses. The imputed dataset (chromosomes 1–22) was restricted to common and low-frequency variants (minor allele frequency ≥ 1%) that were imputed with high confidence (imputation accuracy info score ≥ 0.8) leaving 9,572,556 variants in the dataset.

## Association analyses and candidate SNV identification

We conducted association analyses using REGENIE (v.3.4.1)[48]. REGENIE uses a two-step whole-genome regression method that effectively accounts for population stratification and sample relatedness. For the continuous outcome, we applied rank-based inverse normal transformation to improve the distribution and meet the assumptions of the regression model. The model included the following covariates: age at the time of completing the questionnaire, sex, genotyping array and the top ten principal components provided by the UKB. REGENIE step 1 was run on a set of the directly genotyped variants, filtered using PLINK2 (v.2.00a5; https://www.cog-genomics.org/plink/2.0/)[47] that included sample genotyping rate ≥ 90%, autosomes, minor allele frequency ≥ 1%, Hardy–Weinberg equilibrium test not exceeding $P = 1 \times 10^{-15}$, variant genotyping rate ≥ 99%, not present in high LD regions and LD pruning using a $R^2$ threshold of 0.9 with a window size of 1,000 markers and a step size of 100 markers. REGENIE step 2 was run on the imputed dataset.

To identify risk loci and their lead variants, we performed LD clumping using the Functional Mapping and Annotation of Genome-Wide Association Studies (FUMA)[49]. We set a range of 250 kb and an $r^2$ threshold of >0.6 to define independent significant SNVs. For lead SNVs, we used an $r^2$ threshold of >0.1. The analysis was based on the respective ancestry from the 1000 Genomes Phase 3 EUR reference panel[50]. After clumping, we combined genomic risk loci within 1 Mb of each other into a single locus. Moreover, we leveraged resources from Open Targets Genetics, which integrate data from human GWAS and functional genomics, including gene expression, protein levels, and chromatin interactions across various cell types and tissues[51]. This comprehensive approach enabled us to confirm the connections between GWAS-associated loci, variants and their probable causal genes.

## pheWAS

To determine potential associations between the lead SNV associations and their corresponding genes from our study and additional traits, we conducted a pheWAS. This analysis involved generating pheWAS plots from a comprehensive dataset comprising 4,756 GWAS summary statistics, acquired from the GWAS ATLAS. We incorporated all relevant GWAS and associated genes into our selection criteria. For the pheWAS SNV plots, SNVs were deemed to be significant with $P$ values of less than 0.05 and applied the Bonferroni correction method to adjust for multiple comparisons.

## Gene-based tests, pathway exploration and enrichment analyses

In our study, we used the FUMA software for gene-based tests, pathway exploration and enrichment analyses. FUMA leverages GWAS summary statistics to prioritize genes, assess gene expression and enrich pathway processes. To address the issue of multiple testing, FUMA applied the Bonferroni correction with a threshold of $P_{bon} < 0.05$. Moreover, FUMA incorporates multimarker analysis of genomic annotation (MAGMA) for both gene-based and gene-set analysis.

## Ethics

Data for this study were obtained from the UKB for project "Risk factors for chronic pain", application ID: 49572. UKB has approval from

the North West Multicentre Research Ethics Committee (MREC) as a Research Tissue Bank (RTB) approval, REC reference: 21/NW/0157, IRAS project ID: 299116. This approval means that researchers do not require separate ethical clearance and can operate under the RTB approval.

## Metabolome correlation analysis

Correlation analysis to identify potential substrates was performed as described previously[28]. In brief, RNA-seq raw counts data for cell lines in the CCLE were downloaded and processed using median of ratios normalization. Metabolomics data were downloaded from the CCLE website (https://sites.broadinstitute.org/ccle/datasets). Cell lines were matched, and Spearman's rank correlation coefficients were calculated between each SLC protein and each metabolite. The significance of the correlation coefficients was adjusted using the Benjamani−Hochberg multiple-testing method and plotted as a volcano plot with the value of the correlation coefficient on the $x$ axis and the $\log_{10}$ of the significance on the $y$ axis.

## Cloning, expression and purification of *Hs*SLC45A4

The gene encoding full-length human SLC45A4 (UniProt: Q5BKX6) was inserted into pDDGFP-Leu2D[52], containing a C-terminal tobacco etch virus (TEV) protease cleavable eGFP−His$_8$ tag, for expression in *Saccharomyces cerevisiae* strain BJ5460 (ATCC-208285). Transformed yeast were cultivated in synthetic complete medium without leucine (−Leu), supplemented with 2% (w/v) glucose, at 30 °C to an optical density at 600 nm (OD$_{600}$) of 5.0–6.0 and diluted ninefold into −Leu with 2% (v/v) lactate pH 5.1. Once the OD$_{600}$ reached 1.8–2.2, expression was induced by addition of 1.5% galactose and expression was maintained for 20–24 h. Cells were collected, lysed through high pressure cell disruption (40 kpsi) and membranes were isolated through ultracentrifugation at 200,000$g$ for 90 min. Membranes were washed in wash buffer, 20 mM HEPES pH 7.4, 1 M KAc, and isolated again at 200,000$g$ for 90 min before being resuspended in 1× PBS and stored at −80 °C.

For expression in tissue culture, SLC45A4 was inserted into the pLexM vector[53] with a C-terminal Avi tag (GLNDIFEAQKIEWHE), followed by TEV-cleavable eGFP−His$_6$ fusions. HEK293F cells (Thermo Fisher Scientific), which tested negative for mycoplasma, were cultured in FreeStyle 293 Expression Medium (Thermo Fisher Scientific) at 37 °C under 8% CO$_2$ and 24 h before transfection, were passaged to 0.7 × 10$^6$ cells per ml to give a density of 1.3–1.4 × 10$^6$ cells per ml after transfection. Transfection was carried out with 1.1 mg plasmid and 2.2 mg linear polyethyleneimine (PEI) MAX (Mw 40,000; Polysciences) per l culture. After transfection, sodium butyrate was added to 10 mM to increase protein expression. Cells were collected 40 h after transfection. Membranes were prepared by lysing the cells by sonication and unbroken cells and cell debris were pelleted at 10,000$g$ for 10 min at 4 °C and membranes were collected through centrifugation at 200,000$g$ for 1 h and washed once with 20 mM HEPES pH 7.5, 20 mM KCl. After washing, the membranes were resuspended in PBS and snap-frozen for storage at −80 °C until required. For purification, thawed membranes were solubilized in buffer containing 1× PBS pH 7.4, 150 mM NaCl, 10% (v/v) glycerol and 2% (w/v) *n*-dodecyl-β-D-maltopyranoside (DDM, Anatrace) with 0.4% (w/v) cholesteryl hemisuccinate (CHS) for nanodisc reconstitution or 1% lauryl maltoside neopentyl glycol with (LMNG, Anatrace) 0.1% (w/v) CHS for 90 min at 4 °C under gentle agitation using a magnetic stir plate. Insoluble material was removed through centrifugation for 1 h at 200,000$g$. SLC45A4 was purified to homogeneity using standard immobilized metal-ion affinity chromatography protocols in either DDM:CHS (5:1 ratio) or LMNG:CHS (10:1 ratio). In brief, 4 ml of nickel NTA resin (Thermo Fisher Scientific) was added with 25 mM imidazole (Sigma-Aldrich) for 3 h at 4 °C under gentle agitation using a magnetic stir plate. The resin was loaded onto a gravity flow column (Bio-Rad) and washed with ten column volumes (CVs) of buffer containing either 0.15% DDM:CHS (5:1 ratio) or 0.15% LMNG:CHS (5:1 ratio) and 25 mM imidazole, followed by 20 CVs of buffer with 30 mM imidazole and 300 mM NaCl. The protein was eluted in four CVs of buffer containing 250 mM imidazole and dialysed overnight with TEV protease (1:0.5 M ratio) against 20 mM Tris pH 7.5, 150 mM NaCl with 0.03% DDM:CHS (5:1 ratio) or 0.003% LMNG:CHS (10:1 ratio) at 4 °C under gentle agitation using a magnetic stir plate. After TEV cleavage, the protease and cleaved His-tagged GFP were removed through nickel affinity chromatography on a 1 ml HisTrap column (Sigma-Aldrich). Unbound material was then concentrated to 500 µl using a 50 kDa molecular-weight cut-off spin concentrator (Sartorius) at 4 °C and subjected to size-exclusion chromatography (Superdex 200) in the same buffer as above.

## Cell-based ¹⁴C-SPD transport assays

The transport activity of overexpressed SLC45A4 was assayed in Neuro-2A cells (ATCC CCL-131), which tested negative for mycoplasma, using ¹⁴C-radiolabelled SPD (American Radiolabeled Chemical, ARC3138, 100 mCi mmol$^{-1}$). Cells were maintained in Gibco DMEM (high glucose, GlutaMAX Supplement, pyruvate, 31966021) at 37 °C under 5% CO$_2$. For transport assays, 1.0–1.7 × 10$^5$ cells were seeded per well in 12-well plates and transfected 48–52 h later (once confluency had reached about 60–80%) using FuGENE HD (Promega, E2311) transfection reagent, with 1 µg plasmid (WT *SLC45A4* or mutants in pLexM, with a C-terminal Flag tag) and 2.5 µl FuGENE per well. Fresh medium was placed onto the cells 12–15 h thereafter and assays were carried out 40–46 h after transfection.

Before measuring transport activity, cells were washed twice in assay buffer (25 mM HEPES pH 7.5, 135 mM NaCl, 5 mM KCl, 1.2 mM MgCl$_2$, 28 mM glucose) and incubated for 3 min at 37 °C after the second wash. Immediately after the incubation, 240 µl of the assay buffer supplemented with 1 µM ¹⁴C-SPD and 9 µM cold SPD was pipetted onto the cells and incubated for 1.5, 5 and 10 min (time-course assays) or 8 min (single-timepoint assays for mutants) at 37 °C. Uptake was then quenched by washing cells twice in 500 µl ice-cold assay buffer and the cells were lysed in 20 mM Tris pH 7.5, 0.2% Triton X-100. The amount of transported ¹⁴C-SPD was measured by scintillation counting in Ultima Gold scintillation liquid (PerkinElmer), and the amount of transported SPD was calculated using a standard curve for the substrate using the specific activity of the SPD. For competition assays and IC$_{50}$ measurements, only 1 µM of ¹⁴C-SPD was used in the substrate mixture along with the desired concentration of the competing compound. Experiments were performed at least six times to generate an overall mean and s.d. Owing to variability and high background in the assay, originating from endogenous SLC45A4 and possibly to some extent endocytosis/binding to plasma membrane proteins, each experiment was performed with both WT (cells that were transfected with a plasmid containing WT *SLC45A4*) and cells transfected with an empty plasmid. If the WT uptake did not exceed 2.5-fold that of the empty plasmid control, the results from that plate were discarded.

Expression of WT *SLC45A4* and mutants were assessed using western blotting on membrane fractions with an anti-Flag antibody (Merck, F1804) at 5,000× dilution, using an anti-β-actin antibody as a loading control at 10,000× dilution (Merck, A2228) (Extended Data Fig. 1f). Plasma membrane localization was measured using immunofluorescence (Extended Data Fig. 1h). Cells were seeded on glass coverslips, transfected as described above with Flag-tagged SLC45A4 and 36 h after transfection, cells were washed in PBS pH 7.4 and fixed in 4% PFA for 7 min. After quenching in 50 mM ammonium chloride and further washing, the cells were permeabilized in 100 µM digitonin and the coverslips blocked in 1% bovine serum albumin (BSA) for 30 min. Cells were stained with mouse anti-Flag (1:200 dilution) and rabbit anti-Na$^+$/K$^+$ ATPase (1:50 dilution) primary antibodies for 1 h, washed and further stained with goat anti-mouse IgG AF488 (1:200) and anti-rabbit IgG AF647 (1:200) secondary antibody−fluorophore conjugates for imaging. Imaging was performed on the LSM-980 confocal microscope (Zeiss) and images were processed in ZenBlue (v.3.9, Zeiss) and ImageJ[54].

## Thermal stability measurements

For assessment of thermal stability in the absence or presence of compounds, nano-differential scanning fluorimetry measurements were carried out on the Prometheus NT.28 instrument (NanoTemper Technologies). Purified SLC45A4 was diluted to a final concentration of 0.2–0.4 mg ml$^{-1}$ in buffer (20 mM Tris pH 7.5/20 mM MES pH 7.5, 150 mM NaCl, DDM:CHS (0.03:0.003%) with 2.5–50 mM metabolite. Unfolding was monitored as the ratio of Trp fluorescence emission at 330 nm and 350 nm between 20–90 °C using a ramp rate of 1 °C min$^{-1}$. The apparent $T_m$ was determined as the maximum of the first derivative of the emission ratio.

## Nanodisc reconstitution

SLC45A4, purified from HEK293F in DDM:CHS, was reconstituted into MSP1D1 lipid nanodiscs with EBC lipid (85% (mol/mol) *E. coli* polar lipid extract, 10% bovine brain polar lipid extract, 5% cholesterol) using the BioBead method[55]. Then, 100 μg SLC45A4 was mixed with MSP1D1 and EBC lipid solubilized in 0.5 M sodium cholate, at a molar ratio of 1:5:75, respectively, the mixture incubated on ice for 1 h and excess detergent removed through stepwise addition of BioBeads and overnight incubation under gentle agitation. Insoluble material was removed through ultracentrifugation and reconstituted SLC45A4 separated from empty nanodiscs and excess lipid on a Superdex200 Increase 10/300 GL column in 20 mM HEPES pH 7.5, 150 mM NaCl and concentrated to 2 mg ml$^{-1}$ for cryo-EM analysis.

## Cryo-EM sample preparation and data acquisition

For the LMNG:CHS sample, purified SLC45A4 was subjected to a further round of SEC polishing on the Superdex 200 Increase 10/300 GL column in 20 mM Tris pH 7.5, 150 mM NaCl, 0.001% LMNG:CHS (10:1), to separate from empty detergent micelles, and the unconcentrated peak fraction (0.31 mg ml$^{-1}$) used for grid preparation. The sample was adsorbed to glow-discharged holey carbon-coated grids (Quantifoil 300 mesh, Au R1.2/1.3) for 10 s. The grids were then blotted for 2 s at 100% humidity at 10 °C and frozen in liquid ethane using the Vitrobot Mark IV (Thermo Fisher Scientific). Data were collected in counting mode in electron event representation (EER) format on the CFEG-equipped Titan Krios G4 (Thermo Fisher Scientific) system operating at 300 kV with a Selectris X imaging filter (Thermo Fisher Scientific) with slit width of 10 eV at ×165,000 magnification on a Falcon 4i direct detection camera (Thermo Fisher Scientific) corresponding to a calibrated pixel size of 0.732 Å. Videos were collected at a total dose of 57.6 e$^-$ Å$^{-2}$ fractionated to about 1 e$^-$ Å$^2$ per fraction. SLC45A4 reconstituted into EBC:MSP1D1 nanodiscs, as described above, was concentrated to 2 mg ml$^{-1}$ and directly used for preparation of grids. Then, 3 μl of sample was applied onto a glow-discharged holey carbon grid (Quantifoil Cu R1.2/1.3 300 mesh), blotted for 5 s at 100% humidity and 4 °C and plunge-frozen in liquid ethane using a Vitrobot Mark IV. Data were collected in counted super-resolution bin2 mode on the Titan Krios G3 (FEI) system with a K3 camera (Gatan) and Bio-Quantum imaging filter at 300 kV, with a pixel size of 0.832 Å. A total of 20,184 micrographs was collected using a dose of 15.89 e$^-$ Å$^{-2}$ s$^{-1}$ and an exposure time of 2.5 s, giving a total dose of 39.71 e$^-$ Å$^{-2}$.

## Cryo-EM data processing

Patched (20 × 20) motion correction, CTF parameter estimation, particle picking, extraction and initial 2D classification were performed in SIMPLE (v.3.0)[56]. All further processing was performed in cryoSPARC (v.3.3.1)[57] and RELION (v.3.1)[58], using the csparc2star.py script within UCSF pyem[59] to convert between formats. The global resolution was estimated from gold-standard Fourier shell correlations (FSCs) using the 0.143 criterion and local resolution estimation was calculated within cryoSPARC.

The cryo-EM processing workflow for SLC45A4 in LMNG:CHS is outlined in Extended Data Fig. 2. In brief, particles were subjected to two rounds of reference-free 2D classification ($k$ = 300 each) using a 140 Å soft circular mask within cryoSPARC. Four volumes were generated from a 479,080 particle subset of the 2D-cleaned particles after multiclass ab initio reconstruction using a maximum resolution cut-off of 6 Å. Particles from the most populated and structured class were selected, and another multiclass ab initio reconstruction ($k$ = 4) was performed. Output volumes were lowpass-filtered to 7 Å and used as references for a four-class heterogeneous refinement against the full 2D-cleaned particle set (2,815,141 particles). Particles (964,648) from the most populated and structured class were selected and non-uniform refined against their corresponding volume lowpass-filtered to 15 Å, generating a 3.3 Å map. A multiclass ab initio reconstruction ($k$ = 4) using a maximum-resolution cut-off of 6 Å was performed on these particles, generating four volumes that were lowpass-filtered to 7 Å and used as references for heterogeneous refinement against the same particle set. Particles (700,436) belonging to the two most populated and structured volumes (which were in opposite hands) were combined and subjected to non-uniform refinement against one of the corresponding volumes, lowpass-filtered to 15 Å, generating a 3.3 Å map with slightly improved density over the previously refined volume. Bayesian polishing followed by non-uniform refinement (15 Å lowpass-filtered reference) further improved the map quality to 3.0 Å. Per-particle defocus refinement followed by non-uniform refinement (15 Å lowpass-filtered reference) resulted in a 2.8 Å map that was used for model building.

For the SLC45A4 nanodisc structure (Extended Data Fig. 3), the 20,196,090 extracted particles (box size of 240 px) were subjected to three rounds of conservative 2D classification to give 1,473,487 particles, which were further classified into four classes in ab initio reconstruction. Three rounds of less conservative 2D classification of the initial particle set gave 5,812,766 particles, which were classified by two rounds of heterogeneous refinement, using the best ab initio map from the earlier reconstruction. After further sorting particles using ab initio reconstruction, Bayesian polishing was performed and particles were re-extracted with a box size of 320 px. After 2D classification, two further rounds of ab initio reconstruction, non-uniform and local refinement gave a final map of 3.25 Å (where FSC = 0.143) from 227,752 particles.

Model building was performed in Coot (v.0.9.8.1 EL)[60] and ISOLDE[61], refinement in PHENIX (v.1.20.1-4487) real-space refinement[62] and validation in MolProbity[63]. Images were generated using PyMol[64] and ChimeraX[65].

## Animals

Animals were group-housed in temperature-controlled (21.5 ± 0.5 °C) and humidity-controlled (55 ± 10%), specific-pathogen-free facilities in individually ventilated cages. Mice were housed under 12 h–12 h light–dark cycle with food and water provided ad libitum. All experiments were carried out using male and female C57BL/6J mice. Behavioural studies were carried out on age and sex matched littermates. All transgenic mice were backcrossed onto the C57BL/6J background. All procedures complied with the UK Animals (Scientific Procedures) Act (1986) and were performed under a UK Home Office Project Licence in accordance with University of Oxford Policy on the Use of Animals in Scientific Research. This study conforms to the NC3Rs policy on reduction, refinement and replacement of animal research, and to the ARRIVE guidelines[66].

## Generation of the *Slc45a4*-KO mouse

Design and development of the *Slc45a4*-KO mouse was carried out in partnership with Taconic Biosciences and Cyagen Biosciences. In brief, CRISPR–Cas9 technology was used to delete exons 3–8 of the mouse *Slc45a4* gene on chromosome 15. This 13.3 kb deletion accounts for 89.64% of the *Slc45a4* coding region. Guide RNAs were designed to target only the *Slc45a4* locus and injected together with Cas9 into

mouse zygotes. Chimeric $F_0$ offspring mice were bred with WT mice to generate heterozygous founders $F_1$ which were screened for successful *Slc45a4* editing. The targeted region of *Slc45a4* was sequenced to confirm deletion of exons 3–8. Mice were genotyped using primers in Supplementary Table 7 using a standard Taq polymerase-based protocol. WT band, 652 bp; KO band, 468 bp.

## Mouse histology

**Tissue collection.** For histology studies, mice received an overdose of pentobarbital, then vascular perfused with saline and 4% paraformaldehyde (PFA). DRG, EDL muscle and skin tissue was collected and post-fixed in 4% PFA for 1–2 h, and spinal cord for 24 h. DRG, skin and spinal cord were cryoprotected in 30% sucrose and stored for at least 48 h before embedding in OCT. DRG tissue was cryosectioned at 12 μm, skin at 30 μm and spinal cord at 20 μm. Fixed EDL muscle was teased into small fibre bundles of about 1 mm diameter per bundle. For histology on neural cultures, live CellTracker dye CMTMR was added along with IB4-conjugated AF488, or cells were washed and fixed in 4% PFA for 15 min room temperature.

**IHC.** Standard immunohistochemistry (IHC) techniques were used. Antigen retrieval in a citric acid buffer was performed on skin sections before IHC (citrate buffer, sodium citrate dihydrate, EDTA, distilled $H_2O$, pH 6.1, 65 °C, 1 h). In brief, the sectioned or teased samples were washed in PBS and blocked in a blocking solution (5% normal goat serum and 0.3% Triton X-100, PBS) for at least 30 min at room temperature. Primary antibodies (Supplementary Table 8) were diluted in the same blocking solution and were left to incubate on the tissue samples overnight at room temperature or 4 °C. The samples were washed in a washing buffer solution (0.3% Triton X-100, PBS). Secondary antibodies (Supplementary Table 8) were diluted in the washing buffer solution and left to incubate on the tissue for 1–2 h at room temperature in dark conditions. The samples were washed thoroughly in the washing buffer, DRG and skin sections were mounted using Vectorshield (with or without DAPI) and muscle fibres were mounted onto microscope slides (Avantor, 631-0108) using hard-set confocal matrix (Micro Tech Lab). Images were taken on confocal microscopes (Zeiss LSM-710, LSM-780 or Olympus FV3000).

**ISH analysis.** In situ hybridization (ISH) was performed according to the user instructions for the RNAscope 2.5 RED Chromogenic assay kit (Advanced Cell Diagnostics, Bio-techne). In brief, fixed DRG tissue underwent pretreatment with $H_2O_2$, protease III and 100% ethanol. The tissue was then incubated at 2 h at 40 °C with an *Slc45a4* mRNA probe (522131). Probe amplification, washing steps and chromogenic development with fast red were carried out as described in the user kit. The tissue was then co-stained and imaged using the IHC protocol described above.

**Image analysis.** All image analysis was performed using ImageJ/Fuji (NIH). For IHC, at least, three sections per animal were used from three animals per group. Total cells and cells positive for each subpopulation marker were counted using the multipoint tool. For ISH, cells were defined and mRNA intensity calculated. The background intensity was calculated with tissue that underwent the ISH protocol with a standard negative control mRNA probe. A threshold for positive cells was defined as cells whose average intensity was more than mean + 3 s.d. of background intensity. Cell size classification was as follows: cell area < 490 μm² was classified as a small cell, cell area between 490–962 μm² was classified as a medium cell and a large cell had an area greater than 962 μm². For intraepidermal nerve fibre density analysis, nerve fibres crossing into the epidermis were counted live under the ocular lens of the Zeiss LSM-710 system. The distance of each section was calculated post hoc in Fuji and nerve fibres were quantified as fibres per mm. Spinal neurons were first segmented using Cellpose 2.0. Within

Cellpose 2.0, we used a human-in-the-loop pipeline to train a custom model by annotating cells on the Cellpose graphical user interface. Cells were determined to be positive for target probe mRNA if signal intensity was 3 s.d. above the negative control readings. Superficial lamina (I + II) were defined by IB4 staining to mark lamina II. A reference template (Atlas of the Mouse Spinal Cord) was used analyse all other laminae. For assessment of neuromuscular junction (NMJ) size, fragmentation, area and nerve terminal registration, optimized settings for each image were used. Between three and five z stacks were obtained per muscle sample. Ten NMJs were analysed from each animal using ImageJ software. The ImageJ macro (BetterAreaGreyValues) was used to measure NMJ area and fragmentation (Supplementary Methods). Registration of different fluorescence signals was analysed using the JACoP plugin for ImageJ. All comparative image analysis was conducted with the experimenter blinded to genotype. Cartoon illustrations were made in-house or provided by Servier Medical Art (https://smart.servier.com/), licensed under a CC BY 4.0 license.

## RT–qPCR

Tissue was collected from mice after overdose with pentobarbital and transcardiac perfusion with ice-cold saline. Tissues were rapidly dissected and flash-frozen in liquid nitrogen and stored at −80 °C. RNA was isolated from fresh-frozen mouse tissues using a combination of TriPure (Roche) and a RNeasy mini kit (Qiagen). In brief, tissue was homogenized in Tripure using a handheld homogenizer (Cole-Parmer), treated with chloroform and then column-purified and eluted in RNase-free water. Genomic DNA was removed with on-column DNase I digestion. cDNA synthesis was performed using the EvoScript Universal cDNA Master kit (Roche). Gene expression was quantified by detecting amplified material using the LightCycler SYBR Green Master Mix (Roche) on the LightCycler 480 II system (Roche). Three technical replicates were included for each sample. The results were normalized to three reference gene controls (mouse *Actb*, *Gapdh* and *Hprt*) using the $\Delta\Delta C_T$ method. A list of the primers is provided in Supplementary Table 7.

## DRG culture

DRG neurons were quickly and carefully dissected from the vertebral column[67] and enzymatically digested for 75–90 min at 37 °C 5% $CO_2$ (collagenase type II, 4 mg ml⁻¹; dispase type II, 4.7 mg ml⁻¹ in Hank's balanced salt solution $Mg^{2+}$ and $Ca^{2+}$ free (HBSS-Gibco Thermo Fisher Scientific)). Cells were then mechanically dissociated using fire-polished glass pipettes plated into wells of a 24-well plate containing poly-D-lysine/laminin-coated glass coverslips. Cells were maintained in supplemented Neurobasal medium (Gibco, Thermo Fisher Scientific) (2% (v/v) B27, 1% (v/v) GlutaMAX, 1% (v/v) antibiotic–antimycotic, mouse nerve growth factor (50 ng ml⁻¹; NGF, PeproTech) and 10 ng ml⁻¹ glial-derived neurotrophic factor (GDNF, PeproTech). Cells were used/analysed 24–72 h later.

## DRG electroporation

Electroporation was performed on dissociated cells before plating. Neurons were resuspended in 10 μl buffer R containing 1 μg of plasmid DNA. Neurons were immediately withdrawn with an electroporation pipette/tip, and inserted into a Neon transfection system (Thermo Fisher Scientific). Neurons were transfected using the following protocol: three 1,500 V pulses of 10 ms duration. Cells were used/analysed 24–72 h later.

## Metabolomic analysis of polyamines and GABA

**Chemicals and materials.** Reagents used in this study were purchased from Sigma-Aldrich unless stated otherwise. 2-Amino-3-(2-chlorophenyl)propanoic acid was purchased from FluoroChem. Methanol was obtained from Merck. Putrescine dihydrochloride and spermidine trihydrochloride were from Insight Biotechnology. Spermine tetrahydrochloride and

*N*-methyl-*N*-trimethylsilyltrifluoroacetamide (MSTFA) with 1% of chlorotrimethylsilane (TMCS) were purchased from Thermo Fisher Scientific. Deuterated polyamine standards: putrescine-D8 (PUTD8), spermidine-D8 (SPDD8) and spermine-D8 (SPMD8) were obtained from Cambridge Isotope Laboratories through CK Isotopes. GABA was obtained from MP Biomedicals.

**Sample collection and preparation.** Mouse tissues used for extraction were as follows. For brain tissue, half a hemisphere was used (average 230 mg). DRGs were dissected and collected across all regions (cervical to lumbar), with an average of 8 mg. Blood was collected during perfusion and plasma separated after centrifugation at 1,300 rpm for 10 min at room temperature. Spinal columns were removed and kept on HBSS. Each spinal column was transferred to a silicon-coated Petri dish containing HBSS and fixed in place with the dorsal side facing up using four needles. The whole spinal cord was collected (~51 mg) after exposure and removal of the overlying tissue, vertebrae and meninges. To delineate dorsal versus ventral horn, the dorsal fourth of the spinal cord (average 22 mg) was collected, and the remaining tissue was collected as ventral (average 31 mg).

Polyamines and GABA were extracted as previously described[68,69]. All tissues were suspended in 800 µl of lysis buffer (80% methanol diluted in $H_2O$ with 2% TFA). For plasma, 100 µl was diluted in 400 µl of 100% methanol with 2% TFA. For polyamine analysis, the samples were spiked with 5 µl of 1 mM deuterated polyamine samples and 10 µl of each polyamine standard (10 mM solution, 10 nmol per sample). For GABA analysis, 2-amino-3-(2-chlorophenyl)propanoic acid was used as an internal standard (10 mM solution, 20 nmol per sample). The suspensions were homogenized in a bead beater (Precellys 24, Bertin Technologies) for three cycles (6,500 rpm, 45 s) (5 cycles for DRG tissues) with 3 min incubation on dry ice between cycles. The samples were left on dry ice for 1 h and then centrifuged at 17,000$g$ for 30 min at 4 °C. The supernatants were collected and transferred into a glass vial to dry under vacuum (SpeedVac) overnight.

**Chemical derivatization.** For GABA analysis, the dried samples were resuspended in a mixture of 50 µl MSTFA with 5% TMCS and 50 µl pyridine, followed by incubation for 1 h at 60 °C at a shaking speed of 1,200 rpm. The samples were cooled down to ambient temperature, centrifuged at 8,000$g$ for 30 min at 4 °C and injected directly for two-dimensional gas chromatography mass spectrometry (GC×GC–MS) analysis (described below) with splits of 1/20 or 1/100, respectively.

For polyamine analysis, the dried samples were resuspended in 300 µl trifluoroacetic anhydride. The vials were then sealed and incubated for 1 h at 60 °C. The samples were left to dry at ambient temperature for an hour and then evaporated to dryness under vacuum (SpeedVac). Derivatized samples were resuspended in 50 µl of 100% acetonitrile and vortexed until any solid was adequately mixed. Finally, the samples were centrifuged for 5 min at 8,000$g$ at 4 °C and injected for GC×GC–MS analysis (described below) with splits of 1/2 or 1/10, respectively.

**GC×GC–MS analysis.** The samples were immediately analysed using a GC×GC–MS system comprising a gas chromatograph coupled to a quadrupole mass spectrometer (Shimadzu GCMS QP2010 Ultra) and a Shimadzu AOC-20i/s auto sampler. The first-dimension separation was carried out on a SHM5MS capillary column (30 m, 0.25 mm inner diameter, 0.25 µm film thickness, Shimadzu) while the second-dimension separation was on a BPX-50 capillary column (5 m, 0.15 mm inner diameter, 0.15 µm film thickness, SGE). Helium gas was used as a carrier gas at a 73 psi constant inlet head pressure. The modulation period was set as 4 s. The samples (1 µl) were injected at 280 °C. For GABA analysis, the oven temperature was programmed from 60 °C to 240 °C at 10 °C min$^{-1}$, and then to 320 °C at 40 °C min$^{-1}$, held at 320 °C for 6 min. For polyamine analysis, the oven temperature was programmed from 60 °C to 320 °C at 10 °C min$^{-1}$ and held at 320 °C for 8 min. The interface temperature

to the mass spectrometer was set at 300 °C and ion source was heated at 230 °C. The MS was operated at scan speeds of 20,000 amu, covering *m/z* 45–600. Electron ionization spectra were recorded at 70 eV. The standard curves were generated using mixtures of deuterated polyamines (10 nmol) and different ratios of endogenous polyamines. Phosphate-buffered saline (PBS pH 7.4) was used as the matrix to do the standard curves.

**Data processing and analysis.** Raw GC×GC MS data were processed using GCMSsolution software (v.2.72/4.50 Shimadzu) and Chromsquare software (v.2.1.6, Shimadzu) in combination with the NIST 11/s, OA_TMS, FA_ME and YUTDI in-house libraries that were used for data analysis. The annotation of metabolites was carried out by comparing them to the standards (IM spectra and retention times). The confidence of identification was further validated by manual inspection of matches between experimentally observed and standard EI spectra. The *m/z* values for identification and quantification (in bold) of the metabolites are shown in Supplementary Table 9.

**Spinal cord slice recordings**
Adult mice (5 WT and 6 KO) were anaesthetized with urethane (i.p. 2 g per kg), and perfused transcardially with ice-cold oxygenated (95% $O_2$, 5% $CO_2$) sucrose-based artificial cerebrospinal fluid (sACSF) containing 50 mM sucrose, 92 mM NaCl, 5 mM KCl, 7 mM $MgSO_4$, 0.5 mM $CaCl_2$, 1.25 mM $NaH_2PO_4$, 26 mM $NaHCO_3$ and 15 mM glucose. The lumbar spinal column was removed and immersed in ice-cold sACSF and the spinal cord was quickly obtained by laminectomy. Parasagittal slices (300 µm) were cut on a vibratome (Leica VT 1200) in ice-cold sACSF. Slices were then transferred to a submerged chamber containing oxygenated NMDG-based recovery ACSF (rACSF) for 15 min at 34 °C, containing 93 mM NMDG, 2.5 mM KCl, 1.2 mM $NaH_2PO_4$, 30 mM $NaHCO_3$, 20 mM HEPES, 25 mM glucose, 5 mM Na-ascorbate, 2 mM thiourea, 3 mM Na-pyruvate, 10 mM $MgSO_4$ and 0.5 mM $CaCl_2$, and adjusted to pH 7.4 with HCl. After recovery incubation, slices were maintained in oxygenated at room temperature until recording. ACSF was composed of 126 mM NaCl, 2.5 mM KCl, 2 mM $MgCl_2$, 2 mM $CaCl_2$, 1.25 mM $NaH_2PO_4$, 26 mM $NaHCO_3$ and 10 mM glucose. Lamina II neurons were visually identified in an Olympus BX51 microscope equipped with infrared differential interference contrast (IR-DIC) and a ×40 water-immersion-objective. Patch pipettes (4–6 MΩ) were filled with 140 mM KCl, 2 mM $MgCl_2$, 10 mM HEPES, 5 mM MgATPg, 0.4 mM NaGTP, 0.1% Lucifer-Yellow (LY, Sigma-Aldrich), pH 7.3 adjusted with KOH. Whole-cell patch clamp recordings were obtained using a Axopatch 200B amplifier (Molecular Devices), digitized with a Digidata 1440 (Molecular Devices), and recorded using pClamp 10 software (Molecular Devices). Data were filtered at 5 kHz and sampled at 10 kHz.

Neurons were recorded in current clamp to obtain measures of resting membrane potential (RMP), input resistance and rheobase. Thereafter, tetrodotoxin (TTX; 1 µM) and 6-cyano-7-nitroquinoxaline-2,3-dione disodium salt (CNQX; 10 µM) were superfused onto slices to record miniature inhibitory postsynaptic currents. Neurons were maintained at −70 mV (liquid junction potential = 5 mV). Only neurons with a resting potential more negative than −40 mV and stable access resistance (<25 MΩ) during the recording were included for subsequent analysis.

**Whole-cell patch-clamp**
Four independent patch-clamp experiments were conducted using 4 (2 male, 2 female) WT and 4 (2 male, 2 female) *Slc45a4*-KO mice. Before patch-clamp, an IB4-conjugated AF488 live dye was added to the cells for 30 min at 37 °C to distinguish IB4-binding/positive and IB4-nonbinding/negative neurons. Using this, we defined small (diameter < 25 µm) IB4$^+$ neurons as predominantly non-peptidergic nociceptors and small IB4$^-$ neurons as predominantly peptidergic nociceptors. This definition is not clear cut—there will be some overlap and some

cells that are negative for both markers (that is, C-LTMRs and CYSLTR2[+] nociceptors[70]). To confirm that the majority of IB4[-] neurons are peptidergic nociceptors, we carried out an independent experiment and cultured $Cgrp^{creERT2}$-tdTomato sensory neurons and live imaged them with IB4–AF488 and a blue live-cell dye. We counted 933 small-sized neurons. In total, 459 small cells bound to IB4–AF488, and 140 small cells (15%) were negative for both tdTomato and IB4–AF488. We determined that of the small neurons that were IB4[-], 70% were tdTomato[+], confirming our assumption that the majority of small-sized IB4[-] neurons are peptidergic nociceptors.

Experiments using the Axopatch 200B amplifier and Digidata 1550 acquisition system (Molecular Devices) were performed at room temperature. Data were sampled at 20 kHz and low-pass filtered at 5 kHz. Series resistance was compensated 80 to 85% to reduce voltage errors. AF488[+] neurons were detected with an Olympus microscope with an inbuilt GFP filter set (470/40× excitation filter, dichroic LP 495 mirror, and 525/50 emission filter). Filamental borosilicate glass capillaries (1.5 mm outer diameter, 0.84 mm inner diameter; World Precision Instruments) were pulled to form patch pipettes of 2 to 5 MΩ tip resistance and filled with an internal solution containing 100 mM K[+] gluconate, 28 mM KCl, 1 mM $MgCl_2$, 5 mM MgATP, 10 mM HEPES and 0.5 mM EGTA; the pH was adjusted to 7.3 with KOH and the osmolarity was set at 305 mOsm (using glucose). Neurons were maintained in a chamber constantly perfused with a standard extracellular solution containing 140 mM NaCl, 4.7 mM KCl, 2.5 mM $CaCl_2$, 1.2 mM $MgCl_2$, 10 mM HEPES and 10 mM glucose; the pH was adjusted to 7.3 with NaOH and the osmolarity was set at 315 mOsm (using glucose). There was a calculated −13 mV junction potential when using these solutions; voltage values were adjusted to compensate for this.

The RMP was measured in bridge mode ($I = 0$). In current-clamp mode, neurons were held at −60 mV. The input resistance was derived by measuring the change in membrane voltage caused by a 80 pA hyperpolarizing current step. Rheobase was determined by applying 50 ms depolarizing current steps, (Δ10 pA), until action potential generation. Suprathreshold activity/repetitive firing was assed using two protocols. First, 500 ms depolarizing current steps (Δ50 pA) were used until a final step current of 950 pA. Second, we used a ramp protocol that gradually injects current from 0 to 1 nA, with the duration of the ramp increasing (Δ100 ms) each time until a final ramp stimulus of 1,000 ms. All data were collected using pClamp10 and analysed by Clampfit 10 software (Molecular Devices).

## Ex vivo skin-nerve preparation

The hind paw glabrous skin and tibial nerve were dissected from adult WT ($n = 8$, 5 male and 3 female) and $Slc45a4$-KO ($n = 8$, 6 male, 2 female) mice. The skin was maintained in a perfusion chamber in the outside out configuration (epidermis facing up). The chamber was perfused with carbogen bubbled synthetic interstitial fluid (SIF: 2.0 mM $CaCl_2$, 5.5 mM glucose, 3.5 mM KCL, 26.2 mM $NaHCO_3$, 0.7 mM $MgSO_4$, 108 mM NaCl, 1.5 mM $NaH_2PO_4$, 9.5 mM Na-gluconate, 7.5 mM sucrose) at 32 °C. The tibial nerve was isolated using mineral oil (Sigma-Aldrich) in an adjacent chamber, desheathed and the nerve fibres were teased apart and placed onto a silver recording electrode. Single-fibre receptive fields were located using a blunt probe and conduction velocity was measured (C-fibre, <1.2 m s[-1]; Aδ-fibre, 1.2–10 m s[-1]; Aβ-fibre, ≥10 m s[-1]) using pulsed suprathreshold electrical currents. Receptive fields were stimulated mechanically using a 0.8 mm diameter flat probe attached to a piezo electric stimulator (Physik Instrument) or a NanoMotor stimulator (Kleindiek), both in conjunction with force sensors (Kleindiek). Stimuli were as follows: consistent force with subsequent stimuli increasing in velocity; 20 Hz vibration with a steadily increasing force; and 7 s step-increasing forces were applied to characterize mechanical thresholds and stimulus–response profiles. Thermal responses were achieved by using a receptive field isolator (ring-isolator) containing an internal

thermistor (Warner instruments), and placed on top of an identified receptive field. Then, 1 ml of 65 °C buffer was pipetted into the ring isolator (starting at 32 °C), creating a consistent temperature ramp from 32 to 50 °C in 1.21 s (±0.055). All stimuli evoked action potentials/spikes were recorded using a Powerlab v.4.0 system in conjunction with LabChart v8 software (ADInstruments).

## Animal behaviour

A total of 15 (8 male, 7 female) WT mice, 14 (7 male, 7 female) heterozygous mice and 7 (3 male, 4 female) homozygous KO mice were used for behavioural studies. All mice were littermates, and age and sex matched where possible. All mice included in the behavioural pipelines underwent all behavioural tests. Mice were tested at a consistent time of day during the light phase, in the same environment by the same experimenter. Mice were habituated to their testing environments and equipment before testing days. The experimenter was blinded to animal genotype until post-behavioural analysis was complete. Mice were randomly assigned a test environment and test order, which was achieved by random selection from their home cages. Unless otherwise stated, each test was conducted three times on different days to obtain an average baseline score.

**Open field.** Each mouse was individually placed into an open black testing box (size) with a grid marked into the floor. Mice were allowed to freely explore the open field for 3 min while being video recorded and tracked using ANY maze software. The following parameters were measured: the number of rears, number of gridline crossing, total distance travelled and maximum speed when travelling.

**Rotarod.** Mice were placed onto a speed-controlled horizontal rod/beam (Ugo Basile). Once mice were placed onto the stationary rod, an increasing speed (ramp) protocol was applied (0.5 rpm s[-1] for 20 s, 0.25 rpm s[-1] for 160 s, 0.16 rpm s[-1] for 120 s to reach a final speed of 80 rpm). Mice were monitored until they fell from the rotating rod. Latency to fall (s) and final speed (rpm) was recorded on two testing days and averaged.

**Mechanical testing.** Mice were placed into a testing box (5 × 5 × 10 cm) elevated on a wire mesh base and the mice allowed to acclimatize for 30–60 min. The plantar hind paws were tested using von Frey hairs (using the up-down method[71]) to calculate the 50% paw withdrawal threshold, or a pin prick attached to a 1 g von Frey hair to measure the latency to withdraw (using a high-frame-rate camera).

**Thermal sensory testing.** Mice were plated onto a Perspex enclosed hot plate that was set to 48 °C, 50 °C or 53 °C. The mice were observed and timed until their hind paws reacted to the hot plate (lifting, flicking, licking, cut off set to 20 s). The latency to respond to each hot plate was measured. For the Hargreaves test, the mice were placed into a test box (5 × 5 × 10 cm) on elevated glass and acclimatized for 30–60 min. The Hargreaves radiant heat source was applied to each paw three times and latency to withdraw was recorded. Noxious cold sensitivity was measured using the dry ice test. Mice were placed in test boxes, and elevated on a 5 mm thick glass platform. Dry-ice pieces were placed into a 2 ml syringe (top removed), which was placed against the glass under a visible hind paw. Latency to withdraw from the dry ice/glass was measured and three measurements were taken for each hind paw.

## Thermal gradient test

Mice were allowed to freely explore a thermal gradient apparatus (BIOSEB), which consisted of a metal platform that was heated at one end and cooled at the other. This created a thermal gradient platform ranging from 54 °C to 6 °C. During the 60 min exploration mice were video tracked using a HD webcam and ANYMaze software, and the

time in different temperature zones was analysed. This test was performed only once to avoid learning confounds. One heterozygous mouse (female) was excluded from this test owing to a camera fault during the 60 min run.

## Tonic pain assay: formalin
Mice received a single subcutaneous injection of 20 μl of 2% formalin in the left hind paw. Mice were immediately placed into a test box ($5 \times 5 \times 10$ cm), elevated on a glass platform, surrounded by angled mirrors, all above a video camera. Mice were video recorded for 1 h while the experimenter left the room. The videos were analysed offline by two blinded experimenters, each measured the duration of nocifensive behaviours (lifting, licking, flinching, shaking) of the injected hind paw every 5 min for 1 h. The first formalin phase was defined as 0–15 min, and the second phase was defined as 15–60 min.

## Statistical analysis and reproducibility
**Biochemical assay.** All biochemical assay data were derived from at least three independent experiments. Exact $n$ values are as follows. Figure 2a: $n$ indicates the number of thermal stability assays. Sucrose: $n = 5$; GABA: $n = 4$; L-ornithine: $n = 4$; L-lysine: $n = 4$; L-arginine: $n = 4$; Put: $n = 3$; Cad: $n = 3$; Spd: $n = 3$; Spm: $n = 3$. Figure 2i: $n$ indicates the number of wells containing cells. Empty: $n = 46$; WT: $n = 42$; K450R/R453E: $n = 6$; E63A: $n = 8$; Y66A: $n = 6$; Y66F: $n = 6$; D169A: $n = 10$; D173A: $n = 7$; E176A: $n = 12$; W519A: $n = 5$; W519F: $n = 5$; Y672A: $n = 5$; Y672F: $n = 4$. Extended Data Figure 1c: $n$ indicates the number of wells containing cells. No competitor: $n = 18$; 100 μM SPM: $n = 7$; 100 μM SPD: $n = 10$; 100 μM Put: $n = 8$; 100 μM Cad: $n = 8$; 1 mM L-Lys: $n = 10$; 1 mM L-Arg: $n = 7$; 1 mM L-Orn: $n = 4$; 1 mM GABA: $n = 8$. Extended Data Figure 1d: $n$ indicates the number of wells containing cells. pH washes: all empty 2 wash cycles: $n = 4$; all empty 5 wash cycles: $n = 5$; all WT 2 pH 7.5 and 9.0: $n = 10$, pH 5.5: $n = 9$; WT 5 pH 7.5: $n = 6$, pH 9.0 and pH 5.5: $n = 5$; NaCl washes: all empty 2 wash cycles: $n = 2$; all empty 5 cycles: $n = 4$; all WT 2: $n = 2$; all WT 5: $n = 4$. Cold SPD wash cycles: all $n = 5$, except for WT 2: 0.1 mM SPD: $n = 4$. 'Empty' refers to empty plasmid. Extended Data Figure 1e: $n$ values indicate the number of wells containing cells. WT: $n = 5$; D173A: $n = 3$; E176A: $n = 5$; D169A/E63A: $n = 3$; D173A/E176A: $n = 3$; Y66A: $n = 4$; Y66A/E176A: $n = 3$; Y66A/W519A: $n = 3$. Extended Data Figure 4e: $n$ values indicate the number of wells containing cells. Empty: $n = 16$; WT: $n = 15$; N718D: $n = 12$; N718A: $n = 8$; N718R: $n = 8$; N718W: $n = 8$.

**Animal work.** Data were analysed using GraphPad Prism 10 and ImageJ/Fiji (NIH). The numbers of samples and the statistical tests used for each experiment are included in the figure legends. In cases in which $n$ represents the number of animals, the number of cells has also been described. In all histology experiments, at least three sections containing cells were analysed per mouse. The number of animals used in behavioural experiments is described in the legends and the relevant Methods section. Sample sizes were calculated using power calculations and historic data.

Statistical analysis used were unpaired $t$-tests (two-tailed), Mann–Whitney $U$-tests (two-tailed), one-way ANOVA with Tukey multiple-comparison test, extra sum of squares $F$-test, Kruskal–Wallis test with Dunn's multiple-comparison test, two-way ANOVA with Holm–Šidák multiple-comparison test, RM two-way ANOVA with Holm–Šidák multiple-comparison test. Exact $n$ numbers and $P$ values are reported in the figure legends. Full statistics for each test used in the study are shown below.

Figure 3f: brain: Put, $t$-test, WT versus KO, $P = 0.828$, $t = 0.22$, d.f. = 5. Spd, $t$-test, WT versus KO, $P = 0.47$, $t = 0.79$, d.f. = 4. Spm, $t$-test, WT versus KO, $P = 0.66$, $t = 0.45$, d.f. = 6. Spinal cord: Put, $t$-test, WT versus KO, $P = 0.51$, $t = 0.69$, d.f. = 6. Spd, $t$-test, WT versus KO, $P = 0.019$, $t = 3.37$, d.f. = 5. Spm, $t$-test, WT versus KO, $P = 0.13$, $t = 1.74$, d.f. = 6. DRG: Put, $t$-test, WT versus KO, $P = 0.015$, $t = 2.84$, d.f. = 12. Spd, $t$-test,

WT versus KO, $P = 0.99$, $t = 0.001$, d.f. = 14. Spm, $t$-test, WT versus KO, $P = 0.97$, $t = 0.03$, d.f. = 14. Plasma: Put, $t$-test, WT versus KO, $P = 0.656$, $t = 0.455$, d.f. = 13. Spd, Mann–Whitney $U$-test, WT versus KO, $P = 0.014$, $U = 7$. Spm, Mann–Whitney $U$-test, WT versus KO, $P = $, $U = 0.743$. Figure 3h: two-way ANOVA, Holm–Šidák test, WT versus KO, CGRP: $P = 0.4809$, $t = 1.492$, d.f. = 20. IB4: $P = 0.7123$, $t = 1.140$, d.f. = 20. NF200: $P = 0.6926$, $t = 1.171$, d.f. = 20. TH: $P = 0.8667$, $t = 0.0678$, d.f. = 20. Figure 3j: glabrous PGP 9.5: $t$-test, $P = 0.23$, $t = 1.31$, d.f. = 7. Glabrous CGRP: Mann–Whitney $U$-test, $P = 0.412$, $U = 6$. Hairy PGP 9.5: $t$-test, $P = 0.26$, $t = 1.22$, d.f. = 7. Hairy CGRP: $t$-test, $P = 0.877$, $t = 0.159$, d.f. = 7. Figure 4a: latency to fall: one way-ANOVA, $F = 4.581$, $P = 0.0176$, d.f.$_n$, d.f.$_d$ = 2, 33. WT versus KO, $P = 0.0132$, $q = 4.267$, d.f. = 33. Final speed: one-way ANOVA, $F = 5.094$, $P = 0.0118$, d.f.$_n$, d.f.$_d$ = 2, 33. WT versus KO, $P = 0.0085$, $q = 4.513$, d.f. = 33. Figure 4b: one-way ANOVA, $F = 0.1043$, $P = 0.9013$, d.f.$_n$, d.f.$_d$ = 2, 33. Figure 4c: one-way ANOVA, $F = 7.824$. $P = 0.0017$, d.f.$_n$, d.f.$_d$ = 2, 33. WT versus KO, $P = 0.002$, $q = 5.271$, d.f. = 33. HET versus KO, $P = 0.0039$, $q = 4.938$, d.f. = 33. Figure 4d: one-way ANOVA, $F = 3.703$, $P = 0.0354$, d.f.$_n$, d.f.$_d$ = 2, 33. WT versus KO, $P = 0.0398$, $q = 3.615$, d.f. = 33. HET versus KO, $P = 0.0546$, $q = 3.413$, d.f. = 33. Figure 4e: two-way ANOVA, Holm–Šidák test, WT versus HET at 29 °C, $P = 0.0096$, $t = 2.959$, d.f. = 544. WT versus KO at 29 °C, $P = 0.0275$, $t = 2.469$, d.f. = 544. Inset: extra sum of squares $F$-test, $F = 2.44$, $P = 0.0242$, d.f.$_n$, d.f.$_d$ = 6, 586. Figure 4f: RM two-way ANOVA, Holm–Šidák test, WT versus KO at 5 min, $P = 0.0113$, $t = 3.298$, d.f. = 19.05. HET versus KO at 5 min, $P = 0.0276$, $t = 2.734$, d.f. = 17.41. HET versus KO at 10 min, $P = 0.0182$, $t = 3.237$, d.f. = 13.75. Figure 4g: phase 1: Kruskal–Wallis test, $P = 0.0208$, KW = 7.743. Dunn's WT versus KO, $P = 0.020$, $Z = 2.713$. Phase 2: ANOVA, $F = 0.4275$, $P = 0.6557$, d.f.$_n$, d.f.$_d$ = 2, 33. Figure 5c: RM two-way ANOVA, Holm–Šidák test, WT versus KO 50–950 pA, $P > 0.95$, $t = 0.007$–1.44, d.f. = 323. Figure 5d: RM two-way ANOVA, Holm–Šidák test, WT versus KO, 100–1,000 ms, $P > 0.99$, $t = 0.011$–0.655, d.f. = 150. Figure 5e: RM two-way ANOVA, Holm–Šidák test, WT versus KO at 100 pA, $P = 0.039$, $t$ 3.038, d.f. = 418; at 150 pA, $P = 0.0027$, $t = 3.807$, d.f. = 418; at 200 pA, $P = 0.0002$, $t = 4.507$, d.f. = 418; at 250 pA, $P = 0.0006$, $t = 4.212$, d.f. = 418; at 300 pA, $P = 0.047$, $t = 2.961$, d.f. = 418. Figure 5f: RM two-way ANOVA, Holm–Šidák test, WT versus KO at 300 ms, $P = 0.036$, $t = 2.623$, d.f. = 220; at 400 ms, $P = 0.021$, $t = 2.904$, d.f. = 220; at 500 ms, $P = 0.011$, $t = 3.314$, d.f. = 220; at 600 ms, $P = 0.013$, $t = 3.22$, d.f. = 220; at 700 ms, $P = 0.021$, $t = 2.998$, d.f. = 220; at 800 ms, $P = 0.015$, $t = 3.127$, d.f. = 220; at 900 ms, $P = 0.021$, $t = 2.951$, d.f. = 220; at 1,000 ms, $P = 0.037$, $t = 2.623$, d.f. = 220. Figure 5h: CM: Mann–Whitney $U$-test, $P = 0.39$, $U = 38$. CMH: Mann–Whitney $U$-test, $P = 0.93$, $U = 94$. Figure 5i RM two-way ANOVA, Holm–Šidák test, WT versus KO, $P > 0.2$, $t = 0.58$–1.2, d.f. = 108. Figure 5k: RM two-way ANOVA, Holm–Šidák test, WT versus KO, 17–402 mN, $P = 0.09, 0.002, 0.0004, <0.0001, <0.0001$, 0.0002, respectively. $t = 1.76, 3.11, 3.64, 5.99, 5.14, 3.76$, respectively. d.f. = 156. Figure 5m: $t$-test, $P = 0.044$, $t = 2.11$, d.f. = 25. Figure 5n: $t$-test, $P = 0.048$, $t = 2.07$, d.f. = 25. Figure 5o: Mann–Whitney $U$-test, $P = 0.0002$, $U = 18$. Extended Data Figure 7d: DRG: one-way ANOVA, $F = 305.7$, $P < 0.0001$, d.f.$_n$, d.f.$_d$ = 2, 10. WT versus heterozygous, $P < 0.0001$, $q = 21.28$, d.f. = 10. WT versus KO, $P < 0.0001$, $q = 34.32$, d.f. = 10. Heterozygous versus KO, $P < 0.0001$, $q = 12.37$, d.f. = 10. Spinal cord: one-way ANOVA, $F = 34.51$, $P < 0.0001$, d.f.$_n$, d.f.$_d$ = 2, 9. WT versus heterozygous, $P = 0.0262$, $q = 4.535$, d.f. = 9. WT versus KO, $P < 0.0001$, $q = 11.65$, d.f. = 9. Heterozygous versus KO, $P = 0.0018$, $q = 7.118$, d.f. = 9. Brain: one-way ANOVA, $F = 16.23$, $P = 0.001$, d.f.$_n$, d.f.$_d$ = 2, 9. WT versus heterozygous, $P = 0.1206$, $q = 3.142$, d.f. = 9. WT versus KO, $P = 0.0008$, $q = 7.996$, d.f. = 9. Heterozygous versus KO, $P = 0.0185$, $q = 4.854$, d.f. = 9. Extended Data Figure 7e: one-way ANOVA, $F = 980.5$, $P < 0.0001$, d.f.$_n$, d.f.$_d$ = 2, 3,749. WT versus heterozygous, $P < 0.0001$, $q = 38.58$, d.f. = 3,749. WT versus KO, $P < 0.0001$, $q = 61.59$, d.f. = 3,749. Heterozygous versus KO, $P < 0.0001$, $q = 27.54$, d.f. = 3,749. Extended Data Figure 8b: one-way ANOVA, $F = 0.2183$, $P = 0.8050$, d.f.$_n$, d.f.$_d$ = 2, 33. Extended Data Figure 8c: one-way ANOVA, $F = 2.308$,

$P = 0.1163$, d.f.$_n$, d.f.$_d = 2, 33$. Extended Data Figure 8d: one-way ANOVA, $F = 0.6064$, $P = 0.5513$, d.f.$_n$, d.f.$_d = 2, 33$. Extended Data Figure 8e: one-way ANOVA, $F = 0.3041$, $P = 0.7398$, d.f.$_n$, d.f.$_d = 2, 33$. Extended Data Figure 8f: one-way ANOVA, $F = 2.012$, $P = 0.1497$, d.f.$_n$, d.f.$_d = 2, 33$. Extended Data Figure 8g: one-way ANOVA, $F = 1.035$, $P = 0.3666$, d.f.$_n$, d.f.$_d = 2, 33$. Extended Data Figure 8h: one-way ANOVA, $F = 2.960$, $P = 0.0657$, d.f.$_n$, d.f.$_d = 2, 33$. Extended Data Figure 8i: one-way ANOVA, $F = 0.7461$, $P = 0.4820$, d.f.$_n$, d.f.$_d = 2, 33$. Extended Data Figure 9b: number of fragments, one-way ANOVA, $F = 0.700$, $P = 0.5283$, d.f.$_n$, d.f.$_d = 2, 7$. NMJ area, one-way ANOVA, $F = 0.4309$, $P = 0.666$, d.f.$_n$, d.f.$_d = 2, 7$. α-BuTx/synapNF, one-way ANOVA, $F = 1.197$, $P = 0.3572$, d.f.$_d = 2, 7$. synapNF/α-BuTx, one-way ANOVA, $F = 1.587$, $P = 0.2703$, d.f.$_n$, d.f.$_d = 2, 7$. Extended Data Figure 10a: brain, $t$-test, $P = 0.32$, $t = 1.104$, d.f. = 5. Spinal cord, $t$-test, $P = 0.006$, $t = 4.56$, d.f. = 5. Extended Data Figure 10b: RMP, $t$-test, $P = 0.652$, $t = 0.453$, d.f. = 41. Input resistance, $t$-test, $P = 0.181$, $t = 1.36$, d.f. = 40. Rheobase, Mann–Whitney $U$-test, $P = 0.114$, $U = 144$. Extended Data Figure 10c: amplitude, Mann–Whitney $U$-test, $P = 0.511$, $U = 71$. Frequency, Mann–Whitney $U$-test, $P = 0.84$, $U = 80$. Extended Data Figure 10d: dorsal horn, $t$-test, $P = 0.854$, $t = 0.19$, d.f. = 7. Ventral horn, $t$-test, $P = 0.04$, $t = 2.50$, d.f. = 7. Extended Data Figure 11a: C/V: CM, $t$-test, $P = 0.67$, $t = 0.419$, d.f. = 18. CMH, Mann–Whitney $U$-test, $P = 0.589$, $U = 91$. AM, Mann–Whitney $U$-test, $P = 0.38$, $U = 30$. D-hair, $t$-test, $P = 0.192$, $t = 1.362$, d.f. = 16. RA, Mann–Whitney $U$-test, $P = 0.60$, $U = 38$. SA, $t$-test, $P = 0.141$, $t = 1.538$, d.f. = 18. Extended Data Figure 11b: AM, $t$-test, $P = 0.41$, $t = 0.847$, d.f. = 16. D-hair, Mann–Whitney $U$-test, $P = 0.712$, $U = 35.50$. RA, $t$-test, $P = 0.70$, $t = 0.38$, d.f. = 17. SA, Mann–Whitney $U$-test, $P > 0.99$, $U = 49.50$. Extended Data Figure 11c: AM, RM two-way ANOVA, Holm–Šidák test, WT versus KO, $P > 0.3$, $t = 1.0–0.008$, d.f. = 96. D-hair, RM two-way ANOVA, Holm–Šidák test, WT versus KO, 75–450, $P > 0.096$, $t = 0.29, 0.15, 0.4$, respectively; 1,500 μm s$^{-1}$, $P = 0.094$, $t = 2.30$, d.f. = 64. RA-LTMR, RM two-way ANOVA, Holm–Šidák test, WT versus KO, 75–1,500 μm s$^{-1}$, $P > 0.97$, $t = 0.16, 0.26, 0.004, 0.49$, respectively, d.f. = 68. SA-LTMR, RM two-way ANOVA, Holm–Šidák test, WT versus KO, 75–1,500 μm s$^{-1}$, $P > 0.98$, $t = 0.25, 0.44, 0.25, 0.32$, respectively, d.f. = 72.

## Reporting summary

Further information on research design is available in the Nature Portfolio Reporting Summary linked to this article.

## Data availability

Data for this study were obtained from the UKB for the project "Risk factors for chronic pain" (application ID: 49572). The UKB has approval from the North West Multicentre Research Ethics Committee (MREC) as a Research Tissue Bank (RTB) approval (REC reference: 21/NW/0157, IRAS project ID: 299116). This approval means that researchers do not require separate ethical clearance and can operate under the RTB approval. The genetic and phenotypic data generated by the UKB analysed during this study are available through the UKB data access process (http://www.ukbiobank.ac.uk/register-apply/). Detailed information about the genetic data available from the UKB is available online (http://www.ukbiobank.ac.uk/scientists-3/genetic-data/ and https://biobank.ndph.ox.ac.uk/ukb/refer.cgi?id=807). The genetic and phenotypic data generated by the MVP during this study are available through the MVP data access process (https://www.mvp.va.gov/pwa/mvp-data-available-research). The genetic and phenotypic data generated by the FinnGen during this study are available through the FinnGen data access process (https://elomake.helsinki.fi/lomakkeet/124935/lomake.html). Detailed information about the genetic data available from the FinnGen is available online (https://finngen.gitbook.io/documentation/methods/genotype-imputation/genotype-data). Structural data have been deposited at the Protein Data Bank and the Electron Microscopy Data Bank: EMD-51377, 9GIU and EMD-51365, 9GHZ. Any addition data requests may be obtained from the corresponding authors. Source data are provided with this paper.

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

**Acknowledgements** This work was funded by the Wellcome Trust (Wellcome Investigator Grants to D.B., 223149/Z/21/Z; and S.N., 215519/Z/19/Z and 219531/Z/19/Z). S. Markússon is funded through a Wellcome graduate studentship (218482/Z/19/Z). Funding was provided by the UK Medical Research Council (grant ref. MR/T020113/1 to D.B. and S.J.M.); Diabetes UK (grant agreement 19/0005984 to G.B. and D.L.B.); MRC and Versus Arthritis (to the PAINSTORM consortium as part of the Advanced Pain Discovery Platform MR/W002388/1); European Union Horizon 2020 research and innovation programme (grant agreement No 633491 to DOLORisk); and the National Institute for Health and Care Research (NIHR) Oxford Health Biomedical Research Centre. M.Å. is funded by Swedish Foundation for Strategic Research (IRC15-0067) and Strategic Research Area EXODIAB (2009-1039). This research was supported in part by the Intramural Research Program of the NIH (to S.M.L.) and by US Department of Veterans Affairs (Merit Award I01 BX004820) and NIH grant R01 AA030056 (S.T. and H.R.K.). The views expressed are those of the author(s) and not necessarily those of the NHS, the NIHR, the Department of Health and Social Care or NIH.

**Author contributions** Conceptualization: B.H.S., D.L.B., G.B., H.L.H., J.C.D., J.E.L., J.L.P., J.P.-S., M.Å., M.T., S.J.M., S. Markússon, S.M.L., S.N., V.L., W.L. and Y.Y.D. Methodology: A.F., B.H.S., B.M.K., G.B., G.K., H.L.H., H.R.K., J.C.D., J.P.-S., M.Å., M.T., P.S., S.J.M., S. Markússon, S. Maxwell, S.M.L., S.T., S.R.Z., W.L. and Z.Y. Investigation: human genetics: M.Å., G.B., W.L. and S.T. Metabolomics: P.S. Protein purification and thermal stability assays: G.K. and S. Markússon Cryo-EM (Grid preparation, data collection, processing and model building): J.C.D. and S. Markússon. Neuro2A transport assays, western blotting and immunofluorescence: G.K., J.L.P. and S. Markússon. Polyamine/GABA GC–MS: A.F., S.R.Z. and Z.Y. Mouse histology and in situ hybridization: S.J.M., S.R.Z., S. Maxwell and M.T. qPCR: M.T. Animal behaviour: M.T. and S.J.M. Sensory neuron electrophysiology and skin-nerve recordings: S.J.M. Spinal cord electrophysiology: J.P.-S. Visualization: D.L.B., G.K., J.C.D., J.L.P., M.Å., M.T., P.S., S.J.M.,

S. Markússon, S. Maxwell, S.N., S.R.Z., V.L., W.L. and Y.Y.D. Funding acquisition: B.H.S., D.L.B., H.R.K., J.L.P., S.J.M., S. Markússon, S.M.L., S.N. and V.L. Project administration: B.H.S., D.L.B., G.B., J.L.P., M.Å., S.J.M., S. Markússon, S.N., V.L. and W.L. Writing—original draft: D.L.B., G.B., J.L.P., M.Å., S.J.M., S. Markússon, S.N., V.L. and W.L. All of the authors reviewed and edited the manuscript.

**Competing interests** D.L.B. has acted as a consultant for 5AM Ventures, AditumBio, AstraZeneca, Biogen, Biointervene, Combigene, GSK, Ionis, Lexicon Therapeutics, Neuvati, Novo Ventures, Olipass, Orion, Replay, SC Health Managers, Third Rock Ventures, Vida Ventures, Vertexon on behalf of Oxford University Innovation over the past two years. The PAINSTORM consortium received funding from Eli Lilly and AstraZeneca. The DOLORisk consortium received funding from Eli Lilly. J.E.L. is an employee of AstraZeneca. H.R.K. is a member of advisory boards for Altimmune and Clearmind Medicine; a consultant to Sobrera Pharmaceuticals and Altimmune; the recipient of research funding and medication supplies for an investigator-initiated study from Alkermes; a member of the American Society of Clinical Psychopharmacology's Alcohol Clinical Trials Initiative, which was supported in the past three years by Alkermes, Dicerna, Ethypharm, Imbrium, Indivior, Kinnov, Lilly, Otsuka and Pear Therapeutics; and is listed as an inventor on US provisional patent 'Multi-ancestry Genome-wide Association Meta-analysis of Buprenorphine Treatment Response'.

**Additional information**
**Correspondence and requests for materials** should be addressed to Simon Newstead or David L. Bennett.

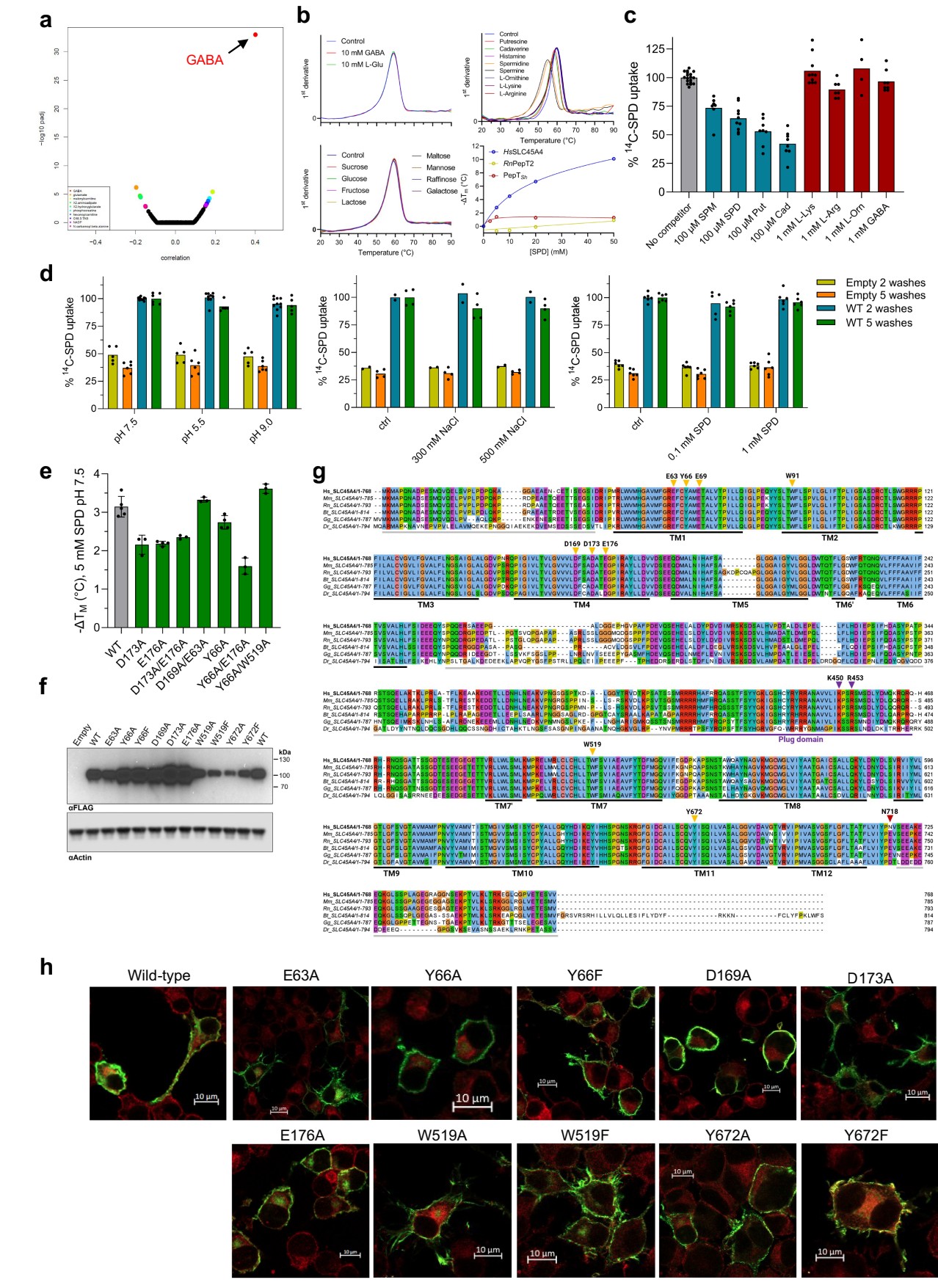

**Extended Data Fig. 1** | See next page for caption.

**Extended Data Fig. 1 | Functional analysis of human SLC45A4. a**, Correlation analysis between metabolomics and expression datasets identified GABA as being positively associated with overexpression of SLC45A4 - details of the statistical analysis are described in the methods section. **b**, Effect of different metabolites on the thermal stability of SLC45A4 in detergent and destabilizing dose-response of SLC45A4 and two other MFS transporters (PepT$_2$ and PepT$_{Sh}$) with SPD. **c**, Competition of $^{14}$C-SPD transport by WT *Hs*SLC45A4 in Neuro-2A cells showed inhibition with polyamines, but no recognition of cationic amino acids or GABA (n = 4 –18 wells containing cells). **d**, Stringent washes on Neuro2A cells confirm the radioactive signal to originate from transport, rather than binding (n = 2–10 wells containing cells). **e**, SPD thermal shift assay on binding site mutants show transport-inactive mutants to still bind SPD, further supporting the signal originating from transport, n = 3–5 wells containing cells.

**f**, Western blots confirm expression of mutants in Neuro2A cells (uncropped in source data), the blot is representative of three independently repeated experiments. **g**, Sequence alignment of human (*Hs*; Q5BKX6), mouse (*Mm*, Q0P5V9), rat (*Rn*, D4ADC6), bovine (*Bt*, E1BGZ7), chicken (*Gg*, A0A1D5NU91) and zebrafish (*Dr*, E9QH03) SLC45A4 sequences. Secondary structure, as observed in the cryo-EM structure of SLC45A4 and residues shown to be important to function are highlighted. Regions not resolved in the cryo-EM maps are highlighted in grey. **h**, Confocal microscopy shows localization of SLC45A4 (Green, αFLAG Ab) in the plasma membrane as determined by co-localization with the Na$^+$/K$^+$ ATPase (red). Micrographs are representative of > 3 images for each mutant. All data shows mean values, where applicable error bars are S.D, further particulars about exact n, and replicates see methods.

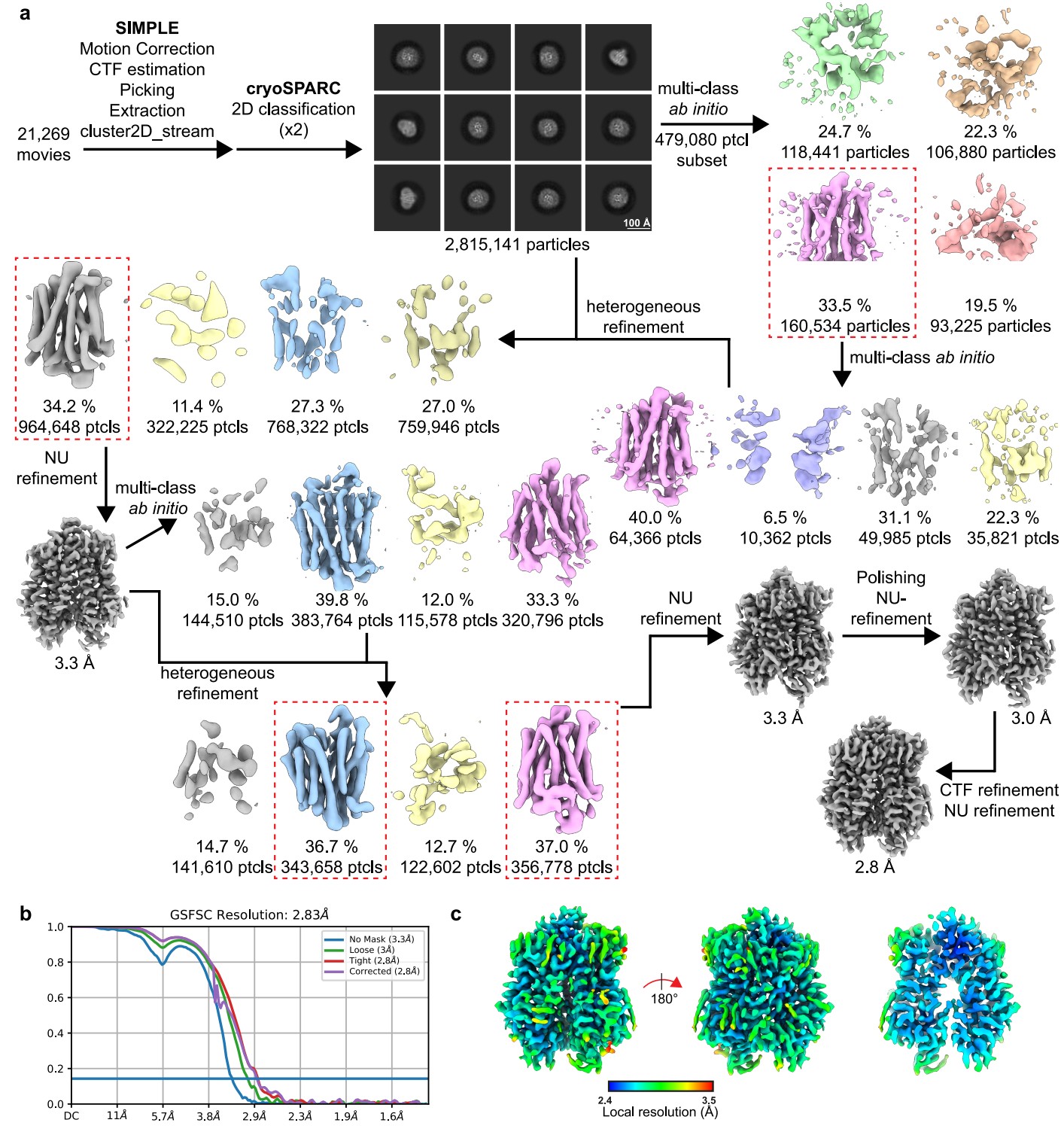

**Extended Data Fig. 2 | Cryo-EM processing workflow of SLC45A4 in LMNG, including local and global resolution estimates. a**, Image processing workflow. **b**, Gold-standard Fourier Shell Correlation (FSC) curves for global resolution estimation. **c**, Local resolution estimate of the volume.

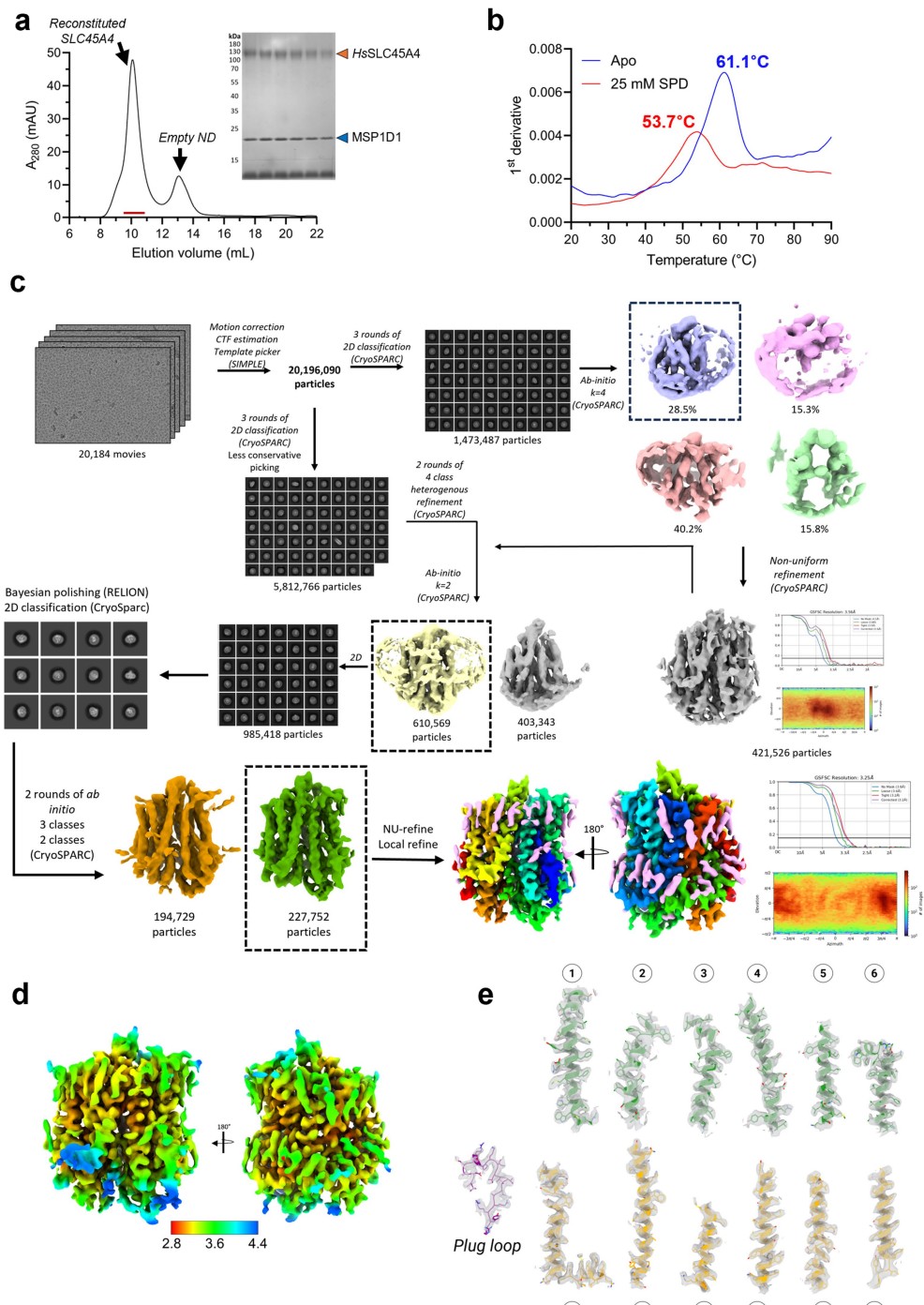

**Extended Data Fig. 3 | Nanodisc reconstitution, CryoEM data acquisition and processing. a** Reconstitution of HsSLC45A4, purified from HEK293F, into EBC:MSP1D1 nanodiscs. (uncropped gel in source data, the blot is representative of three independently repeated experiments). **b** NanoDSF scan (1st derivative) of SLC45A4 nanodiscs with and without 25 mM SPD, showing similar destabilization as observed in detergent. **c** Overview of data acquisition and processing of the SLC45A4 nanodisc sample, resulting in a well-resolved map with a global resolution of 3.25 Å. Density corresponding to protein is shown in rainbow, and lipid/detergent in pink. **d** Local resolution map. **e** Electron density of the transmembrane helices and plug domain loop.

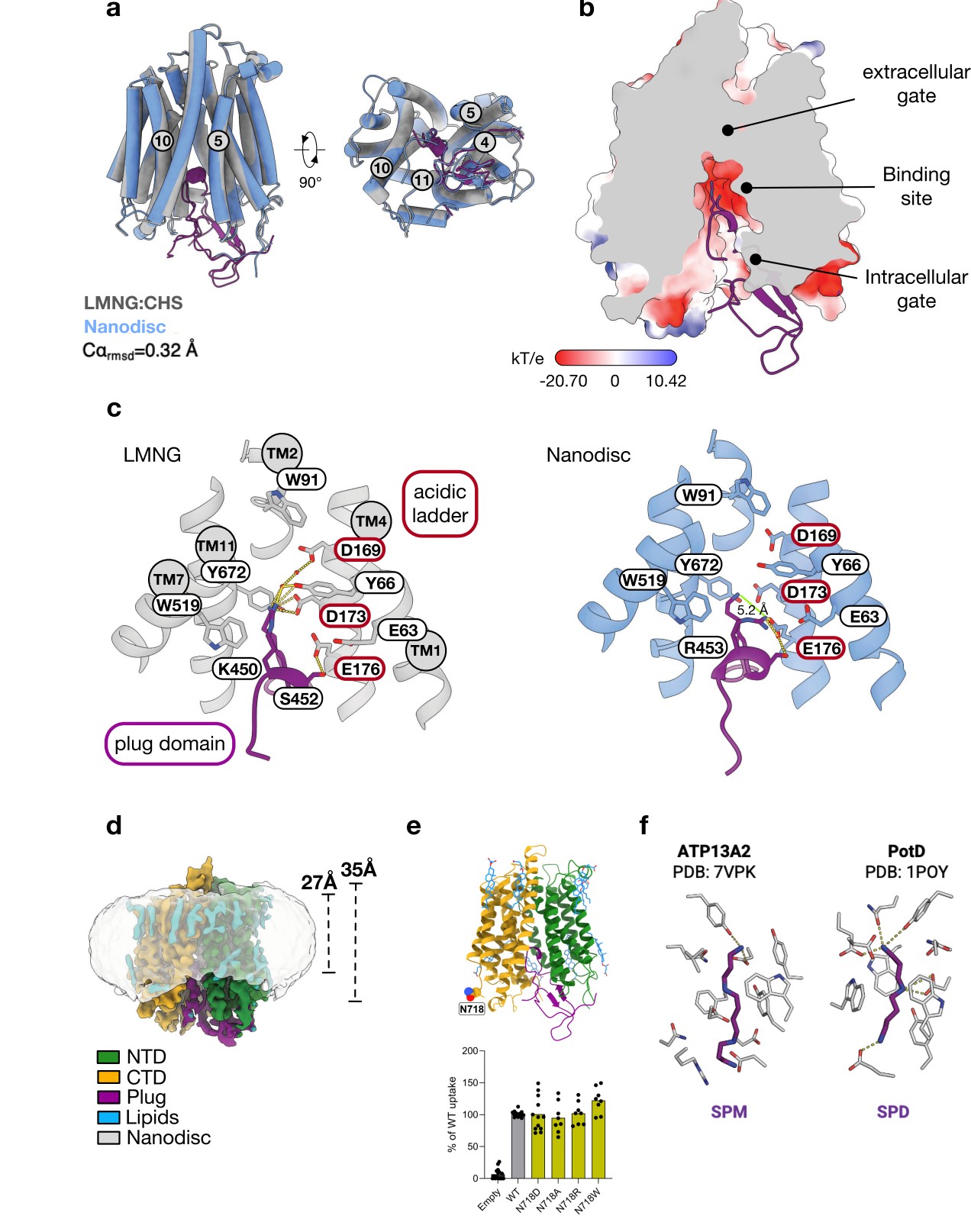

**Extended Data Fig. 4** | See next page for caption.

**Extended Data Fig. 4 | Structural analysis of SLC45A4. a**, Superposition of the SLC45A4 structures determined in LMNG and lipid nanodiscs. Intracellular gating helices are labelled. **b**, Slice through the electrostatic surface of SLC45A4 showing the location of the plug domain relative to the canonical MFS binding site. The sealed extracellular gate is indicated. **c**, View of the canonical MFS binding site showing the main polar interactions formed between Lys450, Ser452 and Arg453 on the plug domain with the side chains on the transmembrane helices. We have used these interactions as a proxy for polyamine recognition. Left hand panel shows the structure from the LMNG sample (grey helices), the right-hand panel the nanodisc structure (blue helices) with altered rotamer position for Arg453. **d**, Cryo-EM density of the SLC45A4 nanodisc structure contoured at a threshold level of 0.2 for the protein molecule (B-factor sharpened map – coloured as Fig. 2e) and 0.07 for the nanodisc (unsharpened map - grey). **e**, Functional analysis of the clinical N718D variant. Top panel shows Asn718 on the LMNG:CHS structure, bottom panel shows functional analysis of Asn718 mutants which showed no change in SPD transport activity in Neuro2A cells (n = 8–16 wells containing cells). **f**, Binding site for SPD in ATP13A2 (PDB: 7VPK) and PoTD (PDB: 1POY) respectively, showing the main polar interactions (dashed lines). All data shows mean values, further particulars about exact n, and replicates see methods.

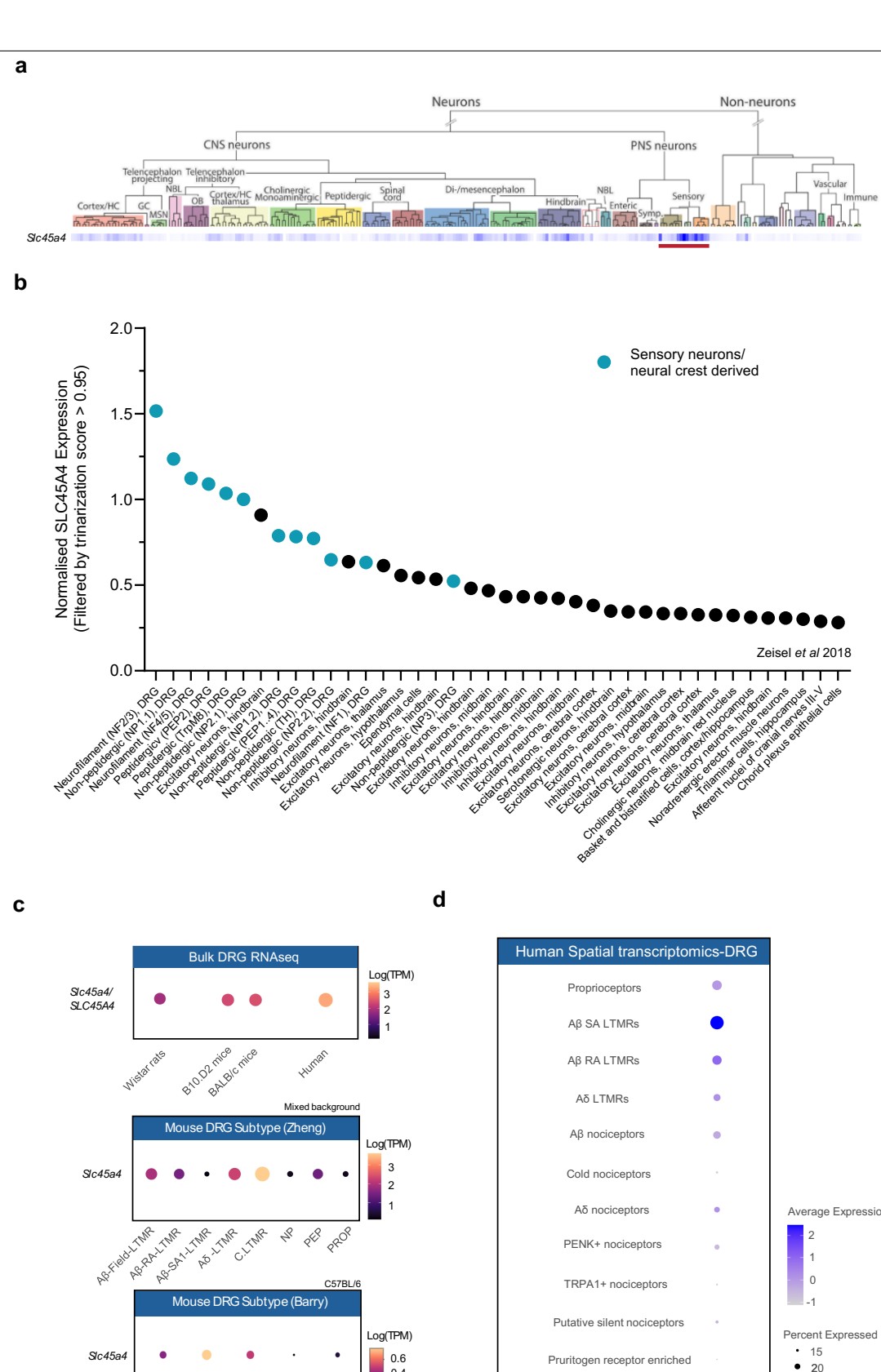

**Extended Data Fig. 5 |** See next page for caption.

**Extended Data Fig. 5 | *Slc45a4*/*SLC45A4* expression in mouse and human sensory neurons evidenced in multiple data sets. a**, Heat map of *Slc45a4* mRNA expression detected using single cell sequencing of the entire mouse nervous system (high expression in blue, no expression in white). Compared across the nervous system, *Slc45a4* mRNA is enriched in sensory neuron subpopulations (red bar). Data from Mousebrain.org,[35]. **b**, Filtering the same data set based on a trimerization score >0.95 (increased confidence of expression), 40 regions were identified, 12 regions were identified as sensory neuron subtypes and neural crest derived. 9 out of the top 10 ten regions were sensory neuron subtypes of the DRG[35]. **c**, Top: Bulk DRG RNA sequencing of different rodent species, strains and human DRG. *Slc45a4*/*SLC45A4* mRNA was detected at high levels in across rats, mouse and human DRG. Middle: Deep sequencing of 8 mouse (mixed back ground) sensory neuron subtypes[36] and Bottom: Deep sequencing of 5 mouse (C57BL6) sensory neurons subtypes[37]. Both illustrate wide expression of *Slc45a4 mRNA* in mouse sensory neurons. **d**, Data from spatial transcriptomics of human dorsal root ganglia. *SLC45A4* mRNA is expressed broadly in most subtypes including nociceptors and mechanoreceptors[38]. c and d were generated from the DRG-directory (https://livedataoxford.shinyapps.io/drg-directory/). Our overall interpretation of these datasets is that *Slc45a4* mRNA is enriched in DRG neurons versus CNS neuronal populations. There are some inconsistencies between datasets as to the relative expression in distinct sensory neuron subpopulations although our own analysis using in situ hybridization (below) supports the interpretation of broad expression across sensory neuron subpopulations.

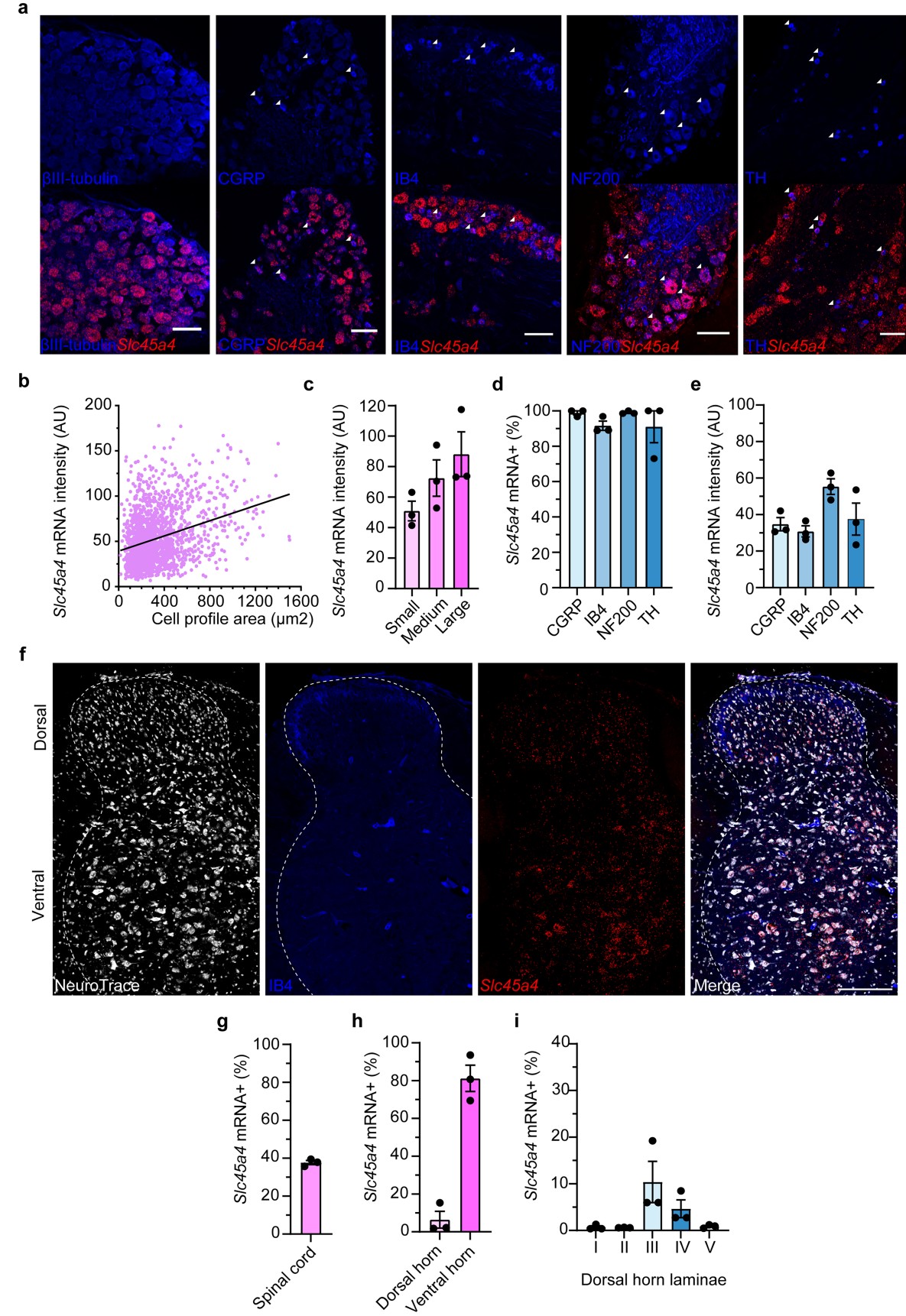

**Extended Data Fig. 6** | See next page for caption.

**Extended Data Fig. 6 | *Slc45a4* mRNA is broadly expressed in mouse sensory neurons. a**, Example images of RNA scope ISH of *Slc45a4* mRNA and co-localization with all sensory neurons (βIII-tubulin) and sensory neuron subtype markers (CGRP, IB4, NF200, TH). **b**, *Slc45a4* mRNA intensity vs cell profile area, line of best fit showing a positive correlation. (1,777 cells, from 3 mice). **c**, This data was subdivided into small medium and large cells with *Slc45a4* mRNA intensity increasing with size (n = 3 mice, 1,461 small, 269 medium, 41 large cells). **d**, Percentage of sensory neuron subtypes that colocalise and express *Slc45a4* mRNA. **e**, *Slc45a4* mRNA intensity in each sensory neuron subpopulation with the highest signal intensity in the NF200 population, (d + e: n = 3 mice, no. cells: 144 CGRP, 273 IB4, 114 NF200, 157 TH). Scale bars 100 μm. **f**, Example images of RNA scope ISH of *Slc45a4* mRNA in the lumbar spinal cord, all neurons (NeuroTrace), primary afferent terminals (IB4), and *Slc45a4* mRNA. Scale bar 200 μm. **g**, Percentage of all spinal cord neurons that express *Slc45a4* mRNA (n = 3 mice, 14,286 cells) **h**, Percentage of dorsal and ventral horn neurons that are *Slc45a4* mRNA+ (n = 3 mice, dorsal 13,179 cells, ventral 5,402 cells). **i**, Percentage of neurons from different dorsal horn laminae that express *Slc45a4* mRNA (n = 3 mice, I 749 cells, II 1,649 cells, III 1,557 cells, IV 1,384 cells, V 2,047 cells). Data mean ± s.e.m.

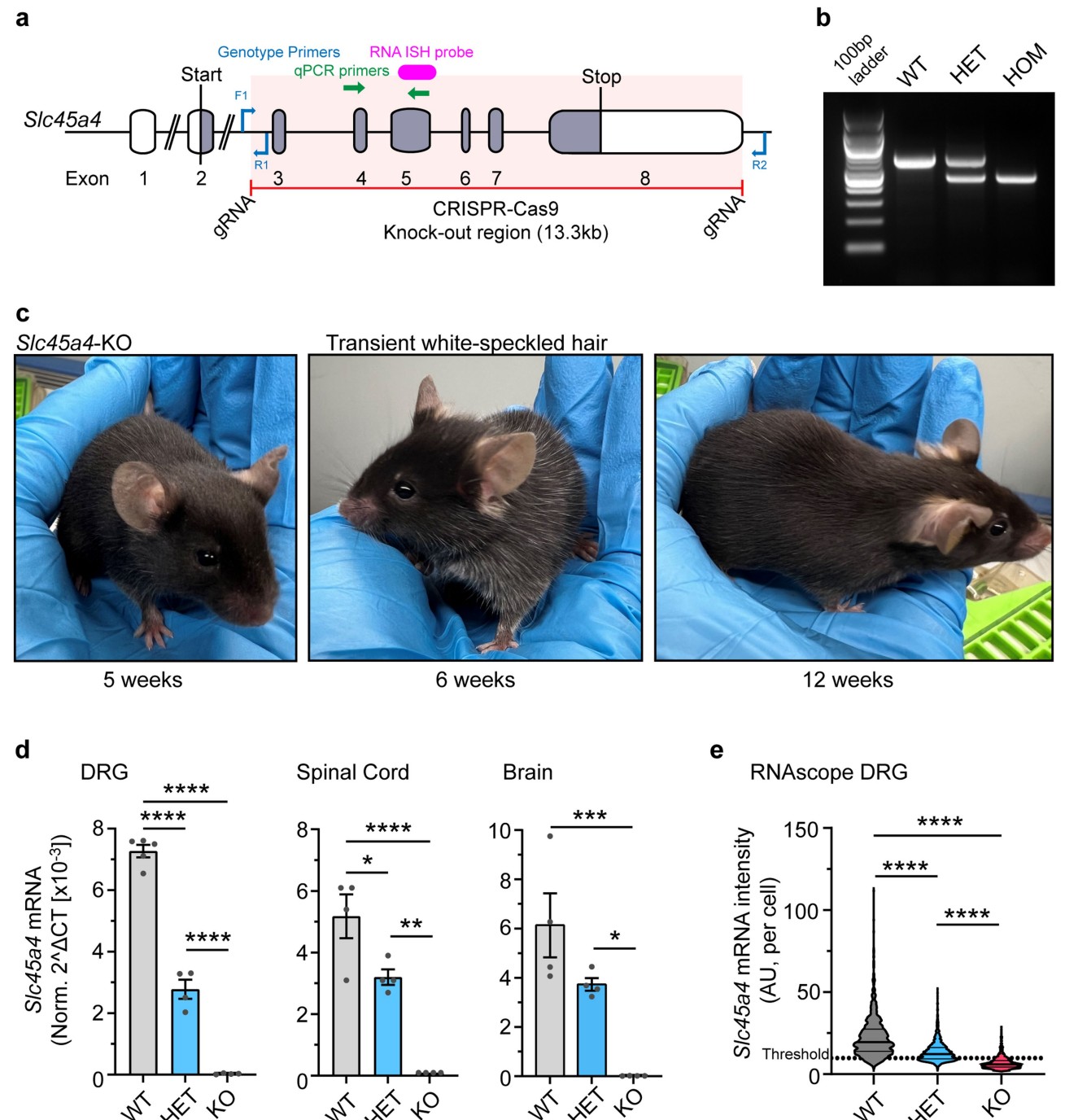

**Extended Data Fig. 7 | Generation and validation of an *Slc45a4* KO mouse.** **a**, Schematic of the *Slc45a4* locus outlining the 8 exons, the start and stop sites, the CRISPR-Cas9 KO region targeted with guide RNAs (gRNA). The illustration also highlights where the genotyping and qPCR primers target and the RNAscope probe. **b**, Example genotyping gel of WT, HET and KO mice. PCR amplified WT band 652 bp, KO band 468 bp. This was performed once for all mice (original gel in source data). **c**, Images of *Slc45a4* KO mice at different ages, at about 6 weeks mice develop a salt and pepper/white speckled hair. This is transient and returns to normal by week 12. **d**, qPCR was used to analyse the expression of *Slc45a4* mRNA in WT HET and KO mice, in DRG, Spinal cord and Brain respectively. Heterozygous mice have significantly reduced *Slc45a4* mRNA compared to WT, and there is a complete absence of expression in KO mice across tissues (n = 4 mice per group) **e**, Analysis of *Slc45a4* mRNA signal intensity in DRG sections from WT, HET and KO mice (WT n = 1349 cells from 4 mice, HET n = 1452 cells from 3 mice, KO n = 951 cells from 3 mice). Threshold set using a negative control probe. d and e, one-way ANOVA with post hoc Tukey test, d, DRG **** P < 0.0001. Spinal cord * P = 0.026, ** P = 0.0018, **** P < 0.0001. Brain * P = 0.018, *** P = 0.0008, e, **** P < 0.0001). Data mean ± s.e.m.

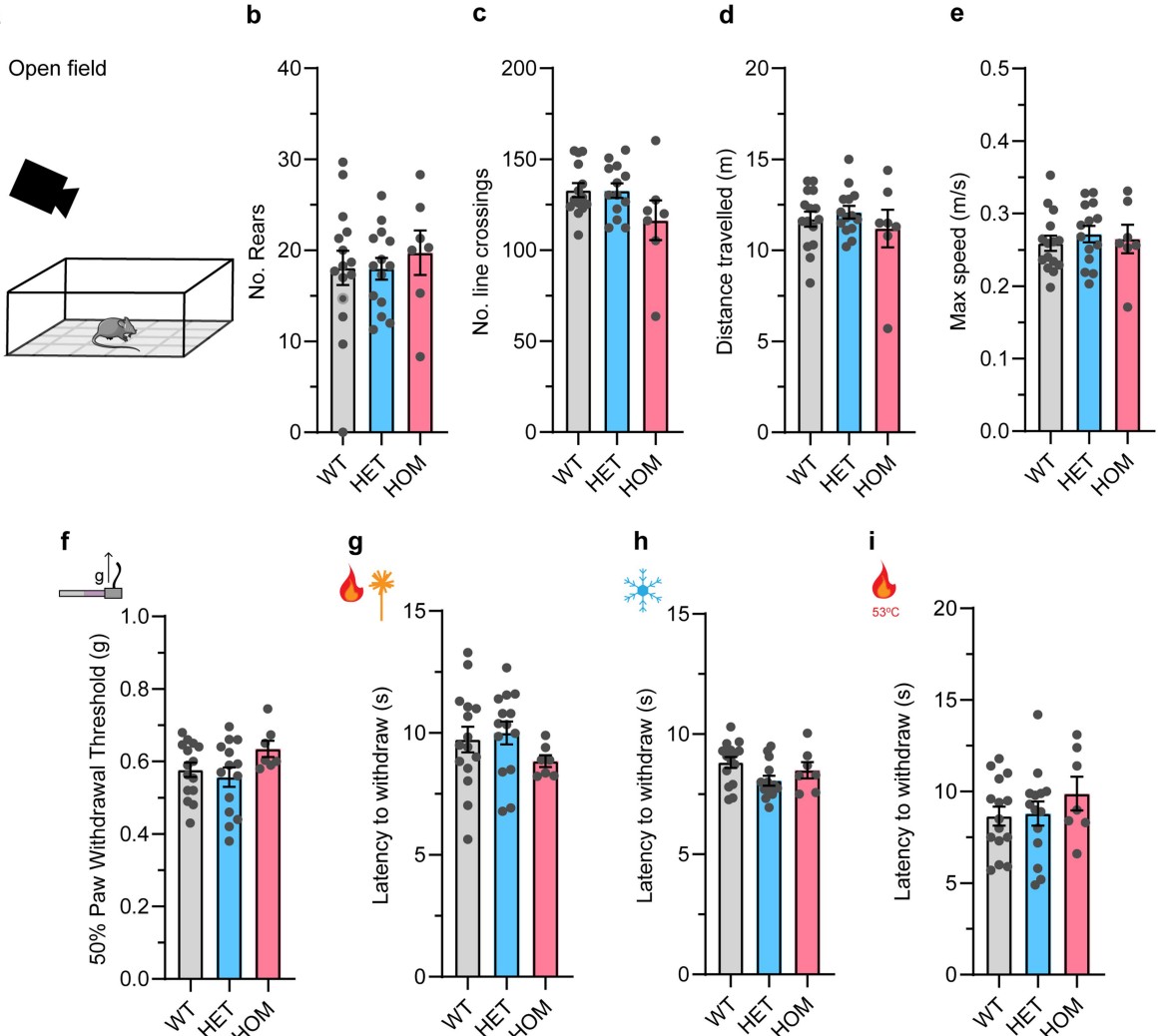

**Extended Data Fig. 8 | Behavioural assays that are normal in *Slc45a4* HET and KO mice. a**, Example of the open field assay where mice are monitored while they explore a chamber with a grid marked into the base. Number of rears **b**, No. of line crossings **c**, distance travelled **d**, and max speed **e**, are all normal in *Slc45a4* HET and KO mice. The withdrawal threshold from a von Frey stimulus **f**, the latency to withdraw from Hargreaves test **g**, the latency to withdraw from a cold stimulus (dry ice) **h**, and the latency to withdraw from a noxious 53 °C hotplate **i**, were all normal in *Slc45a4* HET and KO mice. WT n = 15 mice, HET n = 14 mice, KO n = 7 mice. one way ANOVA, with Tukey post-hoc test (P > 0.05). Data mean ± s.e.m.

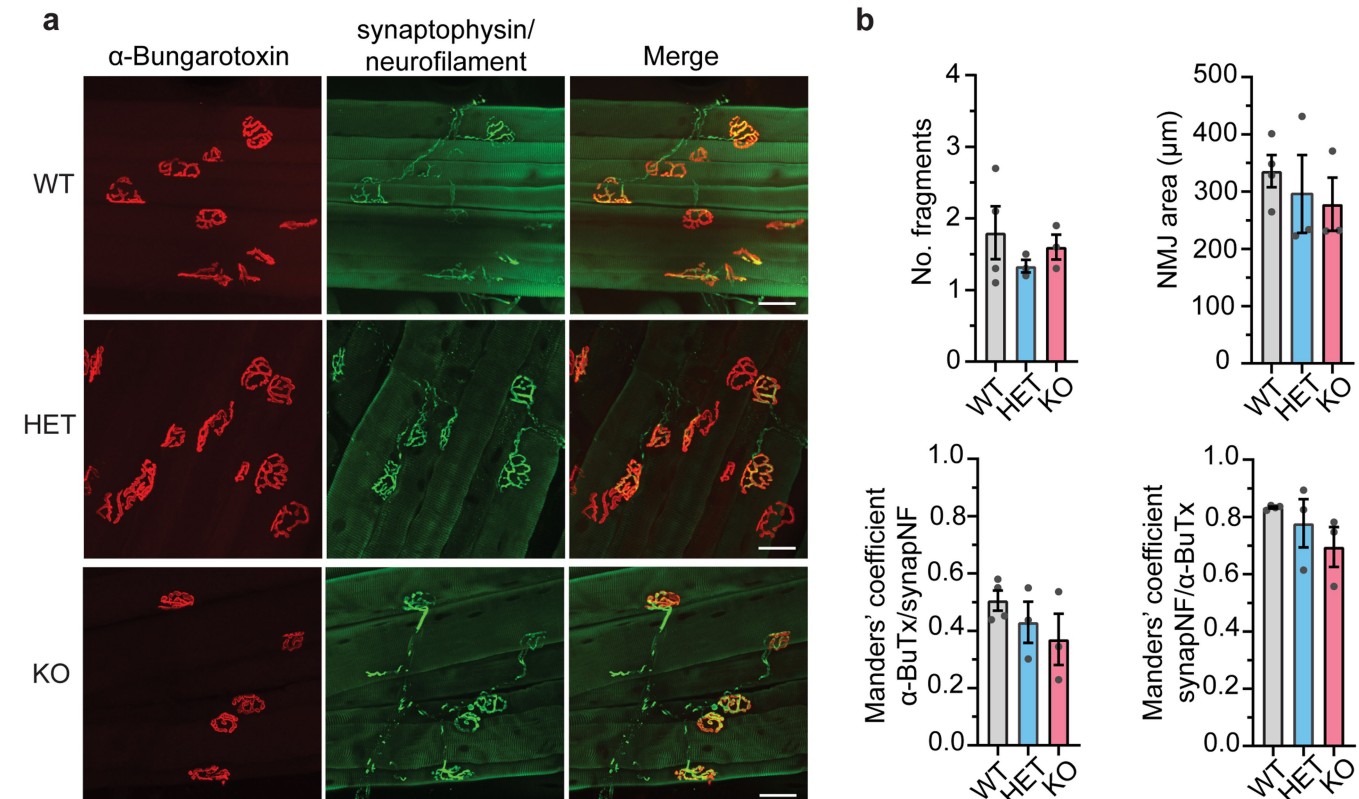

**Extended Data Fig. 9 | *Slc45a4* mutant mice have normal neuromuscular junctions. a**, Example images of neuromuscular junctions (NMJs) from WT, HET and KO mice. Red is α-bungarotoxin, green is synaptophysin & neurofilament. **b**, NJMs were analysed for each genotype, and the no. fragments, NMJ area, co-localization of α-bungarotoxin/synaptophysin & neurofilament and the co-localization of synaptophysin & neurofilament/α-bungarotoxin, were all normal. WT n = 4 mice, HET = 3 mice, KO = 3 mice, 10 NMJs were analysed per animal. one way ANOVA, with Tukey post-hoc test, P > 0.05. Data mean ± s.e.m, scale bars 40 μm. The diagram of the mouse was created using Servier Medical Art (https://smart.servier.com/), licensed under a CC BY 4.0 license.

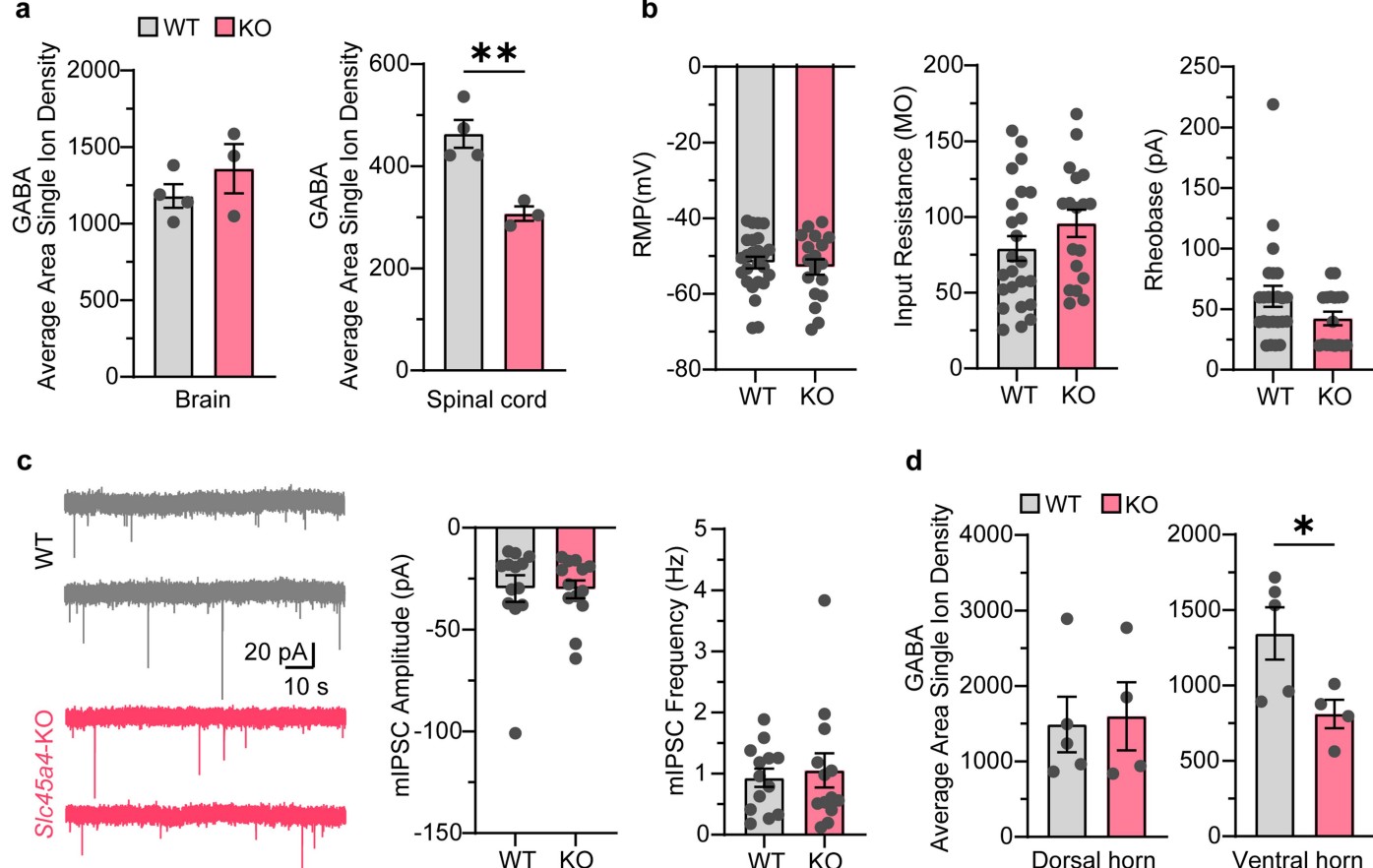

**Extended Data Fig. 10 | Loss of SLC45A4 reduces GABA levels in the spinal ventral horn. a**, Metabolomic analysis of GABA levels in the brain and spinal cord, there is a reduction in GABA abundance in KO spinal cord samples compared to WT (WT n = 4, KO n = 3 mice, t-test, ** P = 0.006). **b**, Whole-cell patch-clamp was used to characterize lamina II neurons in the spinal dorsal horn. Resting membrane potential (RMP), input resistance and rheobase, are all normal in *Slc45a4* KO neurons (WT n = 25, KO = 18 cells, t-tests or Rheobase Mann Whitney test, P > 0.05). **c**, Inhibitory tone was assessed in lamina

II neurons. Miniature inhibitory post-synaptic current (mIPSC) amplitude and frequency were normal in *Slc45a4* KO neurons compared to WT neurons (WT n = 13, KO = 13 cells, Mann Whitney tests, P > 0.05). **d**, Analysis of GABA levels in the dorsal and ventral horn of the spinal cord. GABA levels in the dorsal horn are unchanged between genotypes, but GABA levels in the ventral horn are reduced in *Slc45a4* KO neurons compared to WT neurons (WT = 5, KO = 4 mice, t-test, * P = 0.04). Data mean ± s.e.m.

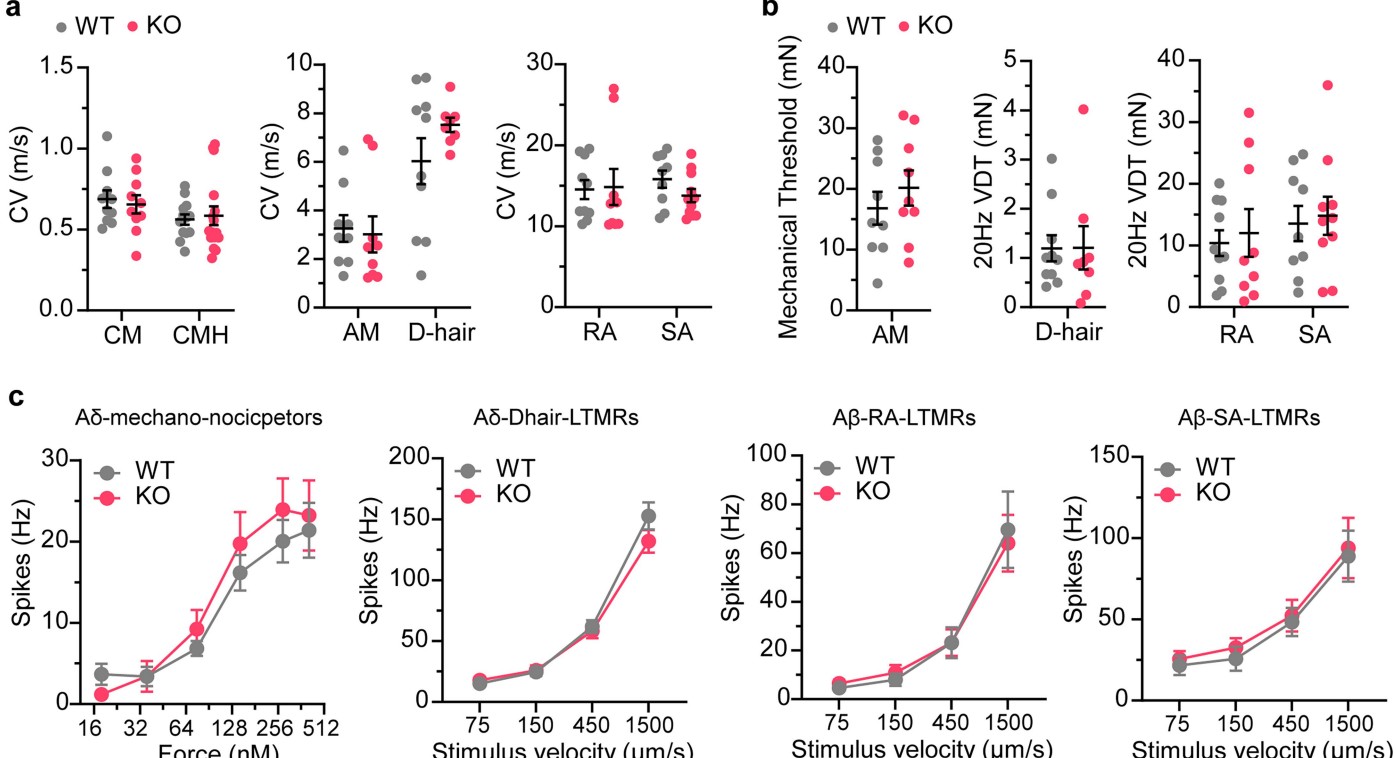

**Extended Data Fig. 11 | SLC45A4 does not regulate primary afferent conduction velocity or the mechano-sensitivity of Aδ-nociceptors or Aδ/Aβ-low threshold mechanoreceptors.** The conduction velocity, **a**, is normal in all *Slc45a4* KO nerve fibres vs WT fibres (t-tests, P > 0.05). **b**, the mechanical threshold or 20 Hz vibration detection threshold (VDT) are normal in *Slc45a4* KO AMs, D-hairs, rapidly adapting, and slowly adapting LTMRs (t-tests, P > 0.05). **c**, The stimulus-response function to increasing force is normal in *Slc45a4* KO

AM fibres. The stimulus-response function to moving mechanical stimuli of different velocities is normal in *Slc45a4* KO D-hairs, RA-LTMRS and SA-LTMRs (RM two-way ANOVA, post hoc Holm-Sidak's tests, P > 0.05). N numbers are as follows; CMs (WT & KO n = 10 units), CMHs (WT = 12, KO n = 16 units) AMs (WT & KO n = 9 units), D-hairs (WT n = 10, KO = 8 units), Rapidly adapting (WT n = 10, KO = 9 units), and slowly adapting (WT n = 9, KO n = 11 units) LTMRs Data mean ± s.e.m.

# Reporting Summary

## Statistics

For all statistical analyses, confirm that the following items are present in the figure legend, table legend, main text, or Methods section.

| n/a | Confirmed | |
|---|---|---|
| ☐ | ☒ | The exact sample size (*n*) for each experimental group/condition, given as a discrete number and unit of measurement |
| ☐ | ☒ | A statement on whether measurements were taken from distinct samples or whether the same sample was measured repeatedly |
| ☐ | ☒ | The statistical test(s) used AND whether they are one- or two-sided<br>*Only common tests should be described solely by name; describe more complex techniques in the Methods section.* |
| ☐ | ☒ | A description of all covariates tested |
| ☐ | ☒ | A description of any assumptions or corrections, such as tests of normality and adjustment for multiple comparisons |
| ☐ | ☒ | A full description of the statistical parameters including central tendency (e.g. means) or other basic estimates (e.g. regression coefficient) AND variation (e.g. standard deviation) or associated estimates of uncertainty (e.g. confidence intervals) |
| ☐ | ☒ | For null hypothesis testing, the test statistic (e.g. *F*, *t*, *r*) with confidence intervals, effect sizes, degrees of freedom and *P* value noted<br>*Give P values as exact values whenever suitable.* |
| ☒ | ☐ | For Bayesian analysis, information on the choice of priors and Markov chain Monte Carlo settings |
| ☒ | ☐ | For hierarchical and complex designs, identification of the appropriate level for tests and full reporting of outcomes |
| ☐ | ☒ | Estimates of effect sizes (e.g. Cohen's *d*, Pearson's *r*), indicating how they were calculated |

*Our web collection on statistics for biologists contains articles on many of the points above.*

## Software and code

Policy information about availability of computer code

| Data collection | Thermofisher EPU used to collect data on microscope, ANYMAZE for mouse tracking, pClamp10 for collecting patch-clamp electrophysiology data, LabChart 8 for Skin-nerve data collections, Zeiss Zen Black for confocal imaging. UK Biobank genotyping: the Applied Biosystems UK Bi LEVE Axiom Array by Affymetrix and the Applied Biosystems UK Biobank Axiom Array. Microscopy: ZeissBlack 2012 (V 8 1 0 484) |
|---|---|
| Data analysis | SIMPLE3.0 - available on https://github.com/hael/SIMPLE<br>cryoSPARC 3.3.1 (https://cryosparc.com/)<br>RELION3.l.3 - published and freely available<br>PHENIX v1.20.1-4487 - published and freely available<br>Coot v0.9.8.1 EL- published and freely available<br>ISOLDE v 1.6- published and freely available<br>MolProbity v4.4 - published and freely available<br>PyMol v2.6.2- published and freely available<br>ChimeraX 1.5rc202211091945 - published and freely available<br>Graph Pad PRISM 10 - commercially available<br>lmageJ/Fuiji - 1.54g, freely available<br>ZeissBlue (v3.9) - commercially available<br>Cellpose 2.0 - freely avalible<br>Clampfit 10 software - commercially availible<br>KING software (version 2.3.2),<br>PLINK (version l.90b6.21,https://www.cog-genomics.org/plink/l.9/),<br>PLINK2 (version 2.00a5, https://www.cog-genomics.org/plink/2.0/) |

REGENIE (version 3.4.1),
R version4 .3.3,
Shimadzu GCMS solution version 2.72 and version 4.50.
Chromsquare software (v2.1.6, Shimadzu)
NIST 11/s, OA_TMS, FA_ME and YUTDI - in house libraries
FUMA v1.5.2 (fuma.ctglab.nl)

For manuscripts utilizing custom algorithms or software that are central to the research but not yet described in published literature, software must be made available to editors and reviewers. We strongly encourage code deposition in a community repository (e.g. GitHub). See the Nature Portfolio guidelines for submitting code & software for further information.

## Data

Policy information about availability of data

All manuscripts must include a data availability statement. This statement should provide the following information, where applicable:

- Accession codes, unique identifiers, or web links for publicly available datasets
- A description of any restrictions on data availability
- For clinical datasets or third party data, please ensure that the statement adheres to our policy

Data for this study were obtained from the UK Biobank for project "Risk factors for chronic pain," Application ID: 49572. UK Biobank has approval from the North West Multicentre Research Ethics Committee (MREC) as a Research Tissue Bank (RTB) approval, REC reference: 21/NW/0157, IRAS project ID: 299116. This approval means that researchers do not require separate ethical clearance and can operate under the RTB approval. The genetic and phenotypic data generated by UK Biobank analysed during this study are available via the UK Biobank data access process (see http://www.ukbiobank.ac.uk/register-apply/). Detailed information about the genetic data available from UK Biobank is available at http://www.ukbiobank.ac.uk/scientists-3/genetic-data/ and https://biobank.ndph.ox.ac.uk/ukb/refer.cgi?id=807. The genetic and phenotypic data generated by the Million Veteran Program (MVP) during this study are available via the MVP data access process (see https://www.mvp.va.gov/pwa/mvp-data-available-research). The genetic and phenotypic data generated by the FinnGen during this study are available via the Finnen data access process (see https://elomake.helsinki.fi/lomakkeet/124935/lomake.html.) Detailed information about the genetic data available from the FinnGen is available at https://finngen.gitbook.io/documentation/methods/genotype-imputation/genotype-data. Structural data has been deposited in wwPDB, EMD-51377, 9GIU, (https://doi.org/10.2210/pdb9GIU/pdb) and EMD-51365, 9GHZ (https://doi.org/10.2210/pdb9GHZ/pdb). Datasets relating to biochemical assays and studies in the mouse have been made available as source data alongside this manuscript. Any addition data requests may be obtained from the corresponding authors.

## Research involving human participants, their data, or biological material

Policy information about studies with human participants or human data. See also policy information about sex, gender (identity/presentation), and sexual orientation and race, ethnicity and racism.

| Reporting on sex and gender | Sex was asked in this study and checked by genotype data. We then removed individuals who showed discordances between self-reported sex and genotyped sex.<br>Overall this study contains 132,552 individuals from which (sex, n, freq), Female 74457 0.562,<br>Male 58095 0.438. |
|---|---|
| Reporting on race, ethnicity, or other socially relevant groupings | We identified five distinct subpopulations within the UK Biobank: African (9,059 samples), Ad Mixed American (605 samples), East Asian (2,572 samples), European (464,586 samples), and South Asian (9,604 samples). Additionally, there were 1,951 samples for which ancestry could not be determined and were thus categorized as missing. For GWAS association analysis we only included European participants. |
| Population characteristics | The GWAS contains 132,552 individuals of European ancestry from the UK Biobank for a continuous outcome (i.e., pain intensity, the most bothersome chronic pain). The characteristics includes mean age 66.7 +-7.63SD(when completed the questionnaire), sex (Female 74457, Male 58095). |
| Recruitment | Individuals were used from UK Biobank (project" Risk factors for chronic pain", ID: 49572), all participants gave informed consent. |
| Ethics oversight | UK Biobank has approval from the North West Multicentre Research Ethics Committee (MREC) as a Research Tissue Bank (RTB) approval, REC reference: 21/NW/0157, IRAS project ID: 299116. This approval means that researchers do not require separate ethical clearance and can operate under the RTB approval. |

Note that full information on the approval of the study protocol must also be provided in the manuscript.

## Field-specific reporting

Please select the one below that is the best fit for your research. If you are not sure, read the appropriate sections before making your selection.

☒ Life sciences ☐ Behavioural & social sciences ☐ Ecological, evolutionary & environmental sciences

For a reference copy of the document with all sections, see nature.com/documents/nr-reporting-summary-flat.pdf

# Life sciences study design

All studies must disclose on these points even when the disclosure is negative.

| | |
|---|---|
| Sample size | Human sample sizes we chosen based on all data avalible to us i.e. UK Biobank. For mouse work work sample sizes were calculated using power calculations. Using previous data generated in the lab, animal behaviour, primary outcome, evoked behaviour (von Frey mechanical response), unit mouse 7 biological replicates, effect size d =1.786, power 0.8, with an alpha of 0.05, using two tailed t-test. Secondary measures were electrophysiology and anatomy, detemined by additional power calculations and previous data. Sample sizes were not pre-determined for biochemical/transport assays. |
| Data exclusions | Participants who have withdrawn (153 as of 15/09/2023) were excluded from the analysis. People who self-reported fibromyalgia (fl20009), chronic fatigue syndrome/myalgic-encephalomyelitis (fl20010) or chronic pain all over the body (fl20021) were excluded from the analysis (11,951 with chronic pain and 821 with no chronic pain). Additional quality control measures were applied that further excluded 367 samples where the reported sex did not match the inferred sex from their genetic data, 651 samples with suspected sex chromosome aneuploidy, and 188 samples with more than ten putative third-degree relatives. In total, 1,024 samples were excluded, with some samples falling into multiple exclusion categories. Animal work, thermal gradient test, one heterozygous mouse (female) was excluded from this test due to a camera fault during the 60 min run. |
| Replication | Structure solved in detergent and lipid nanodisks. Genetic data was replicated in two independent databases/studies, Million Veterans program and FinnGen, all attempts were successful. Animal data was replicated in multiple cohorts of behaviour, cohort replication was successful. Histological data was all replicated in samples collected from multiple animals, each experiment was carried out more than 3 times, in multiple tissues per animal. Electrophysiological data was successfully replicated in multiple independent experiments each on different days in this case biological replication and indpendent replication are equal. In most cases thoughout the manuscript the number of animals is used to demostrate biological replication. Biochemical assays were successfully replicated in biological replicates (e.g. well containing cells) and from at least 3 independent experiments. |
| Randomization | Randomisation was used prior to animal behaviour. Experimenters would randomly choose an animal cage, and then randomly choose an animal from that cage to be tested. Doing so randomised the order each animal was tested on each testing day. A similar approach was taken to selecting animals for electrophysiology and sample collection for tissue analysis. Samples were blinded so that down stream experiments (i.e. histology and biochemical assays) we blind and randomised, until analysis was complete. Randomisation related to group allocation for cell based assays and human participants was not required or relevant, all cell-based assays were sampled simultaneously and all human particpants that passed QC were included in the study (details of QC in the methods). |
| Blinding | Investigators were blinded to group allocation during data collection and analysis |

# Reporting for specific materials, systems and methods

We require information from authors about some types of materials, experimental systems and methods used in many studies. Here, indicate whether each material, system or method listed is relevant to your study. If you are not sure if a list item applies to your research, read the appropriate section before selecting a response.

## Materials & experimental systems

| n/a | Involved in the study |
|---|---|
| ☐ | ☒ Antibodies |
| ☐ | ☒ Eukaryotic cell lines |
| ☒ | ☐ Palaeontology and archaeology |
| ☐ | ☒ Animals and other organisms |
| ☒ | ☐ Clinical data |
| ☒ | ☐ Dual use research of concern |
| ☒ | ☐ Plants |

## Methods

| n/a | Involved in the study |
|---|---|
| ☒ | ☐ ChIP-seq |
| ☒ | ☐ Flow cytometry |
| ☒ | ☐ MRI-based neuroimaging |

## Antibodies

| | |
|---|---|
| Antibodies used | Primary Antibody Source Identifier<br>Rb NeuN (1:500) Abcam ab177487 [EPR12763 monoclonal]<br>Ms βIII-Tubulin (1:500) R&D Systems MAB1195 [# TuJ-1 monoclonal]<br>βIII-Tubulin-FITC conjugated (1:500) Abcam ab224978 [EP1569Y monoclonal]<br>Sh CGRP (1:500) Enzo BML-CA1137 [polyclonal]<br>Rb CGRP (1:500) BMA Biomedicals T-4032 [polyclonal]<br>IB4, streptavidin conjugated (1:100) Sigma L2140<br>Ch NF200 (1:5000) Abcam ab4680 [polyclonal]<br>Sh TH (1:500) Millipore AB1542 [polyclonal]<br>Rb TH (1:500) Millipore AB152 [polyclonal]<br>Ms anti-FLAG (WB: 1:5000, ICC 1:200) (Merck F1804), [M2, monoclonal]<br>Ms anti-β-actin (1:10000) (Merck A2228), [AC-74, monoclonal]<br>Rb anti-Na+/K+ ATPase (1:50) (ThermoFisher MA5-32184), [ST0533, monoclonal]<br>Rb PGP9.5 (1:400) Proteintech 14730-1-AP [polyclonal] |

Rb CGRP (1:400) Merck C8198 [polyclonal]
Secondary Antibody Source Identifier
Rb PcBI (1:250) Life Technology, P-10994
Ms PcBI (1:250) Thermofisher P31582
Stp PcBI (1:250) Life Technology S11222
Sh Alexa 546 (1:500) Life Technology A21098
Rb Alexa 488 (1:500) Life Technology A11008
Rb Alexa 546 (1:500) Life Technology A11010
Stp Alexa 488 (1:500) Life Technology S11223
Ch Alexa 488 (1:500) Abeam ab150169
Ch Alexa 546 (1:500) Life Technology A11040
Stp Alexa 546 (1:500) Life Technology S11225
NeuroTrace (1:10) Life Technology N21382
Goat anti-Mouse lgG (H+L) AlexaFluor-488 (1:200) (ThermoFisher A28175)
Goat anti-Rabbit lgG (H+L) AlexaFluor-647 (1:200)(ThermoFisher A-21245)

Validation

All validation data was taken from the associated manufacturers websites

Rb NeuN (1:500) Abcam ab177487 [EPR12763 monoclonal]
Anti-NeuN antibody [EPR12763] - Neuronal Marker (ab177487) was developed by Abcam using patented rabbit monoclonal antibody technology and is validated for use in Flow Cyt (Intra), ICC/IF, IHC (PFA fixed), IHC-Fr, IHC-P, WB, mIHC in cat, common marmoset, dog, human, mouse, rat, sheep, zebrafish samples. Abcam's high quality manufacturing and validation processes ensure NeuN antigen antibody (ab177487) has high sensitivity and specificity alongside high lot-to-lot consistency and reproducibility.

Ms βIII-Tubulin (1:500) R&D Systems MAB1195 [# TuJ-1 monoclonal]
Detects mammalian and chicken neuron-specific beta -III tubulin but not other beta -tubulin isotypes in Western blots.

βIII-Tubulin-FITC conjugated (1:500) Abcam ab224978 [EP1569Y monoclonal]
Rabbit Recombinant Monoclonal Beta-3-tubulin antibody - conjugated to FITC. Suitable for ICC/IF, Flow Cyt (Intra) and reacts with Human samples. KO validated.

Sh CGRP (1:500) Enzo BML-CA1137 [polyclonal]
Immunogen synthetic peptide corresponding to a portion of rat α-calcitonin gene-related peptide (CGRP). Likely to react with other mammalien species (based on homology, not tested). Test tissues: Rat thoracolumbar spinal cord.

Rb CGRP (1:500) BMA Biomedicals T-4032 [polyclonal]
This antibody has been tested and validated in ELISA against α-CGRP. Other applications like immunohistochemistry (IHC), FACS or Western Blot may work as well. Optimal dilutions should be determined by the end user.

IB4, streptavidin conjugated (1:100) Sigma L2140
Agglutination activity is expressed in μg/mL and is determined from serial dilutions of a 1 mg/mL solution using phosphate buffered saline, pH 6.8, containing, for each lectin, calcium, magnesium, and manganese at different concentrations. This activity is the lowest concentration to agglutinate a 2% suspension of appropriate erythrocytes after 1 hr incubation at 25 °C.

Ch NF200 (1:5000) Abcam ab4680 [polyclonal]
Anti-Neurofilament heavy polypeptide antibody (ab4680) is a Chicken Polyclonal antibody and is validated for use in ICC, IHC-FrFl, WB.

Sh TH (1:500) Millipore AB1542 [polyclonal]
Tyrosine Hydroxylase (TH, Tyrosine Monooxygenase). Cross-reacts with all mammalian forms tested to date and some non-mammalian forms. Routinely evaluated by Western Blot on mouse brain lysates. The antibody gives specific labeling of noradrenergic axons in primate cerebral cortex.

Rb TH (1:500) Millipore AB152 [polyclonal]
Tyrosine hydroxylase. By western blot, AB152 selective labels a single band at approximately 62kDa (reduced) corresponding to Tyrosine Hydroxylase. It is expected that the antibody will react with most mammalian and many non-mammalian species. It has been reported that this antibody does not work on paraffin embedded human tissue.

Ms anti-FLAG (Merck F1804), [M2, monoclonal]
The M2 antibody will recognize a FLAG® peptide sequence at the N-terminus, Met-N-terminus, C-terminus, or internal sites of a fusion protein. Binding of the M2 antibody is not dependent on calcium. The monoclonal antibody detects only the target protein band(s) on a Western blot from an E. coli, plant or mammalian crude cell lysate.

Ms anti-β-actin (Merck A2228), [AC-74, monoclonal]
Monoclonal mouse anti-actin antibody was used as a loading control for western blot analysis of immunoprecipitated proteins from rat dorsal root ganglion cocultures. Western blot analysis of MDCK cell lysates were performed using monoclonal anti-actin antibody as a primary antibody.

Rb anti-Na+/K+ ATPase (ThermoFisher MA5-32184), [ST0533, monoclonal]
Western blot was performed and a 110kDa band corresponding to ATP1A1 was observed across cell lines and tissues tested.

Rb PGP9.5 (1:400) Proteintech 14730-1-AP [polyclonal]
Various lysates were subjected to SDS PAGE followed by western blot with 14730-1-AP (UCHL1/PGP9.5 antibody)

Rb CGRP (1:400) Merck C8198 [polyclonal]

Anti-Calcitonin Gene Related Peptide antibody produced in rabbit was used for immunohistochemistry of trigeminal ganglia cell cultures and for immunocytochemistry of mouse lung slices.

## Eukaryotic cell lines

Policy information about cell lines and Sex and Gender in Research

| Cell line source(s) | Neuro-2A mouse neuroblasts (ATCC CCL-131), Human Embryonic Kidney 293-F (Thermofisher Scientific FreeStyle™ 293-F Cells) |
|---|---|
| Authentication | Authentication based on supplier informaiton, not authenticated in-house |
| Mycoplasma contamination | Tested negative for mycoplasma contamination |
| Commonly misidentified lines (See ICLAC register) | n/a |

## Animals and other research organisms

Policy information about studies involving animals; ARRIVE guidelines recommended for reporting animal research, and Sex and Gender in Research

| Laboratory animals | Wild-type mice: C567Bl/6J. Mutant mice: Slc45a4+/- and Slc45a4-/- on a C567Bl/6J background. All experiments were carried out on adult mice, in particular when the mice were above 10-12 weeks old (when salt/pepper hair colour had normalised, and the experiments could be blinded.) |
|---|---|
| Wild animals | No wild animals were used |
| Reporting on sex | Both male and female mice were used, it is reported in the methods how many males and females were used. Histology and metabolomics, samples from males and females. Patch-clamp DRG: 4 (2 males 2 females) WT and 4 (2 males 2 females) SLC45A4 KO mice. Spinal and skin-nerve electrophysiology females (WT, 3 KO 2) males (WT 5 KO 6), Behaviour: A total of 15 (8 male, 7 female) wild type mice, 14 (7 male, 7 female) heterozygous mice, and 7 (3 male, 4 female) homozygous knockout mice were used. No differences were observed based on Sex. |
| Field-collected samples | No field samples were collected in this study |
| Ethics oversight | All procedures complied with the UK Animals (Scientific Procedures) Act (1986) and were performed under a UK Home Office Project Licence in accordance with University of Oxford Policy on the Use of Animals in Scientific Research. |

Note that full information on the approval of the study protocol must also be provided in the manuscript.

## Plants

| Seed stocks | n/a |
|---|---|
| Novel plant genotypes | n/a |
| Authentication | n/a |

