## [Peer Review File · Nature]

SLC45A4 is a pain gene encoding a neuronal polyamine transporter

Corresponding Author: David Bennett

Version 0:

Reviewer comments:

Referee #1

(Remarks to the Author)

Middleton, Markusson, and colleagues present an unusually large effort that spans human GWAS, structural biology, and animal models to describe a channel linked to pain. This group uses human genetic data to identify the SLC45A4 gene locus as being associated with chronic pain. The authors use a series of deductive arguments to hypothesize that the Slc45a4 channel is a polyamine transporter, which is particularly relevant given the connection of polyamines to pain behaviors in both humans and animal models. To test this hypothesis, the authors present two key lines of evidence, one structural and one genetic. In the structural investigation, the authors present a CryoEM structure of the Slc45a4 channel. In the genetic investigation, the authors generate a Slc45a4 knockout mouse and then perform behavioral, electrophysiological, and anatomical characterizations, revealing alterations in thermal sensitivity and changes in the excitability of acutely dissociated IB4-negative neurons.

First and foremost, the authors should be commended for undertaking such an ambitious project. Characterizing a molecule from human genetics to structure to mouse genetic validation is unusual to see in a single study. The findings that implicate Slc45a4 in pain and somatosensation are certainly novel as well. However, I am concerned that the paper covers so much ground that it lacks a deeper investigation into the mechanisms at play. While I am not a structural biologist and therefore cannot fully evaluate the structural component, I can speak to the somatosensory biology aspects of the study. Below, I outline areas where the functional characterization of the mouse model could be made more rigorous.

1. The authors nicely show that Slc45a4 is expressed in the DRG as well as the spinal cord. The knockout mouse described in Figure 3 is well-made and validated; however, it is a total knockout. All the conclusions made in the paper seem to assume that the observed phenotypes must be due to expression in DRG neurons, but why must this be the case? Shouldn't a cell-type-specific knockout (or a similar approach) be performed to confirm this? This general issue of cell specificity could apply to all analyses of the DRG in this study.

2. The authors use tissue histology with BIII-Tubulin-FITC to claim that nerve fibers in the skin are unchanged between wild-type and knockout mice. While the quantification supports this, I argue that this evidence is insufficient. A more careful and comprehensive analysis of skin innervation is needed to convincingly show no change. For example, the authors could show that CGRP staining is not different between WT and KO. It is also worth noting that the representative image shown lacks hair follicle innervation. Given that a subset of CGRP fibers, particularly Adelta-HTMR, form circumferential endings around hair follicles and are implicated in pain (as shown by the Chesler and Ginty labs), it would be valuable to show whether these nerve endings change between WT and KO. 2a. The authors' description of skin innervation is somewhat imprecise. Both glabrous and hairy skin should be examined if they wish to claim no differences between WT and KO animals.

3. The electrophysiological characterization of DRG sensory neurons could be significantly more rigorous. 3a. Referring to IB4-negative neurons as 'peptidergic' is not entirely accurate. I suspect the authors intend to claim that the IB4-negative neurons are CGRP+, but there are certainly IB4-negative, CGRP-double-negative neurons. 3b. Characterizing spiking in dissociated DRG neurons in response to current injections is a starting point, but more could be done. For instance, how does mechanically and thermally induced spiking change between WT and KO neurons? 3c. The authors emphasize that Slc45a4 is a polyamine receptor, but none of the DRG neuron experiments address this directly. For example, are KO DRG neurons now unresponsive to polyamines? Which of the many interesting mutants of Slc45a4 can rescue excitability or possibly the behavioral deficits in the KO?

Minor points:

1. Perhaps this is just my interpretation, but doesn't the scRNA-seq data in Extended Data 5 suggest that Slc45a4 is not

broadly expressed in CGRP+ neurons? If anything, it seems that LTMR populations have higher levels of expression. I consider this a minor point, as the authors provide detailed characterization of Slc45a4 expression through in situ analysis, but some additional discussion on this would help clarify the findings for readers.

Referee #2

(Remarks to the Author)

The manuscript by Middleton, Markusson, et al. presents the association of a novel gene, namely SLC45A4, with chronic pain in humans, and further shows that the gene encodes a long-sought-after plasma membrane polyamine transporter. I am very enthusiastic about the work and I am confident that it will be of interest to researchers in several fields (including neuroscience and membrane transport), for it presents a novel gene associated to an important human disease, a novel function associated to the SLC45A4 gene, along with the determination of the first experimental structures of the transporter, and a novel target for pain treatment. The manuscript is clearly and concisely written, and the graphical part represents well the data.

I do have several concerns regarding the structural and transport data that the authors might want to consider for a re-submission:

TRANSPORT

-Both the Tm-shift and radioactive-SPD experiments convincingly show that polyamines, but not GABA or basic amino acids interact with SLC45A4. However, the radioactive-substrate experiments in cells, or at least the way they have been described in the manuscript, fall short of separating polyamine binding from transport. In other words, it is possible that polyamines, just like it occurs in K⁺ channels, bind to the intracellular cavity of SLC45A (where the "plug" is bound in the nanodisc structure) blocking the carrier and regulating its transport function (of a yet unknown substrate). Ideally, in order to demonstrate transport of a given substrate, one should know the number of carriers and make sure that several turnovers are measured, i.e. that several moles of substrate are measured per mole of carrier. Can the authors estimate the number of expressed transporters at the cell plasma membrane? Why didn't they use purified transporters reconstituted in synthetic liposomes? Patch clamp electrophysiology could also help, because transport, but not binding of polyamines will generate ionic currents.

In the absence of those data or approaches, additional experiments in the Neuro-2 cells including more stringent washes could serve to separate binding from transport. In the methods, it is mentioned that the wells are washed with 4x volumes of buffer without substrate. Have the authors explore other washing protocols? One way to increase the stringency of the washes could be to add an excess of cold substrate to the buffer, in order to displace the substrate that is bound, but not the one transported into the cytoplasm.

-In the competition assays, it is surprising to see that 100 μ M cold SPD only inhibits ~30% of the ¹⁴C-SPD signal (Ext Data Fig. 1c). One would expect that a 100-fold excess of cold over hot substrate would nearly abolish the radioactive signal. The fact that it doesn't suggests that the radioactive background in the cell system is significant. How is the background calculated and subtracted in the radioactive substrate experiments?

-Regarding the mutagenesis experiments, it would have been informative to mutate the basic residues in the plug. In the manuscript, it is reasonably assumed that the plug-bound state observed in the cryoEM structure is an "auto-inhibited" state. However, it could be that the plug binds at the substrate binding site to prevent re-binding of intracellular polyamines and therefore, that it is required for efficient cytoplasmic release and uptake of polyamines.

-There are some typos around line 142. The IC₅₀ values are given in mM instead of μ M.

STRUCTURE

-It is hard to judge without having the map and model, but the cryoEM density corresponding to the plug in the Ext Data Fig. 3d,e seems weak to unequivocally assign the amino acid sequence. Please, expand on how the plug sequence assignment was done and the degree of confidence on the assignment.

-The authors have enough structural information to speculate on the SLC45A4-mediated polyamine transport mechanism, but that is missing in the manuscript. In addition to the inward-facing state structures presented in the manuscript, there is an AlphaFold structure prediction of the outward-facing state. That structural information can be easily complemented with targeted docking of polyamine substrates to put forward a well-informed hypothesis on the transport mechanism. The authors barely mention that the nanodisc SLC45A4 structure suggests a substrate release mechanism that involves some membrane destabilization, as suggested for ATP13A2. In fact, the proposed ATP13A2 mechanism involves a polyamine channel through an ABC transporter (reference 33). From the AlphaFold model and the figures in the manuscript, it seems unlikely that SLC45A5 uses a channel-like mechanism to transport polyamines, and instead it uses a more classical alternating-access/rocker-switch mechanism.

-Further to the transport mechanism, it would be informative to know if there are structural changes in the conserved residues -those that might coordinate the substrate- between the cryoEM structures with (nanodisc) and without (LMNG) the plug bound, as well as compared to the outward-facing state prediction. In the latter, the strictly conserved R62 and R180 are at salt-bridge distance of the conserved acidic residues (E176, E173, E63) suggesting that they play key roles in gating and substrate recognition (see the picture attached). Do R62 and R180 arranged differently in the inward-facing states? Please, consider making a figure showing the differences.

-In Fig.2h, please indicate the distances between atoms.

Referee #3

(Remarks to the Author)

The study by Middleton, Markusson et al. identifies a long-undiscovered polyamine transporter, the identification of which is based on the question of the factors influencing human pain perception.

The identified transporter, encoded by SLC45A4, is comprehensively characterized both functionally and structurally, including a structural representation based on cryo-EM data, and its role in pain is further substantiated by gene expression data and in a Slc45a4 KO mouse model.

The study is highly comprehensive and innovative in many ways and the putative pain modulation via this newly identified pathway is of high interest and significance to a broad readership. The methodology is up to date and the manuscript is written clearly and concisely.

Major point: An intriguing link to the GABAergic and thus pain-modulating system is discussed in the work, but not experimentally deepened, and in my opinion, this is an important point that would need to be better clarified.

In the following, I will refer to the five main aspects on which the study is based:

1) First, a link between the SLC45A4 locus and chronic pain intensity is established using a genome-wide association study (GWAS). The phenotype is based on the "Enhanced Pain Phenotyping Questionnaires" used in the UK Biobank. The association is replicated in the Million Veteran Program, a multinational genetic study of pain intensity in approximately 600,000 veterans (SLC45A4 is among the 125 loci identified there). The GWAS of the Finnish Genetics Project (FinnGen) also confirmed the association between the SLC45A4 locus and pain. Finally, a strong association between SLC45A4 and chronic pain was detected at multiple sites using the Open Targets Genetics Locus-to-Gene pipeline. Overall, an association between the locus in question and pain perception is plausible, but requires downstream validation, which the authors address in subsequent steps.

Questions/comments:

A more detailed description of the central aspects of the "Enhanced Pain Phenotyping Questionnaires" would be helpful for the reader.

The authors consistently use the term "single nucleotide polymorphism" (SNP). The more neutral terminology "single nucleotide variant" (SNV) is increasingly being used, so it may be useful to adopt this terminology.

Since there is a coding SNV (p.(Asn718Asp)) in SLC45A4, the question arises: has this been functionally analyzed? Are there altered rates of polyamine transport? This would be interesting, although it is clear that due to the GWAS approach, a functional alteration does not necessarily have to result.

Additional designation of all hg38 coordinates would be very helpful for faster retrieval and correct interpretation of the rs numbers (see extended Table 1).

Human SLC45A4 was cloned and overexpressed based on the sequence Uniprot Q5BKX6 (768 amino acids), but does not appear to match the longer MANE transcript (https://www.ncbi.nlm.nih.gov/nucore/NM_001286646.2) (808 amino acids). Was there a reason to analyze the shorter isoform in depth, and is there data on putative differences in expression patterns of this and other putative isoforms (e.g., subcellular distribution, tissue expression, functional differences)? Are there transcript-specific recognizable differences in terms of predicted motifs, domains, etc.?

2) SLC45A4, whose function was previously unknown (the functions suggested in the literature do not appear to be correct), is finally and convincingly characterized as the "missing" polyamine transporter (especially for spermidine and spermine) at the cell membrane.

Questions/comments:

The reason for the functional investigation of SLC45A4 with regard to polyamine transport was the correlation with GABA and its alternative synthesis pathway. However, this connection is not investigated in the following, which is surprising. This should be better explained and clarified with functional means. The connection with the pain axis could be much better established by this.

Is there a functional connection to the GABA system based on the data collected? It would be logical to investigate this axis further, e.g. by electrophysiology in KO mice or the GABA-related protein expression and/or function in DRGs/spinal cord.

3) The structure of human SLC45A4 is revealed by cryo-electron microscopy, which contributes significantly to the understanding of the transporter.

Questions/comments: -

4) Consistent with a role of SLC45A4 in pain perception, there is convincing and broad mRNA expression in DRGs and

spinal cord.

Questions/comments:

Are there no commercial or self-made antibodies suitable for detection of the endogenous protein? A short comment would be helpful.

Can the authors provide more information on the role of the transporter in the brain? Do they suspect a role for the transport system in the brain in relation to pain modulation? According to Zeisel et al., the expression appears to be present. The postulated relationship with the GABA system at this CNS level should be commented on. This could lead to an additional influence on pain modulation.

5) SLC45A4 homozygous KO mice partially show altered pain behavior, such as reduced sensitivity to high temperatures and a reduction in their nociceptive behavior during the first phase of the formalin response.

Questions/comments:

The data are convincing. The question remains whether the polyamine levels in the serum of KO mice were analyzed. Are they altered?

General: Statistics are used appropriately. Please define error bars in the figure legends.

Version 1:

Reviewer comments:

Referee #1

(Remarks to the Author)

In this revised manuscript, the authors add several biochemical, biophysical, and neurobiological investigations to address the concerns of each reviewer.

In my initial review of the study, I had suggested a deeper dive into the behavioral and physiological characterization of the sensory neurobiology dimension of this study. As a main highlight, the authors now include recordings from a skin-nerve preparation, which revealed a deficit in C-MH polymodal nociceptor response thresholds to both mechanical and thermal forces. These results are consistent with the changes in current clamp recordings from isolated single neurons.

Overall, these are important and useful additions. There are some things I still find a bit confusing on the sensory neurobiology side.

1. In the response letter, the authors present the case, using CGRPCreER-tdtomato animals, that IB4-negative neurons must be mostly CGRP+. I am not sure I really understand the argument here. Similarly, I am not sure why the neurons relevant here need to be peptidergic. For example, there is a subtype of C-MH DRG neurons that are IB4/CGRP double negative (the Cyslr2+ population, described in PMID: 38442711). This subtype of DRG neurons respond to fairly high temperatures and mechanical forces, therefore could certainly fit the description of a subtype that could be important to the authors.

2. I still find the behavioral and physiological characterization to be a bit underdeveloped. Latency to withdrawal with different temperatures are nice to see, given the changes in thermal force thresholds, but the treatment of this dimension of the study reads somewhat trite. As one example, the behavioral effects seem somewhat subtle (no clear phenotype for mechanical force), yet the gene is clearly described as deeply linked to chronic pain. Relatedly, isn't it somewhat surprising that the channel is broadly expressed, yet the phenotypic consequences are so tightly restricted to a specific subpopulation of DRG neurons (C-MH fibers)? As alluded to by the authors, perhaps under different challenges (e.g. inflammation), the absence of this channel would reveal more somatosensory deficits at the behavioral or physiological level.

All this said, I also recognize that a major portion of this study is structural and human genetic oriented, which seem to be well executed to my eye. I could also understand the argument that the crux of the novelty here rests in defining an important new receptor that should be studied by others, with a starting point on behavioral/physiological measurements presented here.

Taken together, I do not want to hold up this paper, especially if the biochemical and biophysical dimensions of this study are favorably viewed by the other reviewers.

Referee #2

(Remarks to the Author)

I have no further comments to the authors

Referee #3

(Remarks to the Author)

My comments have been fully taken into account and in the revised version this highly exciting finding has been elaborated even more extensively.

Referee #4

(Remarks to the Author)

The manuscript by Middleton et al is an impressive journey from GWAS to extensive functional studies.

The text is balanced, avoiding overstatements. This reviewer finds the modest approach important, as the actual connection of SLC45A4 to clinical phenotypes remains to be understood. It would be surprising if this gene would be the key to understand chronic pain, to which the sentence in the introduction guides the reader.

The identification and verification of the genetic loci has been performed appropriately. The SLC45A4 signal is statistically significant in the UKB GWAS, although not extremely strong, but it is replicated in two independent studies, MVP and FinnGen. This is convincing. The effect sizes of the variants are quite modest.

The format of the text is quite short and thus does not provide much space for discussions about the findings, including its relevance for human pain phenotypes. It should be noted that e.g. the MVP pain intensity study identified 125 GWAS loci, here, in UKB, only two loci were genome wide significant, possibly due to a smaller data set compared to MVP. So, what the role of SLC45A4 as a contributor to pain phenotypes in human remains to be studied further.

The other point of discussion is that all mouse pain experiments have been done with a knockout model. This is an obvious pragmatic solution. However, it is hard to predict what variants are driving the signal in the GWAS, and consequently how appropriate a knockout model is. If the GWAS signals are driven by regulatory variants that contribute to the level of expression of SLC45A4, the knockout model might be of more relevance than in the case of variants that alter the structure of the protein product.

These are mostly points for discussions about the limitations of this quite comprehensive study.

Firstly, we would like to thank the reviewers for their positive outlook on our study, and their useful advice and feedback. We have addressed each comment below.

Referee #1

Middleton, Markusson, and colleagues present an unusually large effort that spans human GWAS, structural biology, and animal models to describe a channel linked to pain. This group uses human genetic data to identify the SLC45A4 gene locus as being associated with chronic pain. The authors use a series of deductive arguments to hypothesize that the Slc45a4 channel is a polyamine transporter, which is particularly relevant given the connection of polyamines to pain behaviors in both humans and animal models. To test this hypothesis, the authors present two key lines of evidence, one structural and one genetic. In the structural investigation, the authors present a CryoEM structure of the Slc45a4 channel. In the genetic investigation, the authors generate a Slc45a4 knockout mouse and then perform behavioral, electrophysiological, and anatomical characterizations, revealing alterations in thermal sensitivity and changes in the excitability of acutely dissociated IB4-negative neurons.

First and foremost, the authors should be commended for undertaking such an ambitious project. Characterizing a molecule from human genetics to structure to mouse genetic validation is unusual to see in a single study. The findings that implicate Slc45a4 in pain and somatosensation are certainly novel as well. However, I am concerned that the paper covers so much ground that it lacks a deeper investigation into the mechanisms at play. While I am not a structural biologist and therefore cannot fully evaluate the structural component, I can speak to the somatosensory biology aspects of the study. Below, I outline areas where the functional characterization of the mouse model could be made more rigorous.

Referee #1 (Remarks to the Author):

1. The authors nicely show that Slc45a4 is expressed in the DRG as well as the spinal cord. The knockout mouse described in Figure 3 is well-made and validated; however, it is a total knockout. All the conclusions made in the paper seem to assume that the observed phenotypes must be due to expression in DRG neurons, but why must this be the case? Shouldn't a cell-type-specific knockout (or a similar approach) be performed to confirm this? This general issue of cell specificity could apply to all analyses of the DRG in this study?

Thank you for raising this point. We have performed extensive further experiments which strengthen our argument that the locus of action is the DRG. The most compelling new data is detailed recordings of the stimulus-response function in distinct primary afferent sub-types, using the skin-nerve preparation (which is ideally suited to this analysis). This new data is included in a revised **Fig. 5** and **extended data Fig. 11**. The key finding is that a specific group of sensory neurons: C-fibre polymodal nociceptors (also termed CMH units) demonstrate significantly reduced responses to noxious mechanical and thermal stimuli in the *Slc45a4* KO mice. The responses of other primary afferent sub-types (C-mechanoreceptors, A δ high threshold mechanoreceptors and D-hairs as well as rapidly and slowly adapting A β low threshold mechanoreceptors) were unchanged. These findings are consistent with the cell autonomous reduction in soma excitability following current injection in IB4-negative small

diameter DRG neurons (discussed further below in relation to your comment 3a). These changes are also consistent with the reduced sensitivity to noxious heat on behavioural testing. The lack of changes in noxious mechanical withdrawal thresholds on behavioural testing likely represents redundancy due to normal mechano-sensitivity of other primary afferent sub-types.

Relating to the functional impact of SLC45A4 expression in other anatomical locations, we do observe SLC45A4 expression in spinal cord, albeit the expression was lower in dorsal versus ventral horn. As reviewer 3 points out the polyamine putrescine provides an alternative pathway for GABA synthesis. We indeed find reduced GABA levels in spinal cord, consistent with a dysregulation of polyamine transport. However, this reduction is in the ventral rather than dorsal horn (**Extended data Fig. 10, a**). For completeness, we also undertook dorsal spinal cord electrophysiology and found normal excitability of spinal dorsal horn neurons (normal resting membrane potential and rheobase) and normal inhibitory tone (normal amplitude and frequency of miniature inhibitory post-synaptic currents) see **Extended data Fig. 10 b-d**.

To summarise, by greatly extending our electrophysiological analysis on the global *Slc45a4* KO mouse. We see clear evidence of reduced nociceptive responses in C-polymodal nociceptors, consistent with the observed phenotypes. We can't entirely exclude that there may be additional subtle effects in somatosensory projection pathways, within the CNS, although we find no obvious effects at the level of the spinal cord. We take the point that ultimately generation of a conditional mouse would be useful for follow up studies but this is outside the scope of the current work (it would take 2 to 3 years to generate, breed and study and would refine rather than change our primary interpretations of the role of SLC45A4). We feel that we can make the firm conclusion that absence of SLC45A4 impacts on polymodal nociceptors which is internally consistent with the human genetic and mouse behavioural data.

2. The authors use tissue histology with BIII-Tubulin-FITC to claim that nerve fibers in the skin are unchanged between wild-type and knockout mice. While the quantification supports this, I argue that this evidence is insufficient. A more careful and comprehensive analysis of skin innervation is needed to convincingly show no change. For example, the authors could show that CGRP staining is not different between WT and KO. It is also worth noting that the representative image shown lacks hair follicle innervation. Given that a subset of CGRP fibers, particularly Adelta-HTMR, form circumferential endings around hair follicles and are implicated in pain (as shown by the Chesler and Ginty labs), it would be valuable to show whether these nerve endings change between WT and KO.

2a. The authors' description of skin innervation is somewhat imprecise. Both glabrous and hairy skin should be examined if they wish to claim no differences between WT and KO animals.

Thank you for this suggestion, which we have taken onboard. We have now extended the analysis of cutaneous innervation, including both glabrous and hairy skin. We have repeated this using an antibody to PGP9.5, which we find gives better definition of fine epidermal nerve terminals compared to Beta-3 Tubulin (see **revised Fig. 3 panels i,j**). We find no difference between WT and *Slc45a4* KO mice in the intra-epidermal nerve fibre density either in glabrous

(consistent with previous observations using a Beta3 tubulin antibody) or hairy skin. We also observed no difference in CGRP immunoreactive intra-epidermal nerve fibre density between WT and *Slc45a4* KO, in either glabrous or hairy skin. We have also studied the morphology of hair follicle innervation using neurofilament NF200 to identify lanceolate endings of low threshold mechanoreceptors and CGRP immunoreactive circumferential endings and these appear normal in *Slc45a4* KO mice (revised Fig. 3 panel K).

3. The electrophysiological characterization of DRG sensory neurons could be significantly more rigorous.

3a. Referring to IB4-negative neurons as ‘peptidergic’ is not entirely accurate. I suspect the authors intend to claim that the IB4-negative neurons are CGRP+, but there are certainly IB4-negative, CGRP-double-negative neurons.

We thank the referee for this clarification. We have now revised our text to be more accurate. We had sought to simplify for the reader, the point that the majority of small sized IB4-negative DRG neurons are peptidergic (expressing CGRP) based on the expression/binding data in DRG (reviewed in Snider and McMahon 1998 PMID: 9581756) and the previous use of live DRG staining using IB4 combined with electrophysiology as a useful means to differentiate nociceptor populations (for instance Stucky and Lewin 1999 PMID: 10414978). We agree that a minority of small sized DRG IB4 negative DRG neurons will also be negative for CGRP and are likely the small population of C-low threshold mechanoreceptors (and other discreet populations).

We have now explored these assumptions more directly using CGRP^{CreERT2}-tdTomato expressing mice to culture genetically labelled peptidergic nociceptors (we have recently shown that the CGRPCreERT2-tdTomato line captures 80% of the CGRP positive population, Barry et al., 2023, PMID 37318015). We incubated DRG cultures with IB4-488 and a live Cell Tracker and performed live cell imaging as we had done for electrophysiology. From this experiment we counted 933 small sized DRG neurons. 459 small cells bound IB4-488, and 140 small cells (15%) were negative for both tdTomato and IB4-488.

	IB4 488+	tdTomato+	IB4-488+ tdTomato+	Double Negative
%	33.01	34.94	17.04	15.01

Quantification of CGRP^{CREERT2}-tdtomato, IB4-488 live cell imaging. Small sized cells were analysed.

We determined that of the small neurons that were IB4-negative, 70% were tdTomato positive, confirming our assumption that the majority of small-sized IB4-negative neurons are peptidergic nociceptors. This affirms our claim that the majority of small-sized IB4 negative DRG neurons are peptidergic however, we acknowledge that this is a simplification. To be fully explicit on this matter, we now use the terms IB4-positive and negative in the results and then as a point of interpretation state that the majority of the IB4-negative DRG neurons represent peptidergic nociceptors without directly equating the two. We have amended the results to state:

‘and focused on two major nociceptor populations, those that bind IB4 (predominantly non-peptidergic nociceptors) and those that do not bind IB4 (which are predominantly but not exclusively peptidergic nociceptors, see methods) (Fig. 5a).’

We have also explained this nuance in the methods (see ‘methods Whole-cell patch-clamp’).

To be more specific on the important point as to which primary afferent sub-type is impacted in the *Slc45a4 KO mouse* and appreciating the fact that the most important issue is their functional properties we embarked on a whole new set of experiments using the skin-nerve preparation to define such populations (below). We hope this new data and text clarification addresses the reviewer’s suggestion.

3b. Characterizing spiking in dissociated DRG neurons in response to current injections is a starting point, but more could be done. For instance, how does mechanically and thermally induced spiking change between WT and KO neurons? 3c.

Thank you and we agree with this point and so we undertook single unit recordings using the skin-nerve preparation, which although time consuming, is ideal for assessing both threshold and suprathreshold responses in different sensory neuron sub-types because thermal and mechanical stimuli are delivered in a highly controlled manner. As described in detail in response to point 1 above, we find selective effects in terms of reduced responses of C polymodal nociceptors to noxious mechanical and thermal stimuli and this new data is now shown in **Fig. 5 and extended data Fig. 11**.

The authors emphasize that Slc45a4 is a polyamine receptor, but none of the DRG neuron experiments address this directly. For example, are KO DRG neurons now unresponsive to polyamines? Which of the many interesting mutants of Slc45a4 can rescue excitability or possibly the behavioral deficits in the KO?

We have now addressed the issue of assessing polyamine levels using 2D-gas chromatography mass spectrometry in a number of different tissues. This new data is now shown in **Fig. 3f**. We find significantly increased levels of spermidine in serum, reduced levels of spermidine in spinal cord and increased putrescine in DRG in *Slc45a4 KO* mice. We appreciate the caveat that in complex tissues such as DRG and spinal cord, it’s not possible to determine the relative concentration of polyamines in the extracellular versus intracellular compartments and the relative contributions of different cell types. The plasma results do however show that there is a change in distribution between compartments. Therefore, in line with our identification of SLC45A4 as a polyamine transporter *in vitro*, we have found a change in polyamine homeostasis *in vivo*.

We have explored whether fluorescently tagged polyamines can be used to assess cellular uptake and trafficking, providing cellular resolution. However, fluorescent tags are bulky and we have found in our heterologous expression systems these fluorescently tagged versions of polyamines are not transported by SLC45A4 and so not appropriate for this purpose. We have investigated whether wild type DRG neurons respond to external application of exogenous polyamines using calcium imaging and we do not see a clear response, so we do not think this would be a helpful assay in this instance.

Acute application of polyamines to WT cultured DRG neurons. Ca²⁺ imaging was used to screen responding cells to acute application of increasing concentrations of different polyamines (PUT, SPM, SPD – respectively). KCl was used to stimulate all neurons as a positive control stimulus.

Our interpretation is that SLC45A4 is likely regulating intracellular polyamines to impact on excitability, although we can't exclude the possibility of more tonic effects (which we don't model with acute application), could also be having an effect. The issue of comparing SLC45A4 mutants is an excellent suggestion, but would ideally use a knock-in mouse approach and would be the topic of future experiments.

Minor points:

Perhaps this is just my interpretation, but doesn't the scRNA-seq data in Extended Data 5 suggest that Slc45a4 is not broadly expressed in CGRP+ neurons? If anything, it seems that LTMR populations have higher levels of expression. I consider this a minor point, as the authors provide detailed characterization of Slc45a4 expression through in situ analysis, but some additional discussion on this would help clarify the findings for readers.

Overall, our interpretation of this data is that it shows that SLC45A4 is expressed by CGRP+ve DRG neurons but not exclusively and we agree it is expressed by LTMRs. The Zeisel et al., data shows that SLC45A4 is expressed across a range of primary afferent sub-types at a higher level than CNS neurons. There are conflicting data, when comparing expression in LTMRs versus peptidergic nociceptors in these datasets; in some cases, expression is higher in LTMRs, and in others, in peptidergic nociceptors. There are technical differences between these studies. We have found that, whilst these large single cell RNAseq datasets are undoubtedly helpful (we often use them as a point of reference), they are not always entirely accurate probably related to the depth of sequencing in single cell studies. It is for this reason that we undertook our own analysis using in situ hybridization. To clarify this for the reader we have added the following at the end of the figure legend for **extended data Fig. 5**:

'Our overall interpretation of these datasets is that SLC45A4 mRNA is enriched in DRG neurons versus CNS neuronal populations. There are some inconsistencies between datasets as to the relative expression in distinct sensory neuron subpopulations although our own analysis using in situ hybridization (below) supports the interpretation of broad expression across sensory neuron subpopulations.'

Referee #2 (Remarks to the Author):

The manuscript by Middleton, Markusson, et al. presents the association of a novel gene, namely SLC45A4, with chronic pain in humans, and further shows that the gene encodes a long-sought-after plasma membrane polyamine transporter. I am very enthusiastic about the work and I am confident that it will be of interest to researchers in several fields (including neuroscience and membrane transport), for it presents a novel gene associated to an important human disease, a novel function associated to the SLC45A4 gene, along with the determination of the first experimental structures of the transporter, and a novel target for pain treatment. The manuscript is clearly and concisely written, and the graphical part represents well the data.

I do have several concerns regarding the structural and transport data that the authors might want to consider for a re-submission:

TRANSPORT

-Both the Tm-shift and radioactive-SPD experiments convincingly show that polyamines, but not GABA or basic amino acids interact with SLC45A4. However, the radioactive-substrate experiments in cells, or at least the way they have been described in the manuscript, fall short of separating polyamine binding from transport. In other words, it is possible that polyamines, just like it occurs in K⁺ channels, bind to the intracellular cavity of SLC45A (where the “plug” is bound in the nanodisc structure) blocking the carrier and regulating its transport function (of a yet unknown substrate). Ideally, in order to demonstrate transport of a given substrate, one should know the number of carriers and make sure that several turnovers are measured, i.e. that several moles of substrate are measured per mole of carrier. Can the authors estimate the number of expressed transporters at the cell plasma membrane? Why didn't they use purified transporters reconstituted in synthetic liposomes? Patch clamp electrophysiology could also help, because transport, but not binding of polyamines will generate ionic currents. In the absence of those data or approaches, additional experiments in the Neuro-2 cells including more stringent washes could serve to separate binding from transport. In the methods, it is mentioned that the wells are washed with 4x volumes of buffer without substrate. Have the authors explore other washing protocols? One way to increase the stringency of the washes could be to add an excess of cold substrate to the buffer, in order to displace the substrate that is bound, but not the one transported into the cytoplasm.

We thank the referee for these detailed comments and advice on this aspect of the study. As the reviewer can no doubt acknowledge, reconstitution of human transporters into liposomes and optimization of conditions, such as lipid composition, is challenging. Often the result is loss of transport activity in reconstituted liposome systems. We have attempted to measure SPD transport in SLC45A4 liposomes. Although the transporter reconstituted well into liposomes of various lipid compositions, we have not yet been able to measure robust liposomal uptake despite extensive efforts. Moreover, we deemed quantification of plasma membrane incorporated SLC45A4 in Neuro2A cells impractical and prone to high errors, especially considering the varying transfection efficiency and activity observed in our assay. Therefore, we turned to the excellent suggestion of employing more stringent washing approaches.

Three different wash strategies were utilized (see **Figure below**). Firstly, the pH was altered (**a**), as changing pH should affect binding through weakening of charge-charge interactions between the transporter and SPD. Secondly, increasing the salt concentration (**b**) should similarly dissociate bound SPD. Finally, we washed the Neuro2A cells with 0.1 mM and 1 mM cold SPD (**c**), respectively, corresponding to a 100- and 1000-fold excess of the 1 μM ^{14}C -SPD used in the assay.

We observed no decrease in the signal when washing the cells either twice or five times with 500 μL of these competing buffers. A slight overall reduction was observed with five washes, likely due to increased detachment of cells from the plate.

We thank the referee for their suggestion of these experiments as these new data strengthen our central hypothesis that SLC45A4 is a PA transporter. We have now added this additional data to **Extended Data Fig. 1** and included the washing protocols in the methods section.

*Stringent washes in Neuro2A ^{14}C -SPD uptake assay. **a** Washes at varying pH, **b** increased ionic strength and **c** with an excess of cold SPD. In all cases, WT HsSLC45A4 was overexpressed in Neuro2A cells for 40-48 hr, cells washed in assay buffer, and transport quenched through washing with ice-cold buffer following 10 min incubation in 240 μL of 1/9 μM hot/cold SPD. Washes were carried out with 500 μL buffer, either twice or five times.*

To further examine the question of binding and transport, we purified seven binding site mutants and carried out thermal shift assays to estimate their capacity of binding SPD.

In our transport assays, mutation of several binding site residues in SLC45A4 seen in proximity to the plug domain abolished SPD transport activity. These include all residues in the acidic ladder on TM4; Asp169, Asp173 and Glu176A and additionally Glu63, Tyr66 and Trp519. Alanine substitution of these side chains, in addition to combinations of substitutions (see below), were purified and subjected to SPD-thermal shift assays (see **Figure below**). Whilst some of the substitutions showed a negative impact on SPD binding, importantly the double substitution (D169A/E63A) retains the ability to respond to SPD (indicating binding), but individually these substitutions ablate transport activity. These results therefore strengthen our original hypothesis that SLC45A4 is a PA transporter.

SPD thermal shift assay on purified SLC45A4 mutants. **a** Plug binding as observed in the 2.83 Å LMNG:CHS structure of SLC45A4. **b** Dose response of binding site mutants in SPD thermal shift titrations. A clear reduction of binding is observed in Y66A/E176A while other transport inactive mutants, most notably D169A/E63A, appear to still bind SPD.

-In the competition assays, it is surprising to see that 100 μ M cold SPD only inhibits ~30% of the 14C-SPD signal (Ext Data Fig. 1c). One would expect that a 100-fold excess of cold over hot substrate would nearly abolish the radioactive signal. The fact that it doesn't suggests that the radioactive background in the cell system is significant. How is the background calculated and subtracted in the radioactive substrate experiments?

The competition assays were carried out on Neuro2A overexpressing WT *HsSLC45A4* and as pointed out by the reviewer, the background signal in this assay is significant. This is almost certainly due to the presence of endogenous SLC45A4 and potentially to some extent endocytosis and/or binding to plasma membrane proteins. Due to variability and the high background in the assay, each experiment was performed with WT vs. cells transfected with empty plasmid. If the WT uptake level did not exceed 2.5-fold that of the empty plasmid control, the result from these wells was discarded. Due to the high background level, we chose not to carry out any background subtraction as it might introduce additional noise in the data. We also consider this a more transparent way to present the data/result. The data is sufficient for comparing inhibition within this assay to compare the different lengths of PAs against each other.

-Regarding the mutagenesis experiments, it would have been informative to mutate the basic residues in the plug. In the manuscript, it is reasonably assumed that the plug-bound state observed in the cryoEM structure is an "auto-inhibited" state. However, it could be that the plug binds at the substrate binding site to prevent re-binding of intracellular polyamines and therefore, that it is required for efficient cytoplasmic release and uptake of polyamines.

To address this point we substituted both Lys450 and Arg453 to aspartic acid and measured their impact on transport. We did not observe an impact on activity, indicating that these side chains do not play a role in the transport mechanism. This result is consistent with these side chains playing a role in occupying the binding site rather than having a direct role in PA uptake or recognition. We have included this data in the revised Ms in **Fig. 2i**.

-There are some typos around line 142. The IC50 values are given in mM instead of uM. Thank you these have now been corrected.

STRUCTURE

-It is hard to judge without having the map and model, but the cryoEM density corresponding to the plug in the Ext Data Fig. 3d,e seems weak to unequivocally assign the amino acid sequence. Please, expand on how the plug sequence assignment was done and the degree of confidence on the assignment.

In the initial 2.8 Å LMNG:CHS structure of *HsSLC45A4*, the region of the plug preceding the two plugging residues, Lys450 and Arg453, was sufficient to allow for distinguishing between side chains (please see **Figure below**). The region was initially identified through sequence alignment, in which the sequence of the plug domain is considerably better conserved than surrounding regions of the ICD (**Extended Data Fig. 1g**). Furthermore, initial *ab initio* model building of the plug was carried out independently by two of the authors (Markússon and Deme), resulting in two identical models of the plug domain. We have included the maps and models in our submission, which the reviewer is very welcome to inspect in more detail.

CryoEM density (2.8 Å LMNG:CHS map) of the plug domain agrees well with the refined structure.

-The authors have enough structural information to speculate on the SLC45A4-mediated polyamine transport mechanism, but that is missing in the manuscript. In addition to the inward-facing state structures presented in the manuscript, there is an AlphaFold structure prediction of the outward-facing state. That structural information can be easily complemented with targeted docking of polyamine substrates to put forward a well-informed hypothesis on the transport mechanism. The authors barely mention that the nanodisc SLC45A4 structure suggests a substrate release mechanism that involves some membrane destabilization, as suggested for ATP13A2. In fact, the proposed ATP13A2 mechanism involves a polyamine channel through an ABC transporter (reference 33). From the AlphaFold model and the figures in the manuscript, it seems unlikely that SLC45A4 uses a channel-like mechanism to transport polyamines, and instead it uses a more classical alternating-access/rocker-switch mechanism.

We thank the reviewer for sharing our interest in the mechanism of transport, gating and plug domain regulation. While we agree the potential mechanism is not explained in the paper, we have tried to avoid comparison of our experimental structure with the AlphaFold (AF) prediction for two reasons. The first is that while the comparison to AF models can be informative, this study focuses on the identification of *SLC45A4* as a new pain gene, rather than trying to explain the mechanism of a previously identified transporter. Our structural, biophysical and transport assay data demonstrate that SLC45A4 is a member of the Major Facilitator Superfamily, that it is expressed in cells and can transport PAs. Given the scope of this study, we feel that a more in-depth mechanistic study would be better placed in context via a separate, mechanistically focused study. Our second reason for not including this comparison is that a complete understanding of the transport mechanism requires the use of either in vitro reconstitution systems and/or electrophysiology assays. Without these tools in place, ideally combined with an experimentally determined outward facing conformations, our view is that speculating about the alternating access mechanism at this stage is premature at this level of publication. We hope the referee can see our perspective on this valid point.

-Further to the transport mechanism, it would be informative to know if there are structural changes in the conserved residues -those that might coordinate the substrate- between the cryoEM structures with (nanodisc) and without (LMNG) the plug bound, as well as compared to the outward-facing state prediction. In the latter, the strictly conserved R62 and R180 are at salt-bridge distance of the conserved acidic residues (E176, E173, E63) suggesting that they play key roles in gating and substrate recognition (see the picture attached). Do R62 and R180 arranged differently in the inward-facing states? Please, consider making a figure showing the differences.

There were no large conformational changes observed between the structures solved in detergent and nanodiscs (**Extended Data Fig. 4a**). The plug domain was slightly better resolved in the nanodisc map and slight differences were observed in the electron density for the C-terminal end of the domain, which leads out of the binding site through the open membrane-facing cavity between TM8 and TM5. As no specific contacts were observed between the plug and ordered lipid molecules, this is likely indicative of the dynamic nature of the domain rather than of a specific mechanism. However, slight differences were observed in the side chains rotamers of the plug domain (highlighted in **Extended Data Fig. 4c**). In terms

of mechanism of membrane deformation, observed in the nanodisc structure, it potentially facilitates substrate release/binding by either exposing the active site to the cytosol or by releasing substrates onto the inner leaflet of the membrane, for which charged polyamines could have affinity. However, as discussed above, given the focus of this study is the identification of SLC45A4 as a new pain gene and demonstrating the link to PA transport, our preference with respect to the mechanism of PA transport is to determine an experimental structure bound with PA before speculating on a mechanism in this study.

-In Fig.2h, please indicate the distances between atoms.

We agree with the referee that bond distances are very helpful in these schematic diagrams. However, given the size of the figure panel we were unable to add in the distances in a font size that was readable when printed. We hope the referee can forgive this omission.

Referee #3

The study by Middleton, Markusson et al. identifies a long-undiscovered polyamine transporter, the identification of which is based on the question of the factors influencing human pain perception.

The identified transporter, encoded by SLC45A4, is comprehensively characterized both functionally and structurally, including a structural representation based on cryo-EM data, and its role in pain is further substantiated by gene expression data and in a *Slc45a4* KO mouse model.

The study is highly comprehensive and innovative in many ways and the putative pain modulation via this newly identified pathway is of high interest and significance to a broad readership. The methodology is up to date and the manuscript is written clearly and concisely.

Major point: An intriguing link to the GABAergic and thus pain-modulating system is discussed in the work, but not experimentally deepened, and in my opinion, this is an important point that would need to be better clarified.

Thank you for this point and yes as noted in the manuscript, there are two broad mechanisms for GABA synthesis: the common synthesis pathways from glutamate via glutamate decarboxylase enzymes or an alternative pathway via the degradation of the polyamine putrescine. The spinal dorsal horn is a key location regulating the processing of nociceptive signals by GABA (Knabl et al., 2008, PMID: 18202657). We have now measured GABA levels and assessed an electrophysiological correlate of GABA in the form of inhibitory neurotransmission within the dorsal horn. This new data is now shown in **extended data Fig. 10**. We found that there was a reduction of GABA at whole spinal cord level in *Slc45a4* KO. However, looking at sub-regions, this reduction was in the ventral horn and not the dorsal horn (**extended data Fig. 10**). In addition, there was no significant difference between WT and *Slc45a4* KO mice in inhibitory tone in the form of miniature inhibitory post-synaptic current (iPSC) amplitude or frequency.

The significant reduction in GABA in the ventral spinal cord of *Slc45a4* KO mice is noteworthy. This is consistent with the expected directionality of SLC45A4 importing polyamines, and indeed, we found higher expression of SLC45A4 in ventral horn compared to dorsal horn (**extended data Fig. 6**). We have been struck by the fact that on the rotarod assay *Slc45a4* KO mice show an increased latency to fall (**Fig. 4a**). It is interesting to speculate, as to whether this improved motor function could relate to less GABA mediated inhibition.

To summarise, we do not think GABA mediates the change in pain behaviour (and also note that a reduction in GABA would normally be associated with enhanced not reduced pain related behaviour). The reduced GABA in the ventral horn, could, however relate to improved motor endurance. Further investigation of altered inhibition within the ventral spinal cord is outside the scope of this current work but certainly of interest for future studies.

In the following, I will refer to the five main aspects on which the study is based:

1) First, a link between the SLC45A4 locus and chronic pain intensity is established using a genome-wide association study (GWAS). The phenotype is based on the “Enhanced Pain Phenotyping Questionnaires” used in the UK Biobank. The association is replicated in the Million Veteran Program, a multinational genetic study of pain intensity in approximately 600,000 veterans (SLC45A4 is among the 125 loci identified there). The GWAS of the Finnish Genetics Project (FinnGen) also confirmed the association between the SLC45A4 locus and pain. Finally, a strong association between SLC45A4 and chronic pain was detected at multiple sites using the Open Targets Genetics Locus-to-Gene pipeline. Overall, an association between the locus in question and pain perception is plausible, but requires downstream validation, which the authors address in subsequent steps.

Questions/comments:

A more detailed description of the central aspects of the “Enhanced Pain Phenotyping Questionnaires” would be helpful for the reader.

Thank you for raising this point. We agree that a more detailed description of this aspect of the study, will improve clarity for the reader. We have now included the following data in the supplementary information:

‘Enhanced Pain Phenotyping Questionnaire details

UK Biobank comprises 501,518 volunteers recruited between 2006 and 2010 from across Great Britain. During the initial self-assessment of the recruited participants a brief set of questions screening for the presence, intensity and localisation of chronic pain were completed. This set did not include validated questionnaires and did not differentiate between neuropathic pain from non-neuropathic pain.

In 2019 a detailed “Experience of Pain” self-assessment questionnaire was administered to all UKB participants that had valid email addresses as part of the electronic follow-up (335,587). Developed with input from leading experts and harmonised with large national and international cohorts (Hébert et al., 2023) this questionnaire uses validated questionnaires to ensure a detailed and accurate assessment of pain. By gathering specific information on pain location, intensity, quality, and related conditions like depression and fatigue, the questionnaire provides a richer understanding of chronic pain (Baskozos et al., 2023).

It covers several key areas through different sections, including:

- Screening for medical conditions that frequently lead to the development of chronic pain
- Screening for chronic pain (> 3months) and detailed location and intensity of pain for each one of the listed body sites. Identification and intensity for the most bothersome pain
- Self-completed screening tool for Neuropathic Pain (DN4 (Bouhassira et al., 2005)) related to the most bothersome pain
- Self-reported detailed pain inventory (BPI (Cleeland & Ryan, 1994))

- Self-completed screening tool for Neuropathy (MNSI (Feldman et al., 1994)) if participants reported cancer pain, diabetes, or nerve damage other than diabetic neuropathy
- Health related quality of life (EQ-5D-5L (Herdman et al., 2011))
- Self-completed psychosocial questionnaires about depression (PHQ (Kroenke et al., 2001)) and fatigue (FSS (Krupp et al., 1989))

Baskozos, G., Hébert, H. L., Pascal, M. M. V., Themistocleous, A. C., Macfarlane, G. J., Wynick, D., Bennett, D. L. H., & Smith, B. H. (2023). Epidemiology of neuropathic pain: An analysis of prevalence and associated factors in UK Biobank. *Pain Reports*, 8(2). <https://doi.org/10.1097/PR9.0000000000001066>

Bouhassira, D., Attal, N., Alchaar, H., Boureau, F., Brochet, B., Bruxelle, J., Cunin, G., Fermanian, J., Ginies, P., Grun-Overdyking, A., Jafari-Schluep, H., Lantéri-Minet, M., Laurent, B., Mick, G., Serrie, A., Valade, D., & Vicaut, E. (2005). Comparison of pain syndromes associated with nervous or somatic lesions and development of a new neuropathic pain diagnostic questionnaire (DN4). *Pain*, 114(1–2), 29–36. <https://doi.org/10.1016/J.PAIN.2004.12.010>

Cleeland, C. S., & Ryan, K. M. (1994). Pain assessment: global use of the Brief Pain Inventory. *Annals of the Academy of Medicine, Singapore*, 23(2), 129–138. <https://europepmc.org/article/med/8080219>

Feldman, E. L., Stevens, M. J., Thomas, P. K., Brown, M. B., Canal, N., & Greene, D. A. (1994). A Practical Two-Step Quantitative Clinical and Electrophysiological Assessment for the Diagnosis and Staging of Diabetic Neuropathy. *Diabetes Care*, 17(11), 1281–1289. <https://doi.org/10.2337/DIACARE.17.11.1281>

Hébert, H. L., Pascal, M. M. V., Smith, B. H., Wynick, D., & Bennett, D. L. H. (2023). Big data, big consortia, and pain: UK Biobank, PAINSTORM, and DOLORisk. *PAIN Reports*, 8(5), e1086. <https://doi.org/10.1097/PR9.0000000000001086>

Herdman, M., Gudex, C., Lloyd, A., Janssen, M., Kind, P., Parkin, D., Bonse, G., & Badia, X. (2011). Development and preliminary testing of the new five-level version of EQ-5D (EQ-5D-5L). *Quality of Life Research*, 20(10), 1727–1736. <https://doi.org/10.1007/S11136-011-9903-X/TABLES/5>

Kroenke, K., Spitzer, R. L., & Williams, J. B. W. (2001). The PHQ-9. *Journal of General Internal Medicine*, 16(9), 606–613. <https://doi.org/10.1046/J.1525-1497.2001.016009606.X>

Krupp, L. B., Larocca, N. G., Muir-Nash, J., & Steinberg, A. D. (1989). The Fatigue Severity Scale: Application to Patients With Multiple Sclerosis and Systemic Lupus Erythematosus. *Archives of Neurology*, 46(10), 1121–1123. <https://doi.org/10.1001/ARCHNEUR.1989.00520460115022>

The authors consistently use the term “single nucleotide polymorphism” (SNP). The more neutral terminology “single nucleotide variant” (SNV) is increasingly being used, so it may be useful to adopt this terminology.

Thank you for this point and we have changed this terminology in line with this suggestion.

Since there is a coding SNV (p.(Asn718Asp)) in SLC45A4, the question arises: has this been functionally analyzed? Are there altered rates of polyamine transport? This would be interesting, although it is clear that due to the GWAS approach, a functional alteration does not necessarily have to result.

In our experimental structures, Asn718 could be built into the maps. However, Asn718 is located on the disordered C-terminal portion of the protein, closely following the C-terminal end of TM12 (**see figure below**). To help establish a functional role for this variant, we substituted Asn718 for Ala/Asp/Arg/Trp and measured SPD uptake in Neuro2A cells. However, no apparent differences in activity were observed. As the referee suggests, this variant may not exhibit significant functional effects on SLC45A4 activity as measured in our current assay. GWAS signals detect common variants, which by their nature, often have small effect sizes, which can be challenging to assess in functional assays. However, the proximity of Asn718 to the detergent micelle and nanodisc membrane suggests a role in sensing changes in the membrane, which is a line of thought we are currently pursuing. We have discussed this point in the results and included this new data in **Extended Data Fig. 4e**.

Additional text in results:

‘Finally, Asn718 (the GWAS missense variant) is located on the cytoplasmic side of the membrane (Extended Data Fig. 4e). Substitution of Asn718 to Ala/Asp/Arg/Trp did not result in a change in transport activity (Extended Data Fig. 4e). Common variants identified in GWAS have small effect sizes which can be difficult to detect in functional assays (which were undertaken in the NEURO2A cell line); alternatively, his variant might impact another aspect of SLC45A4 function in the cell.’

The N718D variant identified in the pain GWAS does not seem to have an implication on the structure and function of HsSLC45A4. Top: Structure of SLC45A4 in detergent, Asn718 is shown as spheres and highlighted. The position of the residue in relation. Bottom: Single timepoint measurement of ¹⁴C-SPD transport of Asn718 mutants in Neuro2A cells (n≥6), which showed close to no change in activity compared to WT SLC45A4.

Additional designation of all hg38 coordinates would be very helpful for faster retrieval and correct interpretation of the rs numbers (see extended Table 1).

Thank you for this point we have now done this.

Human SLC45A4 was cloned and overexpressed based on the sequence Uniprot Q5BKX6 (768 amino acids), but does not appear to match the longer MANE transcript (https://www.ncbi.nlm.nih.gov/nucore/NM_001286646.2) (808 amino acids). Was there a reason to analyze the shorter isoform in depth, and is there data on putative differences in expression patterns of this and other putative isoforms (e.g., subcellular distribution, tissue expression, functional differences)? Are there transcript-specific recognizable differences in terms of predicted motifs, domains, etc.?

The shorter isoform was chosen as it is the main isoform sequence provided in the UniProt database (Q5BKX6) which has been reviewed and manually curated by UniProt. We note that there is an attempt to harmonise canonical protein isoforms in UniProt and MANE transcripts but these assignments are still not always the same (as the reviewer has identified here).

An alignment of the two transcripts is shown below. The two differ only by a C-terminal extension region in the MANE transcripts. This region has no recognisable functional motifs and is not confidently predicted by AF to be structured (see Figure below). We therefore, would not expect the MANE transcript to show different subcellular localisation or PA transport activity, or that the structure of the MANE transcript would differ to that of Q5BKX6.

AlphaFold3 prediction of the SLC45A4 MANE transcript **a** coloured by pLDDT score (confidence of prediction) and **b** to highlight the C-terminal extension in the MANE transcript.

```

Q5BKX6  MKMAPQNADPESMQVQELSVPLPDPQKAGGAEAEENCETISEGSIDRIPMRLWVMHGAVMF  60
MANE    MKMAPQNADPESMQVQELSVPLPDPQKAGGAEAEENCETISEGSIDRIPMRLWVMHGAVMF  60
*****

Q5BKX6  GREFCYAMETALVTPILLQIGLPEQYYSLTWFLSPILGLIFTPLIGSASDRCTLSWGRRR  120
MANE    GREFCYAMETALVTPILLQIGLPEQYYSLTWFLSPILGLIFTPLIGSASDRCTLSWGRRR  120
*****

Q5BKX6  PFILALCVGVLFVGFVFLNGSAIGLALGDVPNRQPIGIVLTVLGVVVLDFSADATEGPIR  180

```

MANE PFILALCVGLFGVALFLNGSAIGLALGDVPNRQPIGIVLTVLGVVLDIFSADATEGPIR 180

Q5BKX6 AYLLDVVDSEEQDMALNIHAFSAGLGGAIGYVLGGLDWTQTFLGSWFRTQNQVLFFFAAI 240
 MANE AYLLDVVDSEEQDMALNIHAFSAGLGGAIGYVLGGLDWTQTFLGSWFRTQNQVLFFFAAI 240

Q5BKX6 IFTVSVALHLFSIDEEQYSPQERSAEEPGALDGGEPHGVPAPFDEVQSEHELALDYPDV 300
 MANE IFTVSVALHLFSIDEEQYSPQERSAEEPGALDGGEPHGVPAPFDEVQSEHELALDYPDV 300

Q5BKX6 DIMRSKSDSALHVPDTALDLEPELLFLHDIEPSIFHDASYPATPRSTSQELAKTKLPLRA 360
 MANE DIMRSKSDSALHVPDTALDLEPELLFLHDIEPSIFHDASYPATPRSTSQELAKTKLPLRA 360

Q5BKX6 TFLKEAAKEDETLLDNHLNEAKVPNGSGSPTKDALGGYTRVDTKPSATSSSMRHHAFR 420
 MANE TFLKEAAKEDETLLDNHLNEAKVPNGSGSPTKDALGGYTRVDTKPSATSSSMRHHAFR 420

Q5BKX6 RQASSTFSYYGKLGSHCYRRRANAVVLIKPSRMSDLYDMQKRQRQHRHRNQSGATTSS 480
 MANE RQASSTFSYYGKLGSHCYRRRANAVVLIKPSRMSDLYDMQKRQRQHRHRNQSGATTSS 480

Q5BKX6 GDTSEEGEGETTVRLLWLSMLKMPRELMRLCLCHLLTWFSVIAEAVFYDFMGQVIFEG 540
 MANE GDTSEEGEGETTVRLLWLSMLKMPRELMRLCLCHLLTWFSVIAEAVFYDFMGQVIFEG 540

Q5BKX6 DPKAPSNSTAWQAYNAGVKMGCWGLVIYAATGAICSALLQKYLDNYDLSVRVIYVLTGLG 600
 MANE DPKAPSNSTAWQAYNAGVKMGCWGLVIYAATGAICSALLQKYLDNYDLSVRVIYVLTGLG 600

Q5BKX6 FSVGTAVMAMFPNVVYVAMVTISTMGIVSMSISYCPYALLGQYHDIKQYIHHSPGNSKRGF 660
 MANE FSVGTAVMAMFPNVVYVAMVTISTMGIVSMSISYCPYALLGQYHDIKQYIHHSPGNSKRGF 660

Q5BKX6 GIDCAILSCQVYISQILVASALGGVVDVAVGTVRVIPMVASVGSFLGFLTATFLVIYPNVS 720
 MANE GIDCAILSCQVYISQILVASALGGVVDVAVGTVRVIPMVASVGSFLGFLTATFLVIYPNVS 720

Q5BKX6 EEAKKEEQKGLSSPLAGEGRAGGNSEKPTVLKLRKEGLQGPVETESV----- 768
 MANE EEAKKEEQKGLSSPLAGEGRAGGNSEKPTVLKLRKEGLQGPVETESVTPAGIDVCQISSH 780

Q5BKX6 ----- 768
 MANE WLVPQLLESIFLYDYFRKKIFFSTMWFS 808

In terms of isoform abundance, the key point is that these two isoforms show almost identical patterns of expression and are much more abundant than any other isoforms. The UniProt database (QB5KX6) canonical isoform, is associated with the ENST00000519067 transcript. The canonical MANE transcript is ENST00000517878. GTEX holds isoform level expression data showing that both isoforms are expressed with an almost identical pattern across a range of tissues and at a higher level than the other isoforms.

Isoform Expression of SLC45A4: ENSG0000022567.10 solute carrier family 45 member 4 [Source:HGNC Symbol;Acc:HGNC:29196]

Our overall interpretation is that there is no compelling reason to choose one isoform over the other for structural studies and that the structural outcomes for Q5BKX6 will be highly informative for the MANE transcript isoform. Finally, we would like to point out that both isoforms are ablated in the *Slc45a4* KO mouse.

2) SLC45A4, whose function was previously unknown (the functions suggested in the literature do not appear to be correct), is finally and convincingly characterized as the “missing” polyamine transporter (especially for spermidine and spermine) at the cell membrane.

Questions/comments:

The reason for the functional investigation of SLC45A4 with regard to polyamine transport was the correlation with GABA and its alternative synthesis pathway. However, this connection is not investigated in the following, which is surprising. This should be better explained and clarified with functional means. The connection with the pain axis could be much better established by this. Is there a functional connection to the GABA system based on the data collected? It would be logical to investigate this axis further, e.g. by electrophysiology in KO mice or the GABA-related protein expression and/or function in DRGs/spinal cord.

We have undertaken further experiments to study this more closely (as described above in response to your first ‘major point’). GABA levels and inhibitory neurotransmission in the dorsal horn of the spinal cord are normal in *Slc45a4* KO mice and as such, we think it unlikely GABA is contributing to the pain phenotype. We have now undertaken further detailed electrophysiological analysis of sensory axons using the skin-nerve preparation and, we find

hypo-excitability of C-fibre polymodal nociceptors (**see response to point 1 of reviewer 1 and updated Fig. 5 which includes this data**). This does not mean that SLC45A4 mediated polyamine transport and regulation of GABA synthesis would not be relevant in other contexts. Indeed, the reduction of GABA in the ventral spinal cord is an interesting finding and, given that there may be a behavioural correlate with motor function, this is an area for future studies (albeit out of scope of the current manuscript).

3)The structure of human SLC45A4 is revealed by cryo-electron microscopy, which contributes significantly to the understanding of the transporter.

Questions/comments: -

The reviewer had no queries on this point.

4) Consistent with a role of SLC45A4 in pain perception, there is convincing and broad mRNA expression in DRGs and spinal cord.

Questions/comments:

Are there no commercial or self-made antibodies suitable for detection of the endogenous protein? A short comment would be helpful.

We have tried a number of antibodies and also tried to generate our own. However, despite extensive effort have not been able to find an antibody that detects the endogenous protein with adequate specificity at least for immunostaining. The most telling point is that we see immunostaining of DRG neurons using these antibodies in WT but this staining is also present in *Slc45a4* KO mice. We know that all major transcripts are very effectively ablated in these mice indicated non-specific binding of these antibodies. In summary, we have not yet found an antibody in which we have confidence.

Can the authors provide more information on the role of the transporter in the brain? Do they suspect a role for the transport system in the brain in relation to pain modulation? According to Zeisel et al., the expression appears to be present. The postulated relationship with the GABA system at this CNS level should be commented on. This could lead to an additional influence on pain modulation.

We agree that the brain does express *Slc45a4* mRNA, as does spinal cord. We did not find a difference in polyamine or GABA levels in the whole brain (the spinal cord outcomes are described in detail above and commented on in the main manuscript in Fig. 3). We acknowledge the point that further studies would be needed to determine if there are some subtle changes in anatomical sub-regions of the brain which we would not have detected in this global analysis.

5) SLC45A4 homozygous KO mice partially show altered pain behavior, such as reduced sensitivity to high temperatures and a reduction in their nociceptive behavior during the first phase of the formalin response.

Questions/comments:

The data are convincing. The question remains whether the polyamine levels in the serum of KO mice were analyzed. Are they altered?

Thank you for this suggestion, which we agree is relevant. To answer this question, we used 2D-gas chromatography mass spectrometry to assess polyamine levels in a number of different tissues. This new data is shown in **Fig. 3f**. We find significantly increased levels of spermidine in serum, reduced levels of spermidine in spinal cord and increased putrescine in DRG in *Slc45a4* KO mice. We appreciate the caveat that in complex tissues (such as DRG and spinal cord), it's not possible to determine the relative concentration of polyamines in the extracellular versus intracellular compartments nor the relative contributions of different cell types. The plasma results do however show that there is a change in distribution between compartments and consistent with our identification of SLC45A4 as a polyamine transporter *in vitro* we have found a change in polyamine homeostasis *in vivo*.

General: Statistics are used appropriately. Please define error bars in the figure legends.

Thank you and we have now made this change.

First, we would like to thank the reviewers for their positive evaluation of our study, and their constructive advice and feedback. We have addressed each comment below.

Referee #1 (Remarks to the Author):

In this revised manuscript, the authors add several biochemical, biophysical, and neurobiological investigations to address the concerns of each reviewer.

In my initial review of the study, I had suggested a deeper dive into the behavioral and physiological characterization of the sensory neurobiology dimension of this study. As a main highlight, the authors now include recordings from a skin-nerve preparation, which revealed a deficit in C-MH polymodal nociceptor response thresholds to both mechanical and thermal forces. These results are consistent with the changes in current clamp recordings from isolated single neurons.

Overall, these are important and useful additions.

Thank you. We agree that these new findings significantly advance our understanding of how SLC45A4 ablation impacts pain behaviour.

There are some things I still find a bit confusing on the sensory neurobiology side.

1. In the response letter, the authors present the case, using CGRPCreER-tdtomato animals, that IB4-negative neurons must be mostly CGRP+. I am not sure I really understand the argument here. Similarly, I am not sure why the neurons relevant here need to be peptidergic. For example, there is a subtype of C-MH DRG neurons that are IB4/CGRP double negative (the Cysltr2+ population, described in PMID: 38442711). This subtype of DRG neurons respond to fairly high temperatures and mechanical forces, therefore could certainly fit the description of a subtype that could be important to the authors.

We suggest that the majority of IB4-negative small DRG neurons are CGRP-positive and this is supported by our data: of small IB4-negative DRG neurons 70% are tdtomato positive using CGRP reporter mice (CGRPCreER-tdtomato as demonstrated in revision 1). We are not suggesting that IB4-negative are all CGRP positive and we have been careful with our wording in order to fairly represent the data:

In the main body of the manuscript:

‘and focused on two major nociceptor populations: those that bind IB4 (predominantly non-peptidergic nociceptors) and those that do not bind IB4 (which are predominantly but not exclusively peptidergic nociceptors, see methods) (Fig. 5a).’

We agree with the reviewer’s point that some small DRG neurons will be double negative for IB4 and CGRP. We had noted the presence of double negative small DRG neurons in the methods and state ‘We counted 933 small sized neurons. 459 small cells bound IB4-488, and 140 small cells (15%) were negative for both tdTomato and IB4-488.’ To expand on this point and the specific reference the reviewer makes to the Cysltr2+ population we have revised the text:

In the methods:

‘Using this we defined small (diameter < 25 µm) IB4-positive neurons as predominantly non-peptidergic nociceptors and small IB4-negative neurons as predominantly peptidergic nociceptors. This definition is not clear cut there will be some overlap and some cells that are negative for both markers (i.e. C-LTMRs and *Cysltr2* positive nociceptors PMID: 38442711).’

We agree with the reviewer that although IB4 is a pragmatic way of broadly defining nociceptors it is an approximation and this is why we went on to undertake extensive skin-nerve recordings to specifically look at the impact of SLC45A4 on the stimulus-response function of nociceptors which has more direct relevance to nociceptive behaviour and finding a specific effect on C-polymodal nociceptors.

2. I still find the behavioral and physiological characterization to be a bit underdeveloped. Latency to withdrawal with different temperatures are nice to see, given the changes in thermal force thresholds, but the treatment of this dimension of the study reads somewhat trite. As one example, the behavioral effects seem somewhat subtle (no clear phenotype for mechanical force), yet the gene is clearly described as deeply linked to chronic pain. Relatedly, isn't it somewhat surprising that the channel is broadly expressed, yet the phenotypic consequences are so tightly restricted to a specific subpopulation of DRG neurons (C-MH fibers)? As alluded to by the authors, perhaps under different challenges (e.g. inflammation), the absence of this channel would reveal more somatosensory deficits at the behavioral or physiological level.

In terms of linking this gene to pain and specifically chronic pain we would make the point that this is one of only a few examples in the literature in which there is both human genetic data linking gene variants specifically to chronic pain (with a discovery cohort and two replication cohorts in humans) as well as preclinical data in a mouse model. There are precedents of important pain genes which when ablated alter thermal and not mechanical pain related behaviour. A good example is TRPV1 which showed altered latency responses on hot plate at the higher end of the noxious range (52.5 but not 50 degrees centigrade) and had no effect on mechanosensation (PMID: 10764638). In fact, the ablation of SLC45A4 has much more striking effects on nociceptor function as assessed by primary afferent recordings than the ablation of TRPV1 (PMID 10764638 and PMID: 15254097). One interesting finding is that although we find clear changes in noxious mechanosensation in C-polymodal nociceptors the mechanical pain related behaviour was unchanged and this is likely due to normal mechanosensitivity of other primary afferent populations. We point this out in the discussion:

‘Mechanical pain responses remained intact following SLC45A4 loss, likely due to the preserved function of other mechanoreceptor populations.’

We agree with the observation that the SLC45A4 transporter is broadly expressed but the phenotype (in terms of sensory function) is restricted to C-fibre polymodal nociceptors. The most parsimonious explanation for this specificity is that polyamines (transported by SLC45A4) are interacting with ion channels and it is the restricted expression of these polyamine sensitive ion channels that is responsible for the specificity of the impact on C-polymodal nociceptors. We have added a sentence to this effect in the discussion.

'Polyamines are known to functionally modulate a number of ion channels that may be candidates for impacting on excitability of nociceptors including: the Transient Receptor Potential family (TRPV1, 3,4)^{14,15}, inward rectifier K⁺⁴³ and acid-sensitive ion channels (ASICs)⁴⁴. **Our observation, that SLC45A4 specifically impacts the C-polymodal nociceptor population may reflect the expression pattern of polyamine sensitive channels in these neurons.'**

We completely agree with the reviewer that under different challenges such as inflammation we may reveal further somatosensory deficits and this will be the subject of future studies.

All this said, I also recognize that a major portion of this study is structural and human genetic oriented, which seem to be well executed to my eye. I could also understand the argument that the crux of the novelty here rests in defining an important new receptor that should be studied by others, with a starting point on behavioral/physiological measurements presented here.

We appreciate and agree with this perspective; the mouse data should be seen in the context of the human genetic and structural data and we are keen to disseminate this to the wider community.

Taken together, I do not want to hold up this paper, especially if the biochemical and biophysical dimensions of this study are favorably viewed by the other reviewers.

Thank you.

Referee #2 (Remarks to the Author):

I have no further comments to the authors

Thank you.

Referee #3 (Remarks to the Author):

My comments have been fully taken into account and in the revised version this highly exciting finding has been elaborated even more extensively.

Thank you.

Referee #4 (Remarks to the Author):

The manuscript by Middleton et al is an impressive journey from GWAS to extensive functional studies.

The text is balanced, avoiding overstatements. This reviewer finds the modest approach important, as the actual connection of SLC45A4 to clinical phenotypes remains to be understood. It would be surprising if this gene would be the key to understand chronic pain, to which the sentence in the introduction guides the reader.

As noted, we have made every effort to be balanced in our approach. Our perspective is that the human genetics has been used to identify a genetic association which leads to new mechanistic understanding both for the role of SLC45A4 as a polyamine transporter and as a relevant pathophysiological mechanism for chronic pain. It is this mechanistic insight that we are highlighting rather than saying that genetic variation in the SLC45A4 locus explains a large amount of the variation in pain intensity in the population.

The identification and verification of the genetic loci has been performed appropriately. The SLC45A4 signal is statistically significant in the UKB GWAS, although not extremely strong, but it is replicated in two independent studies, MVP and FinnGen. This is convincing. The effect sizes of the variants are quite modest.

We agree that this is a robust association which has been independently replicated in two cohorts and that the effect size is relatively modest (as is often the case in GWAS). We have made an effort not to overstate this (see above) but the impact is heightened by our discovery that this is a polyamine transporter which will have wider implications for instance potentially invigorating broader research into how polyamines alter nociceptive processing.

The format of the text is quite short and thus does not provide much space for discussions about the findings, including its relevance for human pain phenotypes. It should be noted that e.g. the MVP pain intensity study identified 125 GWAS loci, here, in UKB, only two loci were genome wide significant, possibly due to a smaller data set compared to MVP. So, what the role of SLC45A4 as a contributor to pain phenotypes in human remains to be studied further.

Thank you for these points and we agree that the assigned word count means that the text has to be short without extensive discussion. We agree that the greater number of GWAS loci identified in MVP likely relate to the larger cohort size (and there may also be advantages of a multi-ancestry approach). We have added a comment to this effect in the main body of the manuscript.

‘Having identified a genetic link between *SLC45A4* and pain perception in the UKB- data, we then analysed data from the Million Veteran Program (MVP) in which a recent multi-ancestry genetic study of pain intensity in approximately 600,000 veterans identified 125 independent genetic loci, one of which was *SLC45A4*²⁵ (the greater number of independent loci identified in MVP, compared to our UKB study, may reflect the increased statistical power due to a larger cohort size and the multi-ancestry approach).

The other point of discussion is that all mouse pain experiments have been done with a knockout model. This is an obvious pragmatic solution. However, it is hard to predict what variants are driving the signal in the GWAS, and consequently how appropriate a knockout model is. If the GWAS signals are driven by regulatory variants that contribute to the level of expression of SLC45A4, the knockout model might be of more relevance than in the case of variants that alter the structure of the protein product. These are mostly points for discussions about the limitations of this quite comprehensive study.

Thank you appreciating the comprehensive nature of our study. We agree that at the moment we have variants that are associated with altered SLC45A4 expression and also a missense variant and it is hard to be definitive as to which is most relevant to human pain (this will require future studies). We have used the knockout mouse as a means to validate SLC45A4 as a pain gene at a behavioural level and to provide mechanistic insight both showing that polyamine levels change *in vivo* (an important finding in light of our biochemical data) that and reduced excitability of C-polymodal nociceptors. We hope you can appreciate the rationale behind our approach and to address the point raised by the reviewer we have added a sentence in the discussion in relation to future studies:

‘SLC45A4 is expressed in sensory neurons and ablation of this function results in altered polyamine homeostasis, thermal coding and pain perception in mice. **Future studies will be needed to assess the impact of specific coding variants on both the regulation of polyamine transport and nociceptive function.** The behavioural changes **in *Slc45a4* KO mice** are accompanied by a marked reduction in the excitability of C-polymodal nociceptors which are known to encode thermal pain and the response to many algogens.’